Impact of topography and meteorological forcing on snow simulation in the
Canadian Land Surface Scheme Including Biogeochemical Cycles (CLASSIC)
Libo Wang[1], Lawrence Mudryk[1], Joe R. Melton[2], Colleen Mortimer[1], Jason Cole[3], Gesa Meyer[2],
Paul Bartlett[1], and Mickaël Lalande[4,5]
1 Climate Processes Section, Climate Research Division, Environment and Climate Change Canada,
Toronto, ON, Canada
2 Climate Processes Section, Climate Research Division, Environment and Climate Change Canada,
Victoria, BC, Canada
3 Canadian Centre for Climate Modelling and Analysis, Climate Research Division, Environment
Canada, Victoria, BC, Canada
4 Centre for Research on Watershed-Aquatic Ecosystem Interactions, Environmental Sciences
Department, Université du Québec à Trois-Rivières, Trois-Rivières, QC, Canada
5 Centre for Northern Studies, Université Laval, QC, Canada
Corresponding author: Libo Wang, libo.wang@ec.gc.ca

**Abstract**

Our study evaluates the impacts of an alternate snow cover fraction (SCF) parameterization on snow simulation in the Canadian Land Surface Scheme Including Biogeochemical Cycles (CLASSIC). Three reanalysis-based meteorological datasets are used to drive the model to account for uncertainties in the forcing data. While the default parameterization assumes a simple linear relationship between SCF and snow depth with no dependence on topography, the alternate parameterization accounts for the topographic effects of sub-grid terrain on SCF. We show that the alternate parameterization improves SCF simulated in CLASSIC during winter and spring in mountainous areas for all three choices of meteorological datasets. Annual mean bias, unbiased root mean squared area, and correlation improve by 75 %, 32 %, and 7 % when evaluated with MODIS SCF observations over the Northern Hemisphere. We also demonstrate that the improvements to simulated SCF lead to further improvements in variables related to surface radiation, energy fluxes, and the water cycle. Finally, we link relative biases in the meteorological forcing data to differences in simulated snow water equivalent and SCF. Assessment of simulations with different combinations of SCF parameterizations and meteorological datasets reveals the large impact of meteorological forcing on snow simulation in CLASSIC. Two out of the three meteorological datasets were bias-adjusted using observation-based datasets. However, simulations forced by the dataset without bias correction outperform relative to simulations forced by datasets with bias correction, suggesting that there are large uncertainties in the observation-based datasets and/or methods used for bias correction. This study underscores the importance of accounting for topographic effects of sub-grid terrain and accurate meteorological forcing on snow simulation in land surface models.

## 1. Introduction

Snow cover exists from six to nine months of the year at the high latitudes and high elevations of mountainous regions. The seasonal transition from snow covered to snow free conditions can have a large impact on the stability of permafrost, the length of the active growing season, and surface water and energy balances due to the much higher albedo of snow cover than other land surfaces (e.g., Myneni et al., 1997; Betts et al., 1998; Osterkamp and Romanovsky, 1999; Frolking et al., 2006). Snow cover plays an important role in the regional and global climate system because of the snow-albedo feedback mechanism (Fletcher et al., 2009; Qu and Hall, 2013). Any uncertainty in the magnitude of this climate feedback decreases our ability to reduce uncertainty in climate sensitivity (Roe and Baker, 2007). Therefore, accurate simulation of snow cover is crucial for future climate predictions in climate and Earth system models (ESMs).

In principle, snow depth (SND) should vary considerably at sub-grid scales of global climate models as a result of multiple heterogeneities in land cover, terrain, and meteorological conditions (Liston 2004). Most land surface models (LSMs) explicitly treat only some of this heterogeneity, for example by accounting for different land cover types within a grid cell (Verseghy et al., 2017). Snow cover fraction (SCF) parameterizations are commonly used to account for unresolved (sub-grid scale) snow depth variability. However, most models from the

Coupled Model Intercomparison Project (CMIP) phase 5 (Taylor et al., 2012) and phase 6
(Eyring et al., 2016) have been found to overestimate SCF in mountainous regions, often with a
corresponding cold bias in surface air temperature (Su et al., 2013; Lalande et al., 2021). These
biases are also present in the most recent Canadian Earth System Models (CanESM5, Swart et
al., 2019; Sigmond et al., 2023) and the latest version of its land surface component, the
Canadian Land Surface Scheme Including biogeochemical Cycles (CLASSIC, Melton et al.,
2020; Seiler et al., 2021). The SCF overestimation has been attributed to many potential causes,
such as too much precipitation and/or overly simplistic SCF parameterizations in ESMs (Lalande
et al., 2021; Miao et al., 2022).
Some early SCF parameterizations assumed a linear increase in snow cover with snow depth or
snow water equivalent (SWE), reaching 100% SCF once a specified threshold was met (e.g.,
Verseghy, 1991; Bonan, 1996). Other approaches incorporated surface roughness length into the
SCF–SND (or SWE) relationships (e.g., Dickinson et al., 1986; Marshall and Oglesby, 1994),
and distinguished SCF estimates between bare ground and vegetated areas (Douville et al., 1995;
Yang et al., 1997). Large uncertainties in modeled SCF from these early schemes motivated
efforts to refine parameterizations by accounting for terrain heterogeneity or incorporating sub-
grid snow distribution (Roesch et al., 2001; Liston, 2004).  More recent SCF parameterizations
have included snow density (e.g., Niu and Yang, 2007; Lalande et al., 2023) and land cover type
(e.g., He et al., 2023), with some schemes adopting separate formulations for snow accumulation
and melt periods (Swenson and Lawrence, 2012). Some of these parameterizations account for
topographic effects of sub-grid terrain on SCF (e.g., Douville et al., 1995; Roesch et al., 2001;
Swenson and Lawrence, 2012; Lalande et al., 2023), which have been shown to be crucial for
accurate SCF simulation in mountainous regions (Miao et al., 2022).
In CLASSIC, the default parameterization historically used is a linear relationship between SCF
and SND with no dependence on topography. A grid cell is considered fully snow-covered when
the diagnosed SND reaches 0.1 m (Verseghy, 1991). Melton et al. (2019) investigated the impact
of two alternative SCF parameterizations on SCF and permafrost area simulated by CLASSIC.
The first was to change the SCF-SND linear relationship to a hyperbolic tangent function (Yang
et al., 1997), and the second was to change the SCF-SND linear form to an exponential form
(Brown et al., 2003). Both alternative SCF parameterizations worsened performance in terms of
the global permafrost area and active layer thickness, so that neither was implemented.
Here we consider another option previously developed by Swenson and Lawrence (2012). Their
parameterization (hereafter referred as SL12) qualitatively reproduces the hysteresis present in
the observational data (SCF-SND relationship) between snow accumulation and ablation seasons
while also accounting for the topographic effects of sub-grid terrain. The SL12 parameterization
was implemented in the Community Land Model version 5 (CLM5, Lawrence et al., 2019), the
land surface component in the Community Earth System Model version 2 (CESM2,
Danabasoglu et al., 2020). Notably, CESM2 was one of the models that showed the lowest
surface air temperature and SCF biases over the High Mountain Asia (HMA) region among the
CMIP6 models (Lalande et al., 2021). Based on these results, the SL12 parameterization was
implemented in the CLASSIC model and here we evaluate the impact of this change on SCF,
SWE, and other snow-related land surface variables. Our evaluation is based on offline
CLASSIC simulations forced by historical temperature and precipitation from reanalyses.
Because there is uncertainty in these historical values, especially in mountainous regions, we use
three different reanalysis-based meteorological datasets to drive CLASSIC. For each
meteorological forcing datasets we perform two CLASSIC simulations, one with the default SCF
parameterization and one with the SL12 parameterization. The two parameterization schemes are
compared with observed SCF and SWE, and the other snow-related land surface variables are
evaluated using the Automated Model Benchmarking R package (AMBER, Seiler et al., 2021).
The remainder of this paper is organized as follows. In Section 2, we describe the CLASSIC
model, the two SCF parameterizations, the forcing data, and model setup. In Section 3, we
describe the observation data and our evaluation methods. Results are detailed in Section 4 and
discussion points in Section 5.  We present conclusions in Section 6.

**2. CLASSIC model, SCF parameterization methods, and model setup**
**2.1 CLASSIC description and snow model characteristics**
CLASSIC is an open-source community land model that is designed to address research
questions that explore the role of the land surface in the climate system. It is the successor to the
coupled modelling framework based on the Canadian Land Surface Scheme (CLASS; Verseghy,
1991; 1993) and the Canadian Terrestrial Ecosystem Model (CTEM; Arora and Boer, 2005;
Melton and Arora, 2016). The physics and biogeochemistry modules of CLASSIC are based on
CLASS and CTEM models, respectively. Older versions of CLASSIC are under the name
CLASS-CTEM. The CLASSIC model simulations can be performed at point, regional, and
global scales both in coupled and offline modes. CLASSIC has been applied in an offline
context, i.e. forced with observed meteorology (e.g. Bailey et al., 2000; Bartlett et al., 2006;
Melton et al., 2019), as the physical land surface component of regional climate models, e.g.
CRCM (Wang et al., 2014; Ganji et al., 2015) and CanRCM (Scinocca et al., 2016), and
integrated into each version of the Canadian Atmospheric Model (CanAM;  von Salzen et al.,
2013), and Earth System Model (CanESM;  Arora et al., 2011; Swart et al., 2019) since the early
1990s.
The physics component of CLASSIC models energy and water balances separately for the
vegetation canopy, snow, and soil (Verseghy, 1991; Melton et al., 2019). As a first-order
treatment of subgrid-scale heterogeneity, each grid cell is divided into four sub-areas: vegetated,
bare soil, vegetated with snow cover, and snow cover over bare soil. Snow is represented as a
single layer, and canopy snow processes such as interception, unloading, sublimation and melt
are included (Bartlett et al., 2006; Verseghy et al., 2017). The grid cell albedo is computed as a
weighted mean based on the fractional coverages for each surface type. In previous versions of
CLASSIC, the snow albedo decreases exponentially with time from fresh snow values according
to empirically derived functions (Verseghy, 1991). In more recent versions, a new physics-based
snow albedo parameterization is available, which accounts for contributions of black carbon
snow mixing ratio and the effective snow grain size on snow albedo (Namazi et al., 2015). The
new snow albedo scheme is the default scheme in CanESM models and is used in this study.
Further details on the CLASSIC model can be found in Melton et al. (2020).

## 2.2 SCF parameterization methods


### 2.2.1 The current default SCF parameterization

In CLASSIC, the thicknesses of all layers (snow and soil) are recommended to be greater than
0.1 m to avoid numerical instability problems. Therefore, the local SND over the snow-covered
portion of a grid cell is not allowed to decrease below this threshold (0.1 m), instead, the
fractional snow cover decreases to conserve snow mass. Snow cover is considered complete
when SND reaches 0.1 m; when SND < 0.1 m, SCF is computed as SCF = SND/0.1, and SND is
reset to 0.1 m. Hereafter we refer to the current default SCF parameterization as the Control
(CTL) parameterization. Previous analysis has shown that increasing or decreasing this threshold
value by 50 % has little effect on the simulated SWE or SCF (Verseghy et al., 2017).

### 2.2.2 The SL12 SCF parameterization

Based on snow cover datasets at relatively high spatial and temporal resolution, Swenson and
Lawrence (2012) demonstrated that the relationship between SCF and SND depends not only on
the amount of snow, but also whether snow mass is increasing (accumulation) or decreasing
(ablation). This dependence is hypothesized to stem from differences in how accumulation
versus ablation processes alter the correlation of the two variables. Based on this they proposed
separate formulations for snow accumulation and melt periods as follows.
During snow accumulation:

$$f_{sno}^n = 1 - ((1 - \tanh(k_{acc}\Delta W))(1 - f_{sno}^{n-1}) \qquad (1)$$

Where $f_{sno}^n$ and $f_{sno}^{n-1}$ are SCF from the current and the previous time step, $k_{acc}$ is a scale
parameter (mm$^{-1}$) and $\Delta W$ (mm) is the amount of new snow that falls within the current time
step. Eq. (1) assumes that precipitation is randomly distributed across the region, which may be
questionable in mountainous areas where snowfall tends to preferentially accumulate at higher
elevations. Nevertheless, SCF simulated using the SL12 parameterization from coarse-resolution
climate models shows reasonable agreement with observations (e.g. Lalande et al., 2023). Note
Eq. (1) is the formulation used in CLM5 code (and implemented in CLASSIC), which is
different from that in Swenson and Lawrence (2012). In most LSMs including CLASSIC, SND
is diagnostically computed through snow water equivalent (W in Eq. (1)-(4)) and snow density
($\rho_s$): SND=W/$\rho_s$. Swenson and Lawrence (2012, their Fig. 7) illustrated that the rate of SCF
increase with SND depends on the $k_{acc}$ parameter, such that a larger $k_{acc}$ parameter would result
in faster SCF increase with SND. The default value from Swenson and Lawrence 2012) is 0.1
mm$^{-1}$, which is also used in our study. The impact of this choice will be discussed in Section 5.2.
During snowmelt:
$$f_{sno} = 1 - \left[\frac{1}{\pi}\text{acos}\left(2\frac{W}{W_{max}} - 1\right)\right]^{N_{melt}}$$
(2)

$$N_{melt} = \frac{200}{max(10, \; \sigma_{topo})}$$
(3)

$$W_{max} = \frac{W}{0.5\left(cos\left(\pi(1-f_{sno})^{\frac{1}{N_{melt}}}\right)+1\right)}$$
(4)

where the W and $W_{max}$ are the current and the maximum accumulated snow water equivalent
(mm), and $N_{melt}$ (unitless) is a parameter determined from the standard deviation of topography,
$\sigma_{topo}$ (m). Eq. (4) is used to reconcile the relationship during periods of mixed accumulation and
melt. Eq. (2) and Eq. (3) suggest that the rate of SCF decrease with SND depends on the $N_{melt}$
parameter, such that SCF decreases faster with (normalized) SND in mountainous areas (small
$N_{melt}$) than flat areas (large $N_{melt}$, Fig.9 in Swenson and Lawrence, 2012).
In our implementation we do not distinguish the use of these two formulations by time of year
but based on whether SWE is increasing or decreasing with respect to the previous time step
(Wang et al., 2025). To avoid the numerical instability issues mentioned above (Section 2.2.1),
the SL12 parameterization is only used when the local SND over the snow-covered portion of a
grid cell is greater than 0.1 m. When SND < 0.1 m, SCF is computed in the same way as in the
default parameterization. Therefore, the largest difference in SCF between the default and SL12
parameterization as implemented in CLASSIC is expected in mountainous areas during the melt
period. In these regions and times the topographic effects of sub-grid terrain are accounted for in
SL12 but not in CTL.

### 210    2.3 Forcing data and simulation setup

The modeling domain chosen for this study is a global land-only latitude-longitude grid at 1º
resolution (Fig. 1a). Three gridded meteorological datasets are used to drive CLASSIC in this
study: CRUJRA, ERA5, and GSWP3-W5E5, described below. CRUJRA is regularly used to
drive LSMs participating the annual Global Carbon Project which provides analysis of the land
carbon sink (Friedlingstein et al., 2025). It was constructed by regridding data from the Japanese
reanalysis (JRA, Kobayashi et al., 2015) and adjusting where possible to align with the Climatic
Research Unit (CRU) TS4 data (Harris, 2020; 2023). The blended product spanning January
1901 to December 2020 has the 6-hourly temporal resolution of the JRA reanalysis product but
monthly means adjusted to match the CRU data at 0.5º spatial resolution.
ERA5 is the fifth generation European Centre for Medium-Range Weather Forecasts atmospheric
reanalysis of the global climate covering the period from January 1940 to present (Hersbach et
al., 2020). ERA5 data are available at hourly temporal and 0.25º spatial resolution. Currently it
has the highest spatial and temporal resolutions available among all global reanalysis products.
GSWP3-W5E5 (here after referred as GSWP3W5) is a combination of two datasets: GSWP3
v1.09 (Dirmeyer et al., 2006; Kim 2017) from 1901-1978 and W5E5 v2.0 (Cucchi et al. 2020;
Lange et al. 2021) from 1979-2019. It is one of the forcings used in the Inter-Sectoral Impact
Model Intercomparison Project (ISIMIP). The GSWP3 dataset is a dynamically downscaled
version of the Twentieth Century Reanalysis version 2 (20CRv2; Compo et al. 2011), bias-
corrected using three separate observational data sets (see Kim 2017 for details). The W5E5
dataset is an interpolated version of ERA5 reanalysis, bias-corrected using CRU TS4.  W5E5
also provides a second set of precipitation forcing data, bias-corrected with observations from the
Global Precipitation Climatology Project (GPCP; Adler et al., 2003). The GPCP dataset includes
around 3–4 times as many precipitation stations as CRU, thus we use this version of the
precipitation forcing in our experiments. The GSWP3W5 data are available at daily temporal and
0.5º spatial resolution.
The three meteorological forcing datasets are regridded using the first order conservative
remapping method to the 1º model grid via Climate Data Operators. They are disaggregated on
the fly within CLASSIC into half-hourly data following the methodology of Melton and Arora
(2016) for the following seven meteorological variables that are used to force the model: 2 m air
temperature, total precipitation, specific humidity, downward solar radiation flux, downward
longwave radiation flux, surface pressure, and wind speed. In CLASSIC, the phase of
precipitation is determined by a threshold surface air temperature according to three possible
options described in (Bartlett et al., 2006). Jennings et al. (2018) showed that the snowfall-
rainfall transition temperature varied from -0.4°C to 2.4°C across the NH. Based on this, we used
the option where the partitioning between rainfall and snowfall varies linearly between all
rainfall at temperatures above 2º C, and all snowfall at temperatures below 0°C.
The plant functional types used in CLASSIC are derived from the Climate Change Initiative land
cover product produced by the European Space Agency (Wang et al., 2023). The atmospheric
$CO_2$ concentration values are provided by the Global Carbon Project (Le Quere et al., 2018). The
soil texture information consists of the percentage of sand, clay, and organic matter and is
derived from the SoilGrids250m dataset (Hengl et al., 2017), and the permeable soil depth is
based on Shangguan et al. (2017).
CLASSIC simulations use either the CTL or the SL12 parameterization forced by the CRUJRA,
ERA5, and GSWP3W5 respectively, yielding six simulations over the historical period. We refer
to these simulations hereafter as: CRUJRA-CTL, CRUJRA-SL12, ERA5-CTL, ERA5-SL12,
GSWP3W5-CTL, and GSWP3W5-SL12. Pre-industrial spin-up simulations were performed to
allow the model to equilibrate carbon fluxes to conditions corresponding to the first year of the
forcing data. During spin-up, we loop climate data from the earliest 25 years available for
CRUJRA/ERA5 and 100 years of spin-up data for GSWP3W5 (Lange et al., 2022), and hold
atmospheric CO2 concentrations at the pre-industrial level (286.46 ppm). The transient runs use
time-varying CO2 concentrations and climate. The period from 2005 to 2014 is selected for
analyzing the simulated results, when there is overlap with the three observational SCF datasets
(see Section 3.1).

**3. Observation data and evaluation methods**

### 3.1 Study area and evaluation methods

Our analysis will include evaluation of SCF, SWE, meteorological forcings, and other land surface variables. Assessment of SCF, SWE, and meteorological forcings will focus on the mountain ($\sigma_{topo}$ > 200 m) and flat ($\sigma_{topo}$ <= 200 m) regions over the Northern Hemisphere (NH), and sub-regions of North America (NA), Eurasia (EA), and HMA. Classification of mountain and flat regions is based on standard deviation of the sub-grid terrain from the ETOPO1 elevation data at 1 arc-minute resolution (NOAA, 2009, Fig. 1a). In the SL12 parameterization, the topographic effects of sub-grid terrain are considered via the Nmelt parameter (Eq. (2)), which is inversely related to $\sigma_{topo}$ (Eq. (3)). Figure 1a shows that at 1° resolution, the magnitudes of $\sigma_{topo}$ are around 200 m – 600 m for most of the mountainous regions except for the HMA and the Andes where the magnitude of $\sigma_{topo}$ can reach 1200 m or more.

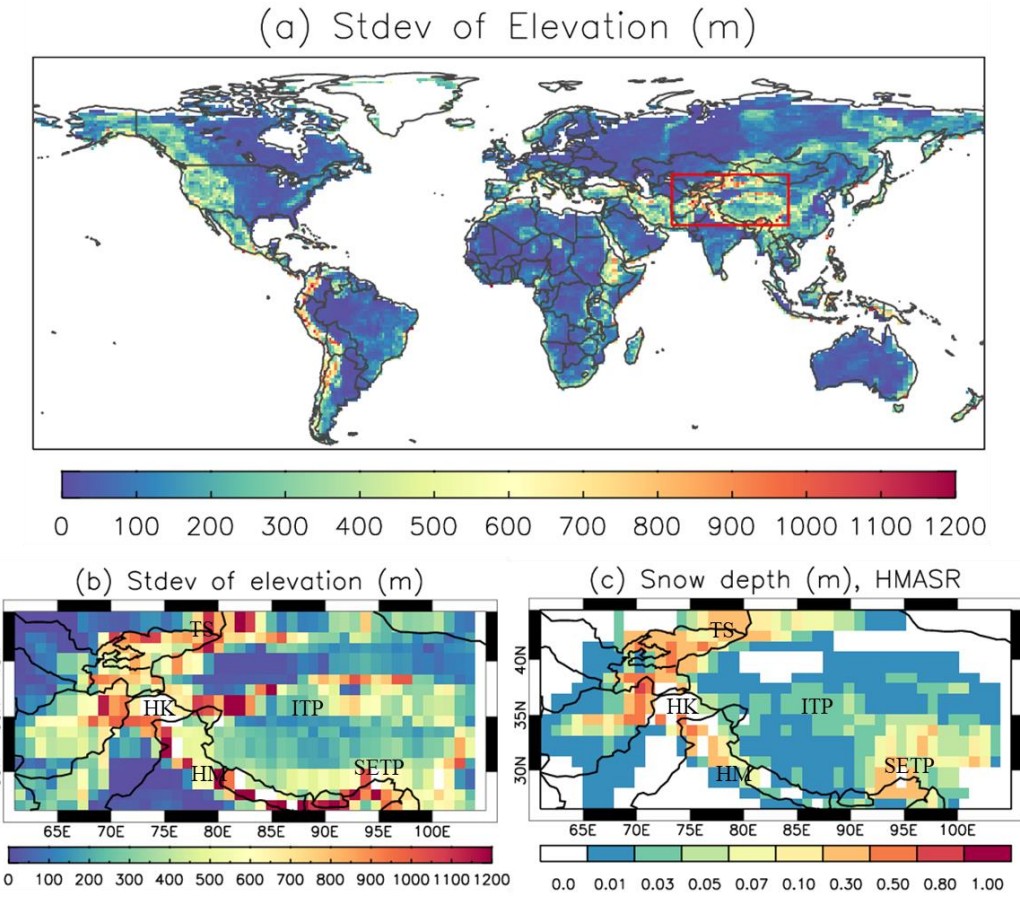

**Figure1.** (a) The standard deviation of elevation over the whole model domain; (b) the standard deviation of elevation in the HMA region (red rectangle box in (a)); (c) HMA mean snow depth during the main snow season (Sep – May) over the 2005-2014 period. Labels in (b) and (c) represent: Tibetan Plateau (TP), interior TP (ITP), southeastern TP (SETP), Tian Shan (TS), Hindu Kush–Karakoram (HK), and western Himalayas (HM).

The HMA region is one of the most complex topographic areas on Earth, with very high sub-grid
scale variability (Fig. 1b). It surrounds the Tibetan Plateau (TP), with an average elevation of
4000 m (Du and Qingsong, 2000). Considering the large SCF biases found in CanESM5 and
other CMIP models in this region (e.g. Lalande et al., 2021), we will present results for HMA
separately. Different regions of HMA exhibit different spatiotemporal patterns in snowfall and
SWE due to its unique topography (Yao et al., 2012; Bolch et al., 2019). According to the High
Mountainous Asia Snow Reanalysis (HMASR) dataset (see Section 3.2), during Sep. to May
over 2005 to 2014 period, SND is only a few centimeters over most of the interior TP, with
relatively deeper snow in southeastern TP (Fig. 1c). Deeper snow (SND > 0.2 m) is concentrated
at the high elevations of the mountains where $\sigma_{topo}$ is usually greater than 500 m, such as Tian
Shan, Hindu Kush–Karakoram, and western Himalayas (Fig. 1c).
Gridded data are regridded using the first order conservative remapping method to the 1º
latitude-longitude grid. In addition to the SCF and SWE data detailed below, the monthly air
temperature and precipitation from CRU TS4 (Harris et al., 2020) are used as references to
compare with the three meteorological forcing datasets. Evaluation metrics for SCF, SWE and
meteorological forcing include the mean bias, unbiased root mean squared error (uRMSE) and
Pearson correlation. The uRMSE is defined as the square root of the mean square error minus the
squared bias: $uRMSE = sqrt (RMSE^2 - Bias^2)$. Evaluation of other land surface variables is
according to AMBER and detailed in Section 3.4.

## 3.2 SCF observations

The monthly SCF was obtained from the Moderate Resolution Imaging Spectroradiometer
(MODIS) /Terra snow cover monthly L3 0.05º Climate Modeling Grid product (MOD10CM,
version 61). This dataset provides monthly mean SCF based on the clearest views of the surface
from 28 – 31 days of MOD10C1 daily observations and are available from the National Snow
and Ice Data Center (Hall and Riggs, 2021). The MODIS snow detection algorithm, which is
based on the Normalized Difference Snow Index (NDSI), applies processing steps to alleviate
snow detection commission errors and to flag uncertain snow detection (Hall et al., 2002). Due to
spectral similarities between cloud and snow, cloud/snow confusion situations remain in MODIS
version 6.1 snow products despite continued efforts in improving cloud masking and snow
mapping algorithms (Riggs et al., 2019). Regardless of these inherent challenges, the NDSI-
based snow detection technique has proven to be a robust indicator of snow presence under
diverse situations, as demonstrated by numerous studies reporting accuracy statistics in the range
of 88–93% (Riggs et al., 2019).
To mitigate the uncertainties in the MODIS product due to frequent cloud cover and/or complex
terrains, SCF from the Interactive Multisensor Snow and Ice Mapping System (IMS) produced
by the U.S. National Ice Center (2008) was also used as a reference in our analysis. The IMS
snow cover analysis system consists of an interactive workstation for snow cover mapping by a
snow analyst (Ramsay,1998; Helfrich et al., 2007). It relies mainly on visible satellite imagery
(including MODIS data) but is augmented by station observations and passive microwave data.

The IMS dataset consists of binary snow/no snow information on a 4 km resolution polar stereographic projection grid. Though the binary format of this dataset is not ideal for SCF estimation, especially in areas around the snow line, SCF estimates from IMS are included because the resolution of our model is coarse (1º) and IMS data has been used to evaluate modelled SCF in previous studies (e.g. Wang et al., 2014; Orsolini et al., 2019). Daily IMS data were converted to monthly snow cover duration fraction (SCF = total number of days with snow cover in a month divided by the number of days in the month) following the method in Brown et al. (2010).

Previous studies suggested that there were large uncertainties in the SCF data from MODIS and IMS datasets in the HMA region (Hao et al., 2019; Orsolini et al., 2019). Thus, the daily SCF from the HMASR dataset (Liu et al., 2021a) is used as an additional reference for the HMA region in this study. HMASR is based on a Bayesian snow reanalysis framework with model-based snow estimates refined through the assimilation of high resolution SCF data from MODIS (500 m) and Landsat (30 m) sensors (Liu et al., 2021b). The framework also accounts for a priori uncertainties in meteorological forcings and utilizes an ensemble approach (Margulis et al, 2019). The dataset provides daily data of posterior snow estimates at ~500 m spatial resolution over the HMA region. Ensemble mean values of SCF and SND are used in this study. The method used for HMASR is best suited for seasonal snow characterization (Liu et al. (2021a), thus grid cells with semi-permanent snow and ice greater than 30% are masked out in our analysis. The monthly SCF data from MODIS, IMS, and HMASR over the 2005-2014 period are used to evaluate modelled SCF.

## 3.3 SWE measurements

As shown in Eq. (1) and Eq. (2) simulated SCF is calculated from SWE directly in the SL12 parameterization, and from SND in the CTL parameterization (Section 2.2.1). Therefore, to better understand the sources of bias in simulated SCF, we also evaluate simulated SWE using snow course and airborne gamma SWE observations from Mortimer and Vionnet (2024) covering 1980 – 2014 (Fig. 2). Both types of in situ SWE information have previously been used to evaluate gridded products (e.g. Cho et al. 2019; Mortimer et al. 2020; Mudryk et al. 2025) and details of these data are described elsewhere (Mortimer et al. 2024, Mortimer and Vionnet 2025). Briefly, snow courses generally consist of multiple snow depth and density measurements collected along a predefined transect several hundred meters to several kilometers in length averaged together to obtain a single SWE value for each transect on a given date (WMO, 2018). Airborne gamma SWE estimates are calculated by differencing snow-free and snow-covered measurements of gamma radiation collected along a 15-20 km long flight line with a 300 m wide footprint after accounting for background soil moisture (Carroll, 2001). Spatial distribution and measurement frequency of the observations varies by measurement method and jurisdiction (e.g. Fig. 2 in Mortimer and Vionnet, 2025). These measurements are better able to capture the larger-scale average compared to single point observations and have been shown capable of discerning subtle differences in SWE products (Mortimer et al. 2022) and of ranking such products based on their relative performance (Mudryk et al. 2025).

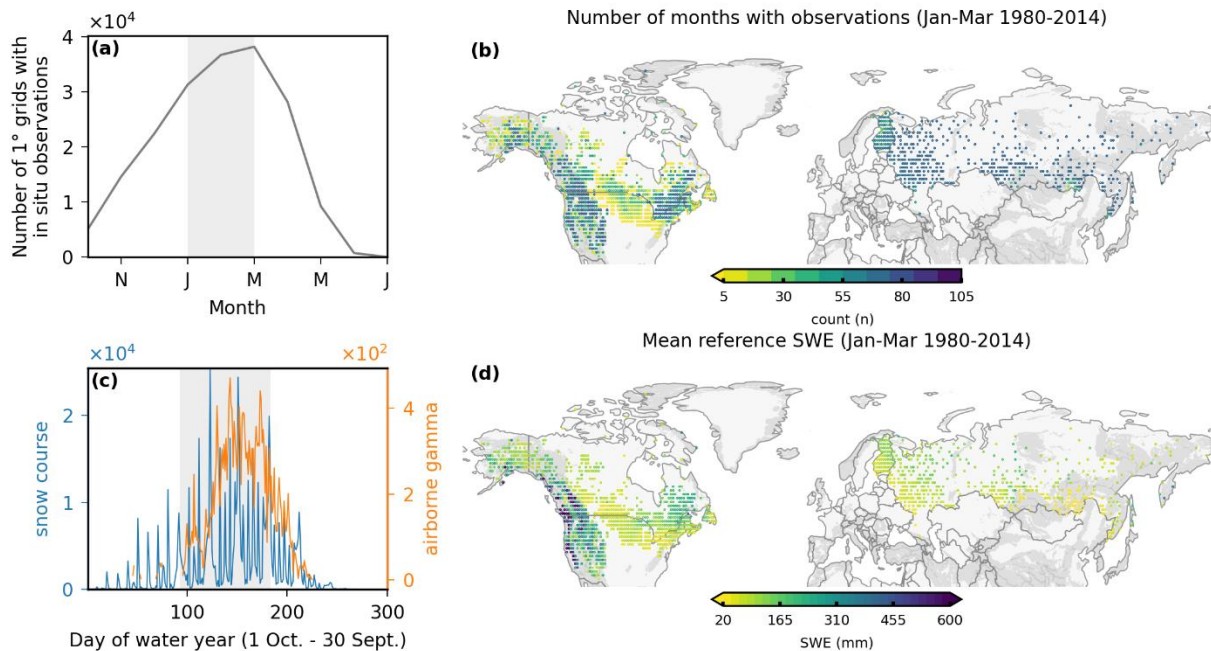

**Figure 2**. Distribution of in situ reference data. (a) Number of monthly 1°x1° grid cells with reference data during 1980-2014 (each monthly 1° grid with reference data is a data point), (b) Number of months during Nov-May 1980-2014 with reference observations by 1° grid. (c) Temporal distribution of raw in situ SWE observations. (d) Mean February-March reference SWE for grid cells with at least 5 months of data. Vertical lines in (a) and (c) indicate Nov-May period used in the analysis.

The reference SWE observations do not account for snow-free periods because they are only conducted when there is snow. During the accumulation and ablation seasons, the monthly mean of available reference SWE will therefore often overestimate the true monthly mean value. For this reason, we restrict the comparisons of product SWE with reference SWE to January-March. Additionally, the infrequent sampling of the reference data (Fig. 2c; see also Table 4 in Mortimer and Vionnet, 2025) means that, even when there is continuous snow cover, the monthly value calculated from the available dates with observations may not be representative of the true monthly mean. Investigation of the timing of the in-situ measurements within a month showed that, for the full domain, the timing of the observations is fairly well distributed across a month. However, this varies regionally and by network with some networks (e.g. Canada) biased towards the first half of the month and others (e.g. Russia) slightly biased towards the latter two thirds of the month (Fig. A1). We are unable to account for these biases in our analysis. The statistics calculated from comparisons with in-situ data are not intended to be used as absolute performance measures. Rather, we are interested in assessing how the relative performance of CLASSIC SWE varies under the three choices of forcings; as Mortimer et al. (2024) demonstrates, the reference data is well able to discern relative performance of SWE products.

To evaluate monthly model output with reference observations from a specific date, we first match reference SWE observations to the model grid cell estimate from the corresponding month. Next, from these matched data, we calculate the mean reference SWE for each month. If there were multiple reference SWE observations within the same product grid cell on the same

date, they were averaged prior to calculating the monthly mean. Metrics were calculated
separately for mountainous and flat regions (as defined in Section 3.1) for each month (all years
pooled together), for each year (all months pooled together), for the full time period (all years
and months pooled together), and for each product grid cell (all years pooled together). The
analysis is limited to non-zero values with SWE $\leq$ 3000 mm in both the observation and model
outputs, and to the months January to March.

**Table 1.** Overview of the reference datasets used in AMBER, including the following variables: net
surface radiation (RNS), net surface shortwave radiation (RSS), net surface longwave radiation (RLS),

| Dataset | Variables | Method | Period | References |
|---------|-----------|--------|--------|------------|
| CERES | ALBS, RSS, RLS, RNS | Radiative transfer model | 2000-2013 | Kato et al. (2013) |
| CLASSr | RNS, HFLS, HFSS, MRRO | Blended product | 2003-2009 | Hobeichi et al. (2020) |
| FLUXCOM | RNS, HFLS, HFSS | Machine learning ensemble | 1980-2013 | Jung et al. (2019) |
| FLUXNET | RNS, HFLS, HFSS | eddy covariance (204) | 1997–2014 | Pastorello et al. (2017) |
| GEWEXSRB | ALBS, RSS, RLS, RNS | radiative transfer model | 1984-2007 | Stackhouse et al. (2011) |
| GRDC | MRRO | gauge records (50) | 1980–2010 | Dai and Trenberth (2002) |
| GRUN | MRRO | Reconstruction via machine learning | 1902-2014 | Ghiggi et al. (2019) |
| MODIS | ALBS | Bidirectional Reflectance Distribution function | 2000-2014 | Schaaf and Wang (2015) |

surface albedo (ALBS), latent heat flux (HFLS), sensible heat flux (HFSS), and runoff (MRRO).

**3.4 Reference datasets used to evaluate land surface variables in AMBER**

Spatial and temporal variations of snow cover account for most of the variations in surface
albedo due to its much higher reflectivity relative to underlying land surfaces. Changes in SCF
thereby lead to changes in surface albedo, which in turn lead to changes in surface radiation and
energy fluxes. To illustrate the impact of the SL12 parameterization on the simulated radiation,
energy fluxes, and the water cycle in CLASSIC, we computed skill scores using the AMBER
package (Seiler et al., 2021) for the global 1º simulations. AMBER assesses model performance
against a collection of observation-based reference datasets based on five scores: bias ($S_{bias}$),
root-mean-square-error ($S_{rmse}$), phase ($S_{phase}$), interannual variability ($S_{iav}$), and spatial
distribution ($S_{dist}$). An overall score ($S_{overall}$) is calculated by averaging the five scores. The scores
are dimensionless and on a scale from 0 to 1 where a higher value implies better model
performance. Lower values are, however, not necessarily a product of poor model performance
as the scores are also affected by uncertainties in the forcing and the reference data. Further
details regarding the AMBER package as well as the skill score equations are presented in Seiler
et al. (2021) and Seiler (2019). Table 1 shows the 21 reference datasets used in AMBER in this
study, which contain information about seven variables relevant to the radiation, energy, and
water cycle including net surface radiation (RNS), net surface shortwave radiation (RSS), net

surface longwave radiation (RLS), surface albedo (ALBS), latent heat flux (HFLS), sensible heat
flux (HFSS), and runoff (MRRO). These datasets include monthly mean values, and more details
can be found in Seiler et al. (2021).

## 4. Results


### 4.1 Comparison of air temperature and precipitation in meteorological datasets

To better understand biases in the simulated snow cover, we first compare air temperature and
precipitation from the three meteorological datasets with respect to CRU over the NH and HMA
during the 1980-2014 period (Fig. 3 and Fig. A2). Because the CRUJRA data is already bias-
corrected to CRU temperature and precipitation, it exhibits very small biases in both variables in
all regions relative to this product. By comparison, both ERA5 and GSWP3W5 are colder during
most of the months in the NH (Fig. 3a). The magnitude of the cold bias is larger in the
mountainous than in the flat regions and larger in GSWP3W5 than in ERA5. Likewise, both
ERA5 and GSWP3W5 have more precipitation than CRUJRA over the whole snow season. This
difference is especially pronounced in ERA5 in the mountainous regions during the fall and
spring months (Fig. 3b and Fig. A2b). In HMA, the bias patterns in temperature and precipitation
are similar to those for mountainous regions across the full NH. However, the magnitude of the
cold bias (with respect to CRU) is larger in ERA5 than in GSWP3W5 (Fig. 3c). Because
different reference datasets were used to bias-adjust precipitation in CRUJRA (CRU) and
GSWP3W5 (GPCP), we also compare the monthly precipitation from CRU and GPCP in the
above regions and over the same period. This analysis (not shown) indicates that the differences
between CRU and GPCP are within 2 % and 3 % for NH flat and mountainous regions
respectively, but up to 21 % in HMA.











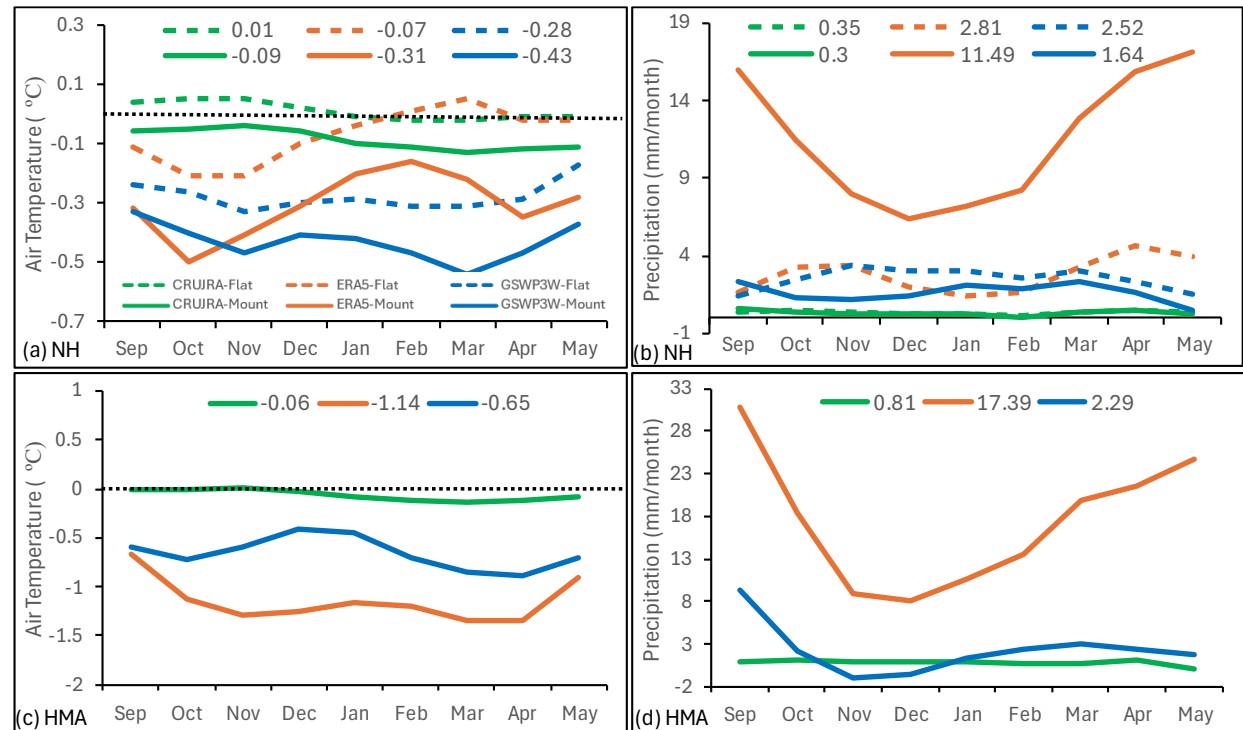

**Figure 3**. Bias in monthly mean air temperature (a and c) and precipitation (b and d) in the NH mountainous (solid line) and flat (dashed line) regions (a and b) and the HMA mountainous region (c and d) over the 1980-2014 period. Values shown at the top of each plot are the mean temperature or precipitation during Sep-May period for each dataset.

## 4.2 Evaluation of SWE

Large differences in SWE from the model runs using the CTL and SL12 parameterizations are limited to small areas near grid cells with land ice because the runs are forced by the same three sets of meteorological datasets, and there is limited feedback in offline runs. Thus, we only present results for SWE from the model runs using the SL12 parameterization. The SWE reference measurements (Section 3.2) indicate that for all choices of meteorological forcing, CLASSIC underestimates SWE in mountainous regions (Fig. 4a) and overestimates SWE in flat regions (Fig. 4b) over the 1980-2014 period. For both types of regions, the magnitudes of the biases increase as the snow season progresses. In the mountainous regions, the biases are similar forGSWP3W5-SL12 (-129.4) and CRUJRA-SL12 (-136.6) and the lowest for ERA5-SL12 (-90.8). In flat regions, GSWP3W5-SL12 (50.3) has more than twice the SWE bias seen in either CRUJRA-SL12 (15.0) or ERA5-SL12 (17.5), which is mainly due to SWE overestimation in eastern NA and northern Europe (Fig. A3). Overall, ERA5-SL12 outperforms the other two model runs with lower bias in mountainous regions and it shows similar performance as CRUJRA-SL12 in flat regions.

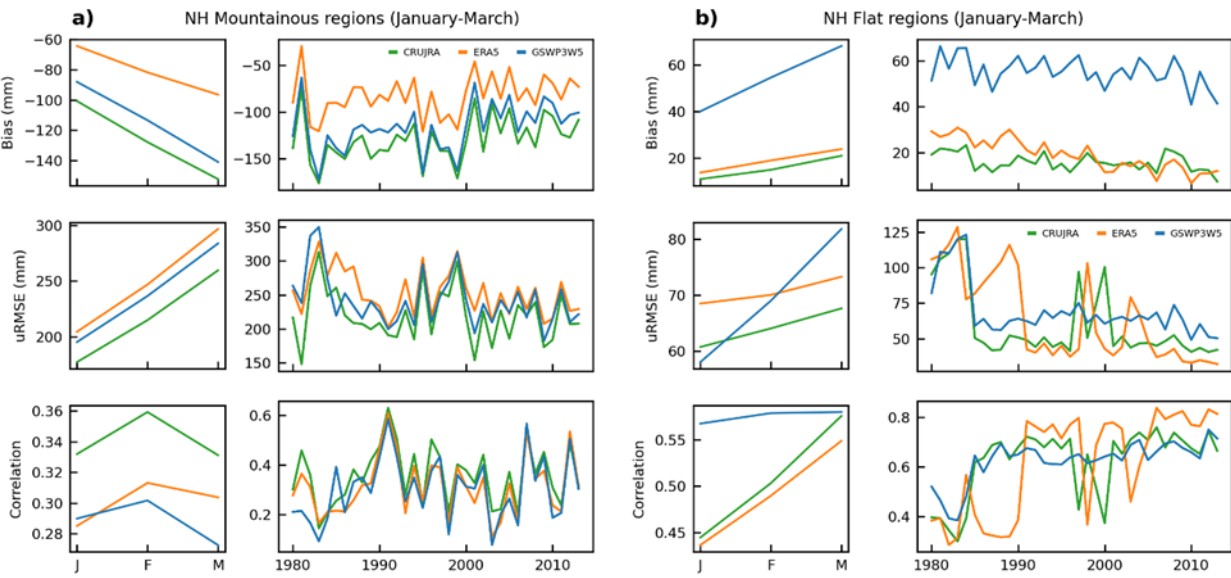


**Figure 4**. Annual and interannual evolution of bias, uRMSE, and correlation for modelled SWE in model
runs using the SL12 parameterization forced by CRUJRA, ERA5, and GSWP3-W5E5 in (a) NH
mountainous regions and (b) NH flat regions over the 1980-2014 period.


**Table 2**. The seasonal mean SCF bias, uRMSE, and Pearson correlation coefficient (r) for the Control and
SL12 simulations over the (a) NH mountainous regions ($\sigma_{topo}$ >200 m), (b) NH flat regions ($\sigma_{topo}$ <= 200
m). The observed SCF from MODIS is used as the reference.

| (a) NH Mountain | SON | | | DJF | | | MAM | | | Annual | | |
|---|---|---|---|---|---|---|---|---|---|---|---|---|
| Met-Scheme | Bias | uRMSE | r | Bias | uRMSE | r | Bias | uRMSE | r | Bias | uRMSE | r |
| CRUJRA - CTL | -0.01 | 0.08 | 0.55 | 0.06 | 0.08 | 0.24 | 0.10 | 0.12 | 0.45 | 0.04 | 0.13 | 0.59 |
| CRUJRA - SL12 | -0.04 | 0.07 | 0.56 | 0.01 | 0.07 | 0.31 | 0.01 | 0.08 | 0.55 | -0.01 | 0.09 | 0.62 |
| ERA5 - CTL | 0.01 | 0.06 | 0.59 | 0.07 | 0.07 | 0.28 | 0.09 | 0.11 | 0.48 | 0.05 | 0.12 | 0.62 |
| ERA5 - SL12 | -0.02 | 0.06 | 0.60 | 0.02 | 0.06 | 0.38 | 0.01 | 0.06 | 0.60 | 0.00 | 0.08 | 0.66 |
| GSWP3W5 - CTL | -0.02 | 0.07 | 0.57 | 0.03 | 0.08 | 0.29 | 0.05 | 0.11 | 0.48 | 0.03 | 0.13 | 0.59 |
| GSWP3W5-SL12 | -0.04 | 0.07 | 0.58 | -0.02 | 0.07 | 0.35 | -0.03 | 0.07 | 0.56 | -0.02 | 0.09 | 0.64 |
| (b) NH Flat | SON | | | DJF | | | MAM | | | Annual | | |
| Met-Scheme | Bias | uRMSE | r | Bias | uRMSE | r | Bias | uRMSE | r | Bias | uRMSE | r |
| CRUJRA - CTL | -0.02 | 0.07 | 0.57 | 0.02 | 0.05 | 0.20 | 0.09 | 0.12 | 0.44 | 0.03 | 0.11 | 0.59 |
| CRUJRA - SL12 | -0.04 | 0.08 | 0.57 | 0.01 | 0.06 | 0.24 | 0.08 | 0.11 | 0.47 | 0.02 | 0.11 | 0.59 |
| ERA5 - CTL | -0.02 | 0.07 | 0.58 | 0.01 | 0.05 | 0.24 | 0.07 | 0.09 | 0.50 | 0.02 | 0.10 | 0.61 |
| ERA5 - SL12 | -0.04 | 0.08 | 0.58 | 0.00 | 0.05 | 0.27 | 0.06 | 0.09 | 0.52 | 0.01 | 0.10 | 0.61 |
| GSWP3W5 - CTL | 0.00 | 0.08 | 0.57 | 0.02 | 0.06 | 0.19 | 0.10 | 0.13 | 0.41 | 0.04 | 0.13 | 0.58 |
| GSWP3W5-SL12 | -0.02 | 0.08 | 0.57 | 0.01 | 0.06 | 0.23 | 0.09 | 0.13 | 0.45 | 0.03 | 0.12 | 0.58 |



## 4.3 Evaluation of SCF

### 4.3.1 NH regions

Figure 5 shows the monthly mean SCF (area weighted) from all six simulations along with the
MODIS and IMS observations over different regions. SCF from MODIS and IMS generally
agree well with each other in all regions except for HMA, where IMS shows ~3 % - 6 % more
SCF than MODIS in the winter months (Fig. 5g). In the NH, NA, and EA mountainous regions
(Fig. 5a-5c and Table 2), both the CTL and the SL12 parameterizations underestimate SCF in the
fall (SON), with the SL12 parameterization performing slightly worse than the CTL
parameterization. However, during winter (DJF) and spring (MAM), the SL12 parameterization
greatly outperforms the CTL parameterization for all three meteorological datasets. For example,
in the NH mountains during the spring, the mean biases are 0.1, 0.09, and 0.05 with the CTL
parameterization for model runs forced by CRUJRA, ERA5, and GSWP3W5 respectively; they
are 0.01, 0.01, and -0.03 with the SL12 parameterization (Table 2a). The uRMSEs are 0.12, 0.11,
and 0.11 with the CTL parameterization, and 0.08, 0.06, and 0.07 with the SL12

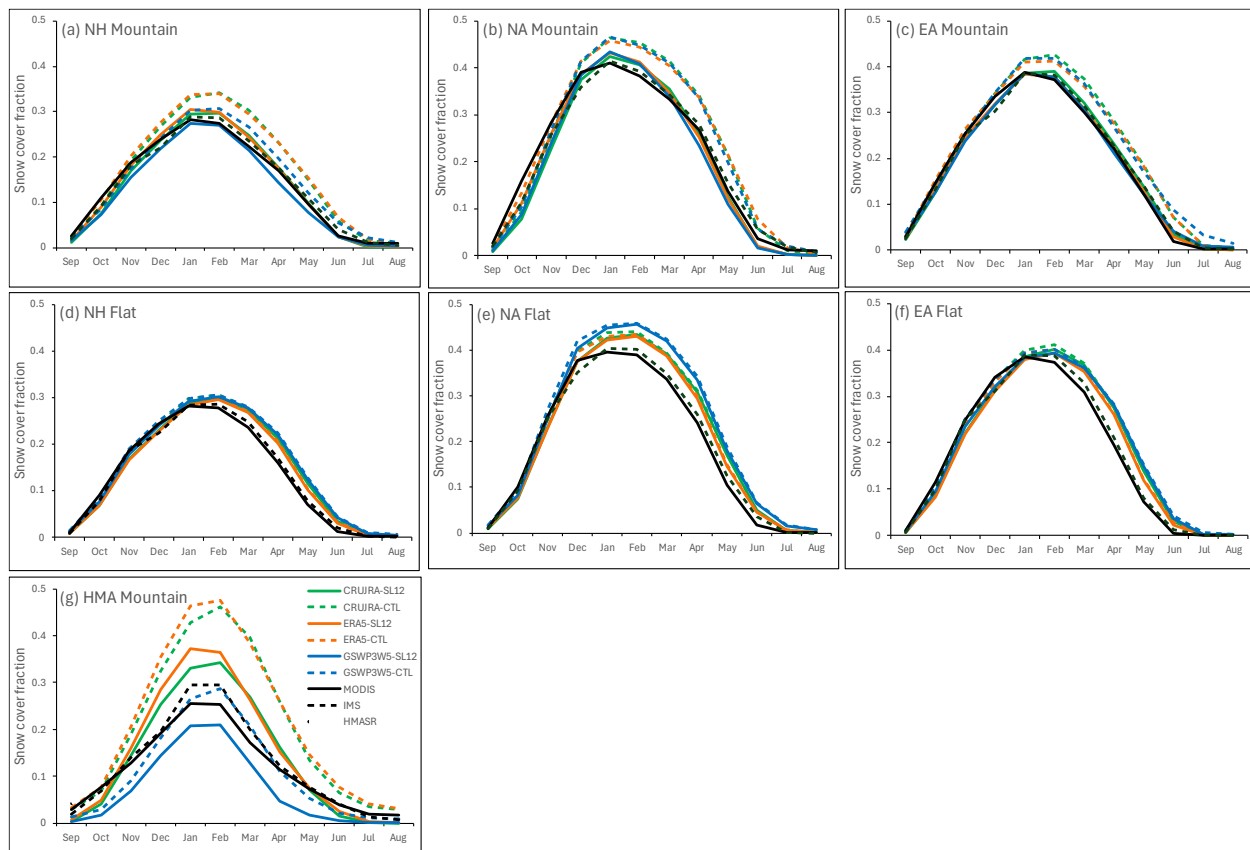

**Figure 5.** The monthly mean SCF from model runs using the Control (dashed line) and SL12 (solid line)
parameterizations for NH, NA, and EA mountainous ($\sigma$topo $\geqslant$ 200 m, a-c) and flat ($\sigma$topo < 200 m, d-e)
regions, and (g) shows the monthly mean SCF for the HMA mountainous region. The black lines
represent observed SCF from MODIS (solid), IMS (dashed), and HMASR (dotted).

parameterization; and the correlation coefficients are 0.45, 0.48, and 0.48 with the CTL
parameterization, and 0.55, 0.60, 0.56 with the SL12 parameterization (Table 2a). On average for
all three meteorological forcing choices, the annual mean bias, uRMSE, and correlation improve
by 75 %, 32 %, and 7 % when evaluated with MODIS SCF observations over the NH
mountainous regions.

In flat regions (all domains), as expected, the performance is similar regardless of the
parameterization with a 2-4 % SCF underestimation in the fall, but a 1-2 % and 6-10 % SCF
overestimation during the winter and spring seasons, respectively (Fig. 5d-5f and Table 2b).
Among the six simulations, ERA5-SL12 has the lowest annual bias (0.0) and uRMSE (0.08), and
the highest correlation (0.66) in the NH mountainous regions, as well as in the flat regions
(bias=0.01, uRMSE=0.1, and r=0.61) (Table 2).

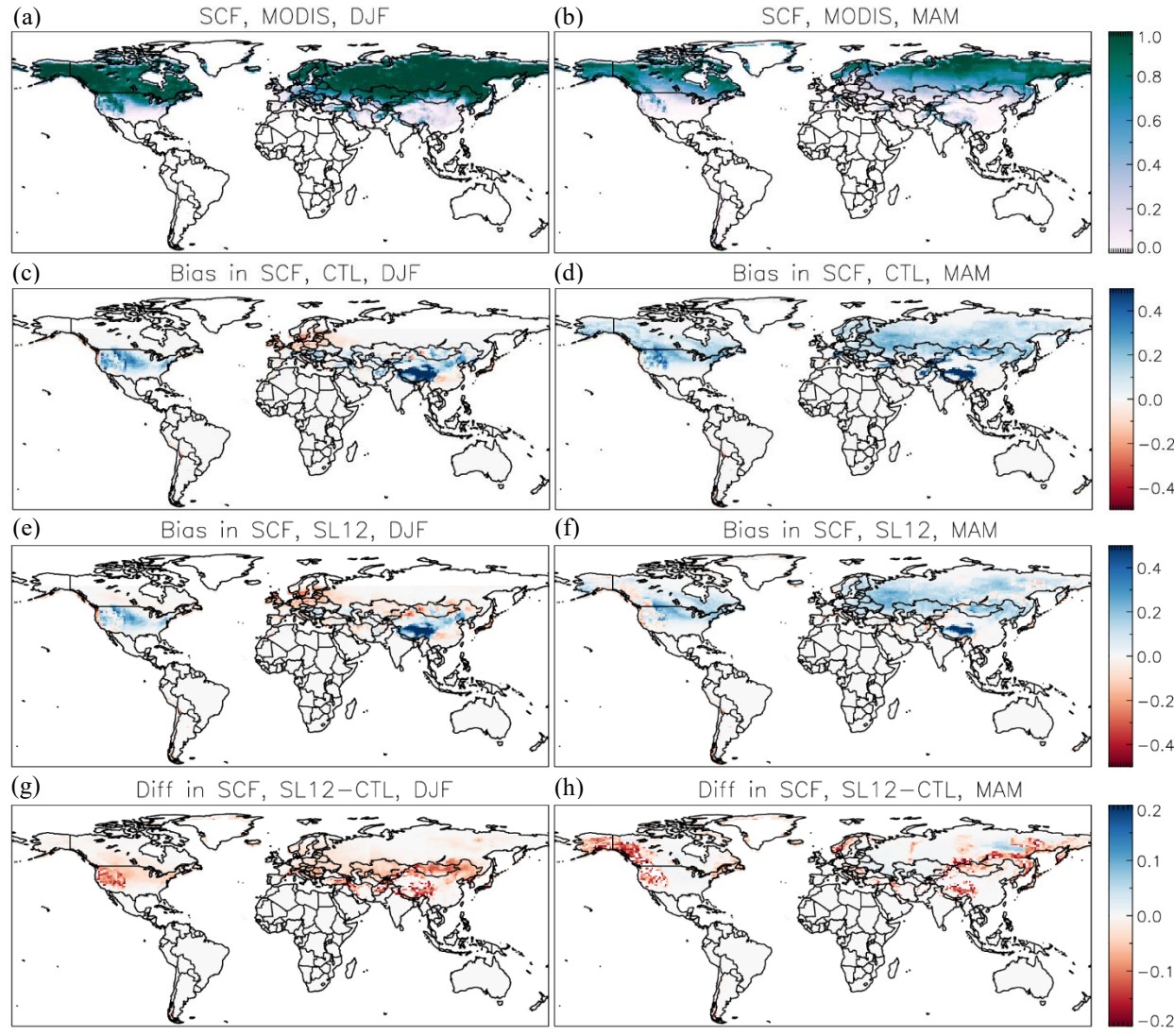

**Figure 6.** Snow cover fraction from MODIS (a and b), SCF bias in model runs using the Control (c and
d) and SL12 (e and f) parameterizations, and difference in SCF between SL12 and Control (g and h)
during the winter (left) and spring (right) season.

On the global scale, the spatial patterns of SCF bias are similar for all three meteorological forcing choices. Figure 6 shows an example of the spatial pattern in SCF bias from the model runs forced by ERA5 during the winter and spring seasons. Compared to observed SCF from MODIS, model runs tend to overestimate SCF in areas where SCF is less than 100 % in both the winter and spring seasons. In the winter, both parameterizations have areas with SCF underestimation, such as in the western NA mountainous areas, northern Europe, and some areas of Asia (Fig. 6c and 6e). In the spring, the CTL parameterization overestimates SCF in most NH regions except for some limited areas in western NA (Fig. 6d). The SCF overestimation is reduced in the run using the SL12 parameterization, and replaced with some SCF underestimation, such as in the western NA mountains (Fig. 6f). Overall, the SL12 parameterization produces less SCF and thus reduces the SCF overestimation found in the model runs using the CTL parameterization over all major mountain ranges across the globe (Fig. 6g and 6h).

### 4.3.2 HMA region

In HMA, large uncertainties have been found in SCF from the MODIS and IMS datasets (Hao et al., 2019; Orsolini et al., 2019), thus SCF from the HMASR dataset is also included as a reference along with MODIS and IMS. Results are only shown for the mountainous region (Fig. 5g) because there are limited flat areas with snow cover (Fig. 1b and 1c). HMASR has a single peak in Feb., while MODIS, IMS, and all the model runs have peaks in both Jan. and Feb. Over this region, simulations using either parameterization exhibit large SCF overestimations during the winter and spring compared to all three reference datasets especially when forced by CRUJRA or ERA5 (Fig. 5g). Compared to SCF from HMASR, the mean biases are 0.30 and 0.35 in CRUJRA-CTL and ERA5-CTL respectively during the winter (Table 3). In contrast, the model runs driven by GSWP3W5 have much lower SCF and smaller biases (Fig. 5g and Table 3). Overall, the SL12 parameterization exhibits improved performance compared to the CTL parameterization. On average from all three meteorological forcing choices, the annual mean bias, uRMSE, and correlation improve by 48 %, 30 %, and 5 % when evaluated with HMASR SCF data over the HMA mountainous areas.

**Table 3.** Same as Table 2 but for the HMA region. SCF from the HMASR dataset is used as the reference.

| HMA Mountain | SON | | | DJF | | | MAM | | | Annual | | |
|---|---|---|---|---|---|---|---|---|---|---|---|---|
| Met-Scheme | Bias | uRMSE | r | Bias | uRMSE | r | Bias | uRMSE | r | Bias | uRMSE | r |
| CRUJRA - CTL | 0.02 | 0.12 | 0.35 | 0.30 | 0.15 | 0.31 | 0.20 | 0.17 | 0.39 | 0.13 | 0.21 | 0.42 |
| CRUJRA - SL12 | -0.03 | 0.09 | 0.37 | 0.16 | 0.11 | 0.35 | 0.06 | 0.10 | 0.45 | 0.04 | 0.15 | 0.44 |
| ERA5 - CTL | 0.05 | 0.12 | 0.43 | 0.35 | 0.14 | 0.36 | 0.23 | 0.15 | 0.40 | 0.16 | 0.22 | 0.45 |
| ERA5 - SL12 | -0.01 | 0.09 | 0.45 | 0.22 | 0.11 | 0.42 | 0.08 | 0.09 | 0.51 | 0.06 | 0.15 | 0.48 |
| GSWP3W - CTL | -0.06 | 0.08 | 0.40 | 0.08 | 0.14 | 0.39 | 0.01 | 0.12 | 0.44 | 0.00 | 0.14 | 0.45 |
| GSWP3W - SL12 | -0.08 | 0.07 | 0.41 | 0.00 | 0.11 | 0.39 | -0.08 | 0.08 | 0.48 | -0.05 | 0.10 | 0.46 |

In HMA, areas with high SCF (> 40 %) are mainly found along the western mountain ranges (e.g. Tian Shan, Hindu Kush–Karakoram, and western Himalayas) and southeast portion of the TP (Fig.7a-7c). SCF is less than 20 % in most of the interior TP, even during the winter (Fig. 7a). On average, maximum SCF occurs in winter in western HMA (i.e. Tian Shan and Hindu Kush–Karakoram), but it occurs in spring in interior TP and southeast TP. Among the model runs using the CTL parameterization, there are significant SCF overestimations in most of HMA when forced by CRUJRA or ERA5 (Fig. 7d, 7e). The run forced by GSWP3W5 still overestimates SCF in the mountainous areas of western HMA but underestimates SCF in the interior TP and southeast of TP (Fig. 7f). Given that all three simulations use the same CTL parameterization (Fig. 7d–f), the substantial differences in simulated SCF, particularly in the GSWP3W5-forced run, suggest that the primary source of the discrepancy lies in the forcing data. This will be discussed further in Section 5. In the model runs using the SL12 parameterization (Fig. 7g-7i), the SCF overestimations are much reduced in the western mountainous areas while across the rest of the plateau the SCF underestimations are very similar for both parameterizations.

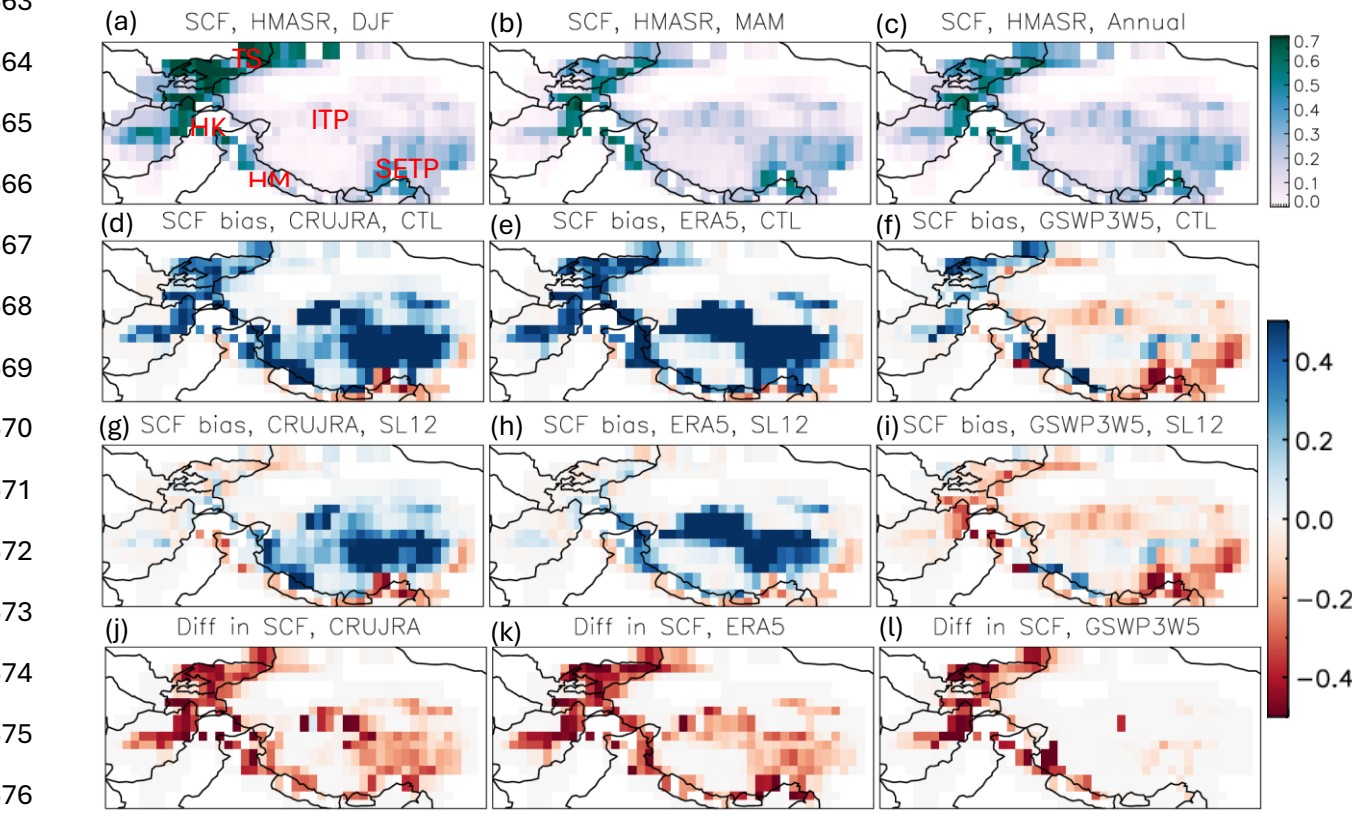

**Figure 7.** The top panel shows SCF from HMASR for (a) winter, (b) spring, and (c) annual mean. The second and third panel shows SCF biases from model runs using the CTL (d-f) and SL12 (g-i) parameterizations forced by the three meteorological datasets respectively during spring. The bottom panel (j-k) shows the difference in SCF between the model runs using the SL12 and CTL parameterizations.

**4.4 Evaluation of other land surface variables**

Evaluation of other land surface variables (besides SCF and SWE) via AMBER scores (Section
3.3) is shown in Fig. 8 for each of the six CLASSIC simulations. Model runs using the SL12
parameterization have the best score for 101 of 119 diagnostic tests while they have the worst
score for only 16 of 119 diagnostic tests (Fig. 8c and 8d). CRUJRA-SL12 (ID=2) and ERA5-
SL12 (ID=4) have the highest overall scores for five radiation reference datasets (one RNS, two
RSS, two RLS), and three surface albedo (ALBS) reference datasets with improvements ranging
from 0.01 to 0.06 when compared to the runs with the lowest scores (Fig. 8b and 8c). The
relatively large score differences in the interannual variability score ($S_{iav}$) for net surface
radiation (RNS) suggest improved interannual variability of net surface radiation when using the
SL12 parameterization (Fig. 8b). For surface albedo, relatively large differences are observed in
the spatial distribution score ($S_{dist}$), suggesting better characterization of the spatial patterns when
using the SL12 parameterization (Fig. 9). Figure 9 shows that surface albedo is generally
overestimated by the control scheme (Fig. 9a), with this overestimation notably reduced in the
mountainous regions when the SL12 scheme is applied (Fig. 9b), consistent with the
improvements seen in SCF.  Previous studies have indicated that the MODIS surface albedo
product may exhibit biases due to the absence of shading corrections in mountainous areas and
underestimation of snow cover in dense forest regions (Hall et al., 2002; Bair et al., 2022). These
limitations may have contributed, at least in part, to the albedo overestimation shown in Figure
9a.

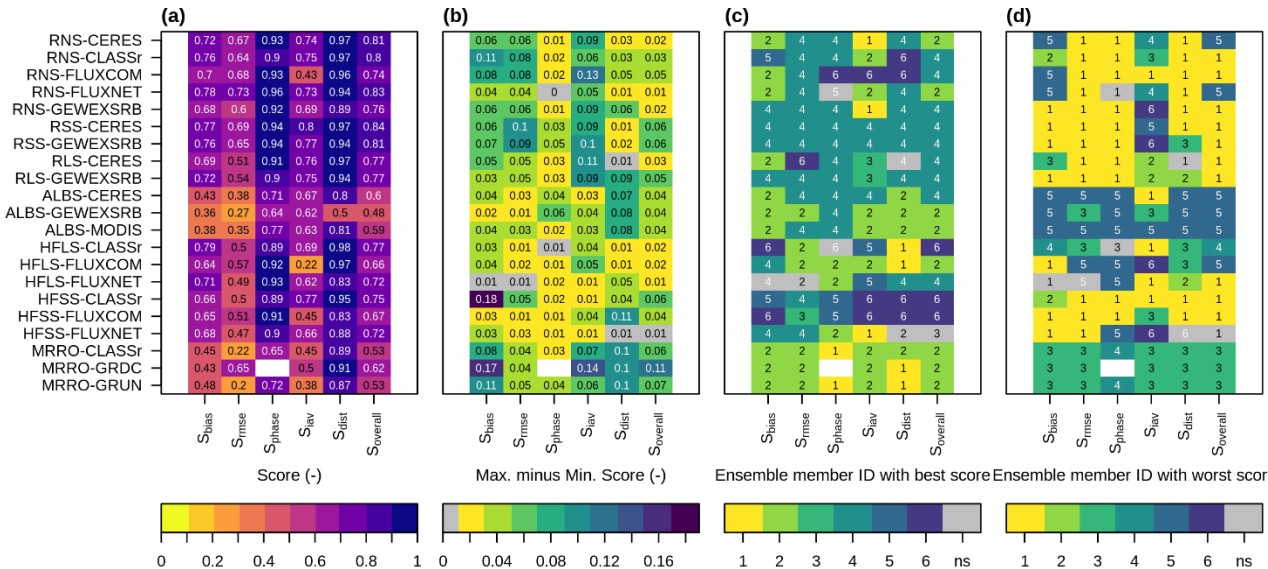


**Figure 8.** AMBER results for other land surface variables from the six model runs, (a) mean ensemble
score, (b) maximum score difference among ensemble members, (c) ensemble member with the highest
score, and (d) ensemble member with the lowest score. Comparisons are grayed out in panels (b–d) when
the difference between the maximum and minimum scores is less than 0.01. Ensemble member IDs
represent the following model runs: 1: CRUJRA-CTL, 2: CRUJRA-SL12, 3: ERA5-CTL, 4: ERA5-SL12,
5: GSWP3W5-CTL, 6: GSWP3W5-SL12.
Though GSWP3W5-SL12 (ID=6) has the lowest frequency of the model runs with the best
scores (Fig. 8c), it has the highest overall performance for some of the heat fluxes datasets - one
out of the three HFLS and two out of the three HFSS reference datasets. For surface runoff,
model runs with the best scores are all forced by CRUJRA, while model runs with the worst
scores are all forced by ERA5 (Fig. 8c and 8d).

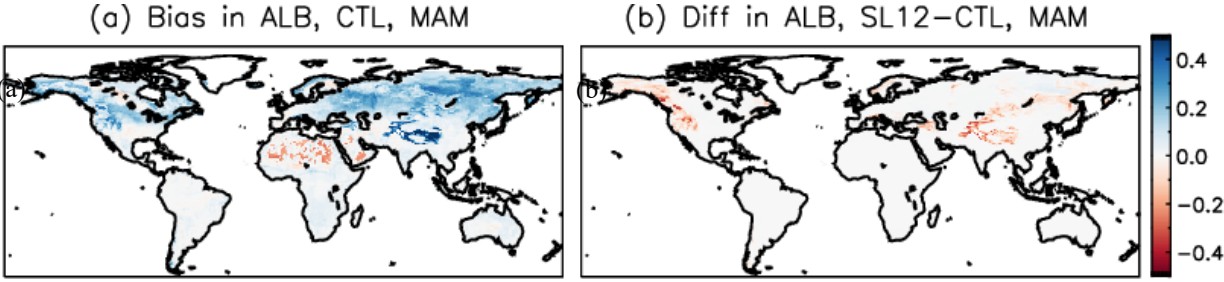


**Figure 9**. (a) Surface albedo (ALB) bias (relative to observed from MODIS) in a model run forced by
ERA5 using the Control parameterization in the spring, (b) the difference in ALB between the model runs
using the SL12 and CTL parameterizations, with red colours indicating lower albedo simulated by the
SL12 parameterization.
To isolate the impact of meteorological forcing data and SCF parameterization on these snow-
related variables, we also calculate AMBER scores for the three model runs separately for the
SL12 (Fig. 10) and the CTL (Fig. A4) parameterizations. The results show that regardless of the
parameterization, overall model runs forced by ERA5 (ID = 2) perform best for most radiation
fluxes, while model runs forced by CRUJRA (ID = 1) perform best for the rest of the variables
except for some heat fluxes where model runs forced by GSWP3W5 (ID = 3) perform best (Fig.
10c). These are generally consistent with results shown in Fig. 8 with both parameterizations
included, suggesting that the score differences among ensemble members are largely due to
differences in the meteorological forcing. However, the overall scores with the SL12
parameterization (Fig. 10a) are slightly larger for most variables than those with the CTL
parameterization (Fig. A4a). Among the three model runs using the SL12 parameterization,
ERA5-SL12 has the most (43/99) frequency in the model runs with the best scores (Fig. 10c),
followed by CRUJRA-SL12 (38), with GSWP3W5-SL12 having the least frequency (18).

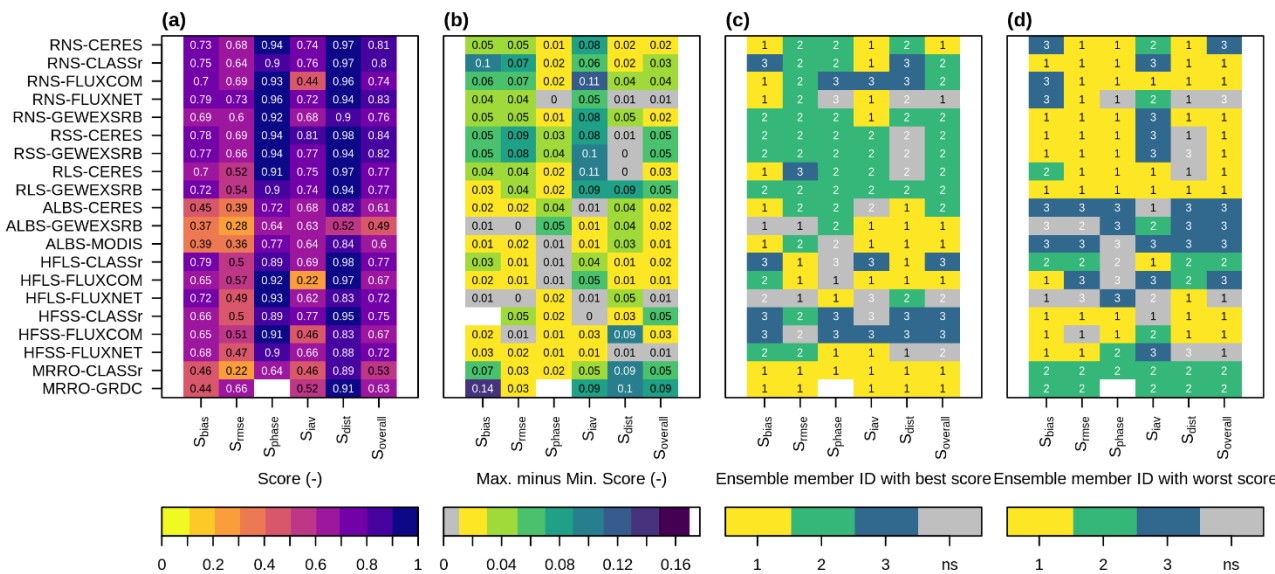

**Figure 10.** Same as in Fig. 8 except for the three model runs using the SL12 parameterization. Ensemble member IDs represent the following model runs: 1: CRUJRA-SL12, 2: ERA5-SL12, 3: GSWP3W5-SL12.

## 5. Discussion

This study evaluates the SL12 SCF parameterization against the current default (CTL) parameterization on snow simulation in CLASSIC. To account for uncertainties in the meteorological forcing data, three reanalysis-based datasets are used to drive the model. Biases in modeled SCF vary between flat and mountainous regions for both SCF parameterizations (Table 2, Fig. 5, and Fig. 6). Previous studies have highlighted the importance of accounting for sub-grid topography on SCF simulations in mountainous regions (Swenson and Lawrence, 2012; Miao et al., 2022). Are the modelled SCF biases related to topographic complexity in this study? To explore this, we generated scatter plots and examined the correlations between SCF biases and the standard deviation of sub-grid topography during the winter and spring seasons for each simulation. As expected, significant correlations were found in all simulations, indicating that SCF biases tend to increase with increasing topographic complexity. However, this relationship is notably reduced under the SL12 scheme, particularly in spring. An example of these scatter plots, based on model runs forced by CRUJRA, is presented in Figure A5. Below we discuss the possible factors contributing to biases in the simulated SWE and SCF including potential biases in the meteorological forcing datasets.

### 5.1 Impacts of meteorological forcing datasets on modelled SWE

Evaluation based on measurements from snow course and airborne gamma data indicates that the magnitude of SWE bias and uRMSE seen in CLASSIC are comparable to those from other gridded SWE products and LSMs (Brown et al., 2018; Mortimer et al., 2024; Cho et al., 2022)

intended to represent historical snow conditions. However, for all three choices of
meteorological forcing SWE is underestimated in mountainous regions (Fig. 4a) and
overestimated in flat regions (Fig. 4b) throughout the snow season (with subsequent impacts on
SCF). Since SCF is directly linked to SWE in the SL12 scheme (see Eq.1 and Eq. 2), these SWE
biases can exert a large impact on simulated SCF in the fall and spring seasons in the model
(limited impact during the peak SWE period because SCF is usually saturated). The consistent
SCF biases shown in Figure 5 are linked to these consistent SWE biases for all three forcing
choices in the model.
Naively, the bias-adjustments applied to temperature and precipitation in both the CRUJRA and
GSWP3W5 forcing data might be expected to result in more accurate simulations. Yet among the
three choices of forcing we used, the unadjusted ERA5 data yielded the lowest bias when
evaluating the simulated SWE in mountainous regions (Fig. 4, Fig. A3). In mountain regions,
this discrepancy may result because the CRU and GPCP data used to adjust the precipitation
values are biased towards locations with less precipitation (e.g. outside of regions with
orographic features; e.g. Nijssen et al., 2001; Adler et al, 2003; Shi et al., 2017). Mountain
precipitation underestimation was also linked to negative SWE biases based on precipitation
observations from the Snowpack Telemetry stations over western U.S. (Cho et al. 2022).
In NH flat regions, precipitation values from CRU and GPCP are expected to be more accurate
than in mountainous regions (Adler et al., 2003), so it is less clear why GSWP3W5 has a much
larger SWE bias despite having a precipitation bias similar to ERA5. The fact that GSWP3W5 is
colder in flat regions compared to the other two forcings could play a role (Fig. 3a). This may
reduce its ability to simulate mid-season ablation events (e.g., Brown et al., 2006; Slater et al.,
2001) and/or alter the timing and location of snowfall. The reason that GSWP3W5 is colder than
CRUJRA is also not immediately clear since both products use CRU TS4 for bias-adjusting their
temperature (see Section 2.3.2). Differences between the interpolation and bias-adjustment
methods may be responsible for the differences since they are more complex for GSWP3W5 (see
Cucchi et al., 2020 and Weedon et al., 2010) than CRUJRA (Harris, 2023). For example, a
constant lapse rate of 6.5 K km$^{-1}$ was applied to temperature correction in GSWP3W5 but not in
CRUJRA.
These results highlight that there is uncertainty in the accuracy of both temperature and
precipitation forcing even when bias-adjusted to observations. These uncertainties can propagate
to uncertainty in simulated SWE directly through precipitation amounts or in the case of
temperature through phase partitioning of rainfall versus snowfall or direct melt. Even with
perfectly constrained bias-adjustments for temperature and precipitation individually, there may
still be spread in simulated SWE stemming from uncertainties in the joint distribution of
temperature and precipitation that determines when snowfall occurs. Although measurements
from snow course and airborne gamma data used in this study can better sample the subgrid-
scale variability than a single-point measurement, we acknowledge that there are still
uncertainties in our evaluation results, e.g. in situ sites may be biased towards locations with
more snow cover.

**5.2 Factors contributing to residual bias in modelled SCF**

Although SCF overestimation in the mountainous regions is much reduced by the SL12 parameterization compared to the CTL parameterization (Fig. 5a – 5c and 5g), there are still areas with notable SCF biases. For example, much of the western NA mountainous areas have negative biases during the spring with the SL12 parameterization (Fig. 6d and 6f). Furthermore, in flat areas, all model runs overestimate SCF (Fig. 5d – 5f). These remaining SCF biases may be at least partly attributable to SWE underestimation in mountainous regions and SWE overestimation in flat regions (Fig. 4). The fact that in flat regions, there are larger SWE biases (Fig. 4b) and correspondingly larger SCF overestimation (Fig. 5d – 5f) in the model runs forced by GSWP3W5 supports this argument (see Section 5.1). Below we present some evidence on the link between differences in meteorological forcing datasets and choices of parameter values in the SL12 parameterization and the bias in modelled SCF.

Overall NH performance for model runs driven by ERA5 is comparable or slightly better than the runs driven by CRUJRA in terms of simulated SWE and SCF (Fig. 3, Fig. 5, and Table 2), while model runs driven by GSWP3W5 are worse everywhere except for HMA. In HMA, there is significant SCF overestimation in model runs forced by CRUJRA and ERA5, while model runs forced by GSWP3W5 have comparable SCF to observations (Fig. 5g and Table 3). For model runs forced by ERA5, this is consistent with the cold temperature bias and large precipitation overestimation in ERA5 (Fig. 3c and 3d). However, CRUJRA and GSWP3W5 exhibit similar biases in temperature and precipitation (Fig. 3c and 3d), yet model runs forced by them have contrasting SCF biases (Fig. 5g). Therefore, biases in temperature and precipitation cannot explain the SCF biases here. Instead, we found that the number of wet days (days with precipitation >= 0.1 mm) differs in each of the three datasets, especially in the HMA region (Fig. 11). Figure 11 shows that on average ERA5 has near-daily precipitation events in the mountainous areas (e.g. Tian Shan, Hindu Kush–Karakoram, and Himalayas) and southeast of TP, while GSWP3W5 has the fewest wet days over the whole HMA region, especially over the interior TP. The number of wet days in CRUJRA falls between the other two. This is consistent with differences in the SCF annual cycles (Fig. 5g) and the SCF bias patterns (Fig. 7) found among the three sets of model runs, suggesting that the different number of wet days in the forcings contributes most to the difference in modelled SCF in this region. This conclusion is also consistent with findings in previous studies (Liu et al., 2022; Orsolini et al., 2019), which suggested that excessive snowfall in ERA5 contributes to overestimation of SND, SWE, and SCF across HMA. In CLASSIC, the large number of wet days in ERA5 would lead to prolonged periods with fresh snow and therefore high snow albedo. In coupled simulations this could lead to or reinforce an existing cold bias. GSWP3W5 also has a smaller number of wet days in some other regions of the globe, such as the middle to high latitudes of NA and eastern Siberia (not shown).

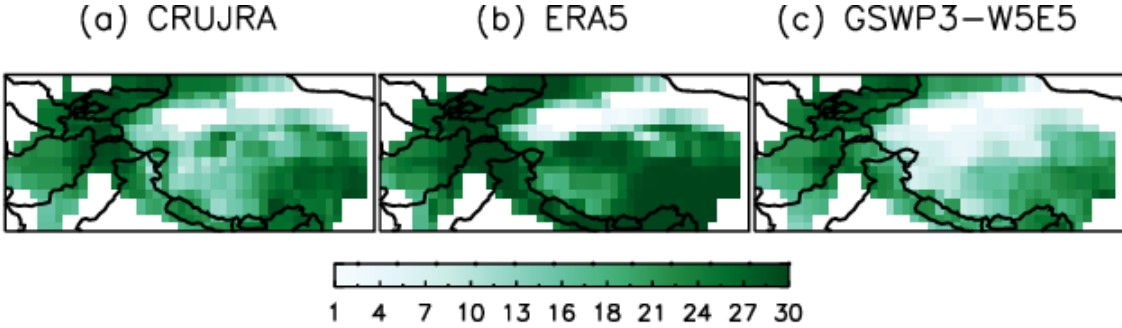

737

**Figure 11.** The monthly mean number of wet days (days with total Pr >= 0.1 mm) in (a) CRUJRA, (b)
ERA5, and (c) GSWP3W5 during the main snow season (Sep – May) in HMA over the 2005-2014
period.

Besides biases in the meteorological datasets, the choice of parameter values in the SL12
parameterization can also contribute to uncertainties in modelled SCF. As illustrated in Swenson
and Lawrence (2012, their Fig. 7), choosing a larger $k_{acc}$ parameter in Eq. (1) would result in
faster SCF increase with SND during accumulation events. All the previously discussed
simulations have used the default value of 0.1 for this parameter. We also performed sensitivity
experiments where the $k_{acc}$ parameter was changed to 0.18 and 0.26. In these simulations, SCF
increases faster with SND especially in the fall, thereby resulting in higher SCF over NH
mountainous regions during that time of year. Notably, increasing $k_{acc}$ to 0.26 produces less
biased SCF values during the fall (similar to those seen in the CTL simulations) while still
maintaining the improvements already presented during winter and spring (Fig. A6).

Likewise, the ablation portion of the SL12 parameterization (Eq. (2)) can be altered via the $N_{melt}$
parameter, which controls the rate at which SCF decreases as a function of SND. SCF decreases
faster with (normalized) SND in mountainous areas (small $N_{melt}$) than flat areas (large $N_{melt}$, Fig.
9 in Swenson and Lawrence, 2012). We adjusted the $N_{melt}$ parameter by increasing the numerator
in Eq. (3) from 200 to 300, thereby increasing the $N_{melt}$ value in mountain regions for the same
value of sub-grid topographic variability and resulting in slower SCF decrease. Results of the test
run show reduced SCF bias in the NA mountains in the spring compared to simulations with the
default $N_{melt}$ value (Fig. A7).

The adjustments to $k_{acc}$ and $N_{melt}$ parameters described above provide ways to fine-tune the
agreement in simulated SCF with observations. However, because none of the three
meteorological forcing datasets used in this study are exempt from biases, there is a limit to how
well optimal parameter values can be chosen for use in CLASSIC. In addition, it may not be
ideal to over-tune the model to a specific observational estimate which may still have
uncertainties (Section 5.3).

**5.3 Other uncertainties**

SCF derived from satellite optical sensors such as MODIS represents the viewable snow cover
from space during cloud-free overpasses (i.e., from above the canopy). Dense forests and steep
terrain may obscure the MODIS sensor's view of snow-covered ground, leading to
underestimation of SCF (Hall et al., 2002; Marchane et al., 2015). For example, Stillinger et al.
(2023) found a consistent negative bias of approximately 10% under intermediate canopy cover
when comparing MODIS SCF with high-resolution airborne lidar data in parts of the western
U.S. The SCF overestimation in flat regions (Fig. 5d–f) may be partially attributable to this
underestimation by MODIS. However, as noted by Riggs et al. (2019), snow commission errors,
often related to residual cloud contamination, are among the most common sources of error in
MODIS snow products. As a result, the SCF derived from MODIS in this study may be subject
to both underestimation and overestimation.
While the IMS snow system primarily relies on visible satellite imagery, it also incorporates
surface station observations and passive microwave data. Therefore, SCF derived from IMS is
generally less affected by cloud cover and forest canopy than that from MODIS. Previous studies
have shown that IMS tends to report higher SCF than MODIS (e.g., Brown et al., 2010), which is
consistent with our results (Fig. 5). Nevertheless, SCF estimates from MODIS and IMS are
largely consistent across all regions except the HMA, suggesting that our evaluation results are
reasonably robust despite known uncertainties.
In LSMs, snow depth is typically diagnosed from SWE and snow density. As a result,
uncertainties in modeled snow density can propagate to uncertainties in SCF, particularly when
the SCF parameterization depends on snow density and/or snow depth, as demonstrated by
Abolafia‐Rosenzweig et al. (2024). In CLASSIC, these uncertainties influence SCF simulated by
the control parameterization but do not directly affect SCF in the SL12 parameterization (Section
2.2). Since our focus is on the SL12 parameterization in this study, we do not explore this issue
further.
Additional uncertainties may arise from the elevation data used to compute the standard
deviation of sub-grid topography ($\sigma_{topo}$), particularly related to its spatial resolution. We
compared $\sigma_{topo}$ derived from two elevation datasets: ETOPO1 (1-arc-minute resolution, used in
this study) and ETOPO2022 (15-arc-second resolution). The results indicate that the differences
are limited in spatial extent and are primarily concentrated along edges of mountain ranges. To
assess the impact on model results, we conducted a test simulation using $\sigma_{topo}$ derived from
ETOPO2022 and compared the simulated SCF with that from a run based on $\sigma_{topo}$ derived from
ETOPO1. The maximum difference in SCF between the two runs was less than 5% (not shown).
These findings suggest that the resolution of the elevation data has a limited effect on the
calculation of sub-grid topographic variability and simulated SCF, consistent with sensitivity
tests reported by Lalande et al. (2023).

**6. Conclusions**
Our results demonstrate that implementing the SL12 parameterization in CLASSIC improves
simulated SCF in mountainous regions. This confirms that the lack of topographic dependency in

the current default parameterization is at least partly responsible for the SCF overestimation and cold bias in the coupled model configuration, CanESM5 (Lalande et al., 2021; Swart et al. 2019; Sigmond et al., 2023). The improved simulation of SCF also improves the simulation of surface albedo, which in turn leads to improved simulation of the surface radiation, energy fluxes, and water cycle in CLASSIC.

The results also demonstrate that the choice of meteorological forcing data can have a large impact on snow simulation in offline LSM runs. Based on our analysis, we suggest that at least part of the SWE underestimation in mountainous areas and SWE overestimation in flat areas can be linked to relative biases in temperature and precipitation from the meteorological forcing datasets. The SWE biases then propagate to biases in modelled SCF. In addition, we highlighted that bias-adjustment methods that improve temperature or precipitation separately may not result in more accurately simulated SWE, with consequences for downstream components of the water and energy cycles related to snow. These meteorological forcing datasets are regularly used to drive LSMs in various projects, such as the Global Carbon Project and ISIMIP, but for snow simulations it is important to better understand how inaccuracies in temperature and precipitation can propagate to errors in modelled SWE and SCF.

Based on the evaluation results presented in this study along with preliminary test results in fully coupled CanESM runs, the SL12 parameterization has been adopted in CLASSIC and will be used in CanESM simulations for CMIP7 submission. Future work will focus on the evaluation of the SL12 parameterization in fully coupled CanESM simulations where a full analysis of feedbacks will be possible.

*Code and data availability*. The full CLASSIC code and resulting model outputs presented in this study are archived on Zenodo at: https://doi.org/10.5281/zenodo.15032447 (Wang et al., 2025).

*Author contributions*. LW conceived this research and LW and LM designed the study. LW, LM, JM, and CM developed the analysis framework. LW and PB implemented the SL12 parameterization into the CLASSIC code. LW conducted the analysis and wrote the first draft of the manuscript. All authors contributed to manuscript review and editing.

*Competing interests*. The contact author has declared that none of the authors has any competing interests.

*Acknowledgements*. We would like to thank Mike Brady (ECCC) and Ed Chan (ECCC) for their technical assistance.

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

**Appendix A**: Supplemental Figures

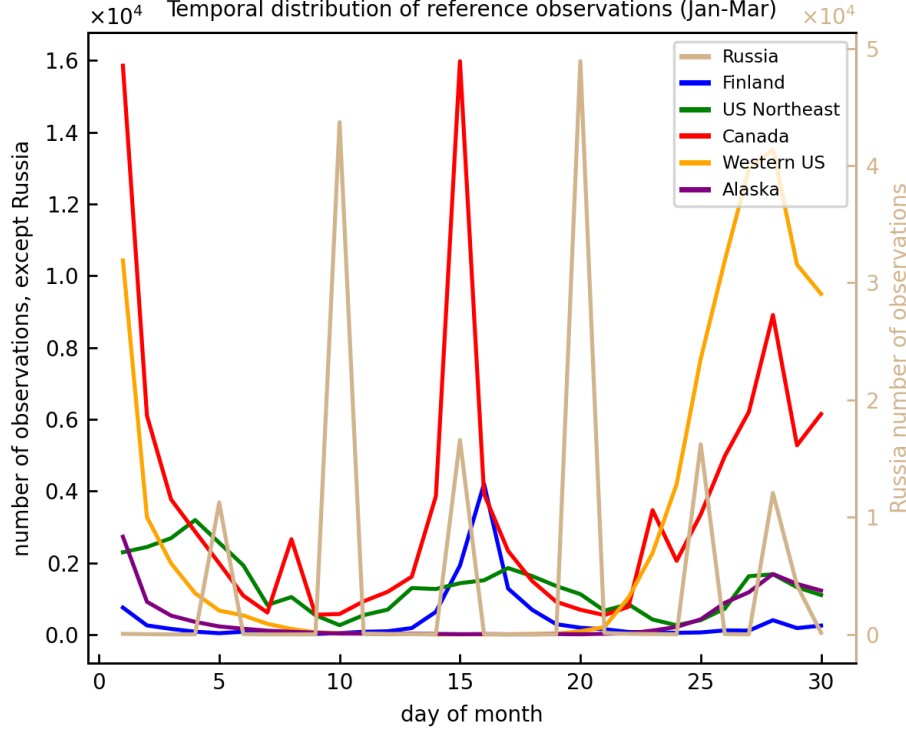


**Figure A1.** Number of reference observations by network and day of the month during 1980-2014 for the
Jan-March period.















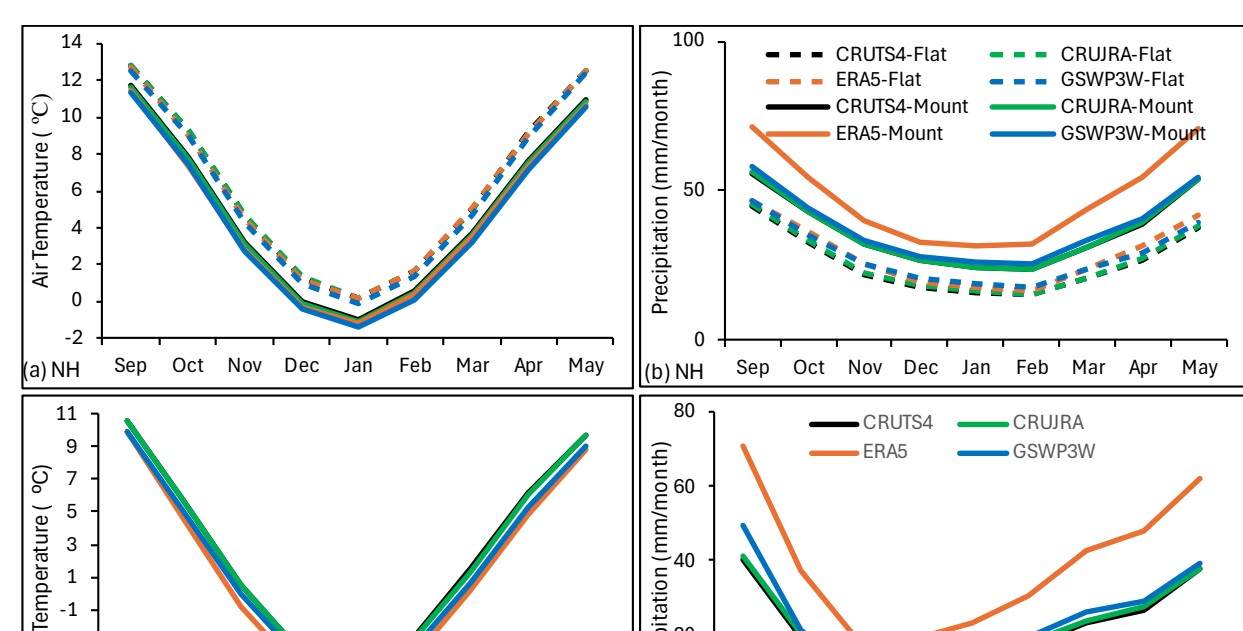


**Figure A2**. Monthly mean air temperature (a and c) and precipitation (b and d) in the NH mountainous
(solid line) and flat (dashed line) regions (a and b) and the HMA mountainous regions (c and d) over the
1980-2014 period.













**March Bias**

**Figure A3.** March SWE bias relative to in-situ measurements over the 1980-2014 period from model runs forced by each of the three meteorological forcings.

















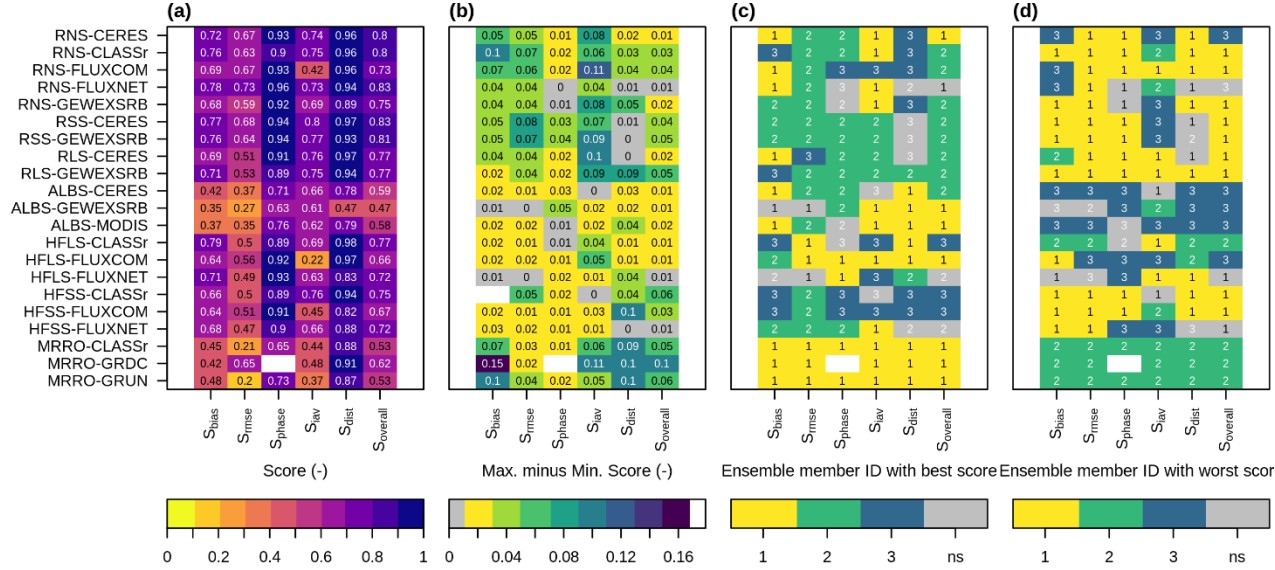


**Figure A4.** AMBER results for surface radiation, albedo, heat fluxes, and runoff from three model runs
using the CTL parameterization, (a) mean ensemble score, (b) maximum score difference among
ensemble members, (c) ensemble member with the highest score, and (d) ensemble member with the
lowest score. Ensemble member IDs represent the following model runs: 1: CRUJRA-CTL, 2: ERA5-
CTL, 3: GSWP3W5-CTL.















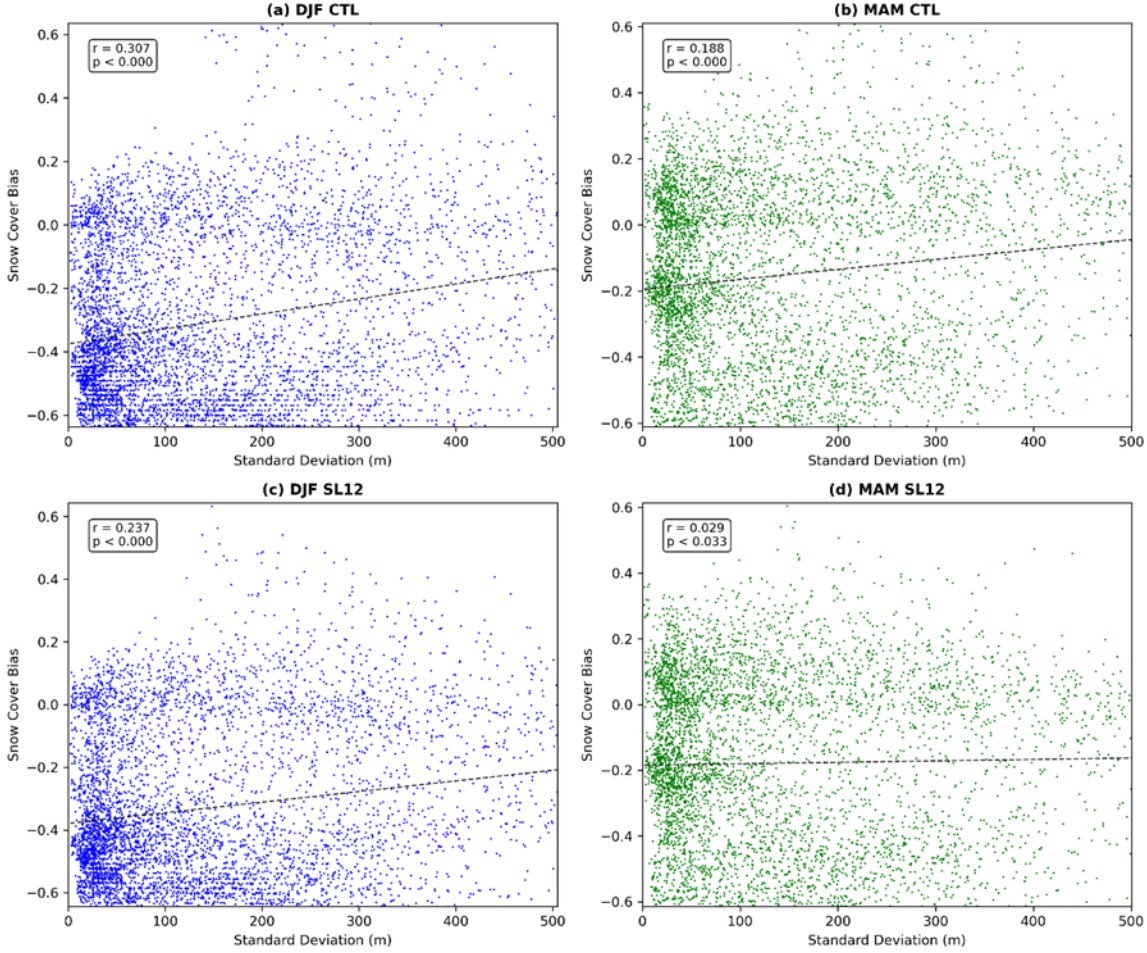


**Figure A5**. Scatter plots between SCF bias and the standard deviation of sub-grid topography during the
winter (left) and spring (right) seasons for model runs using the CTL (top) and SL12 (Bottom) schemes
forced by CRUJRA. The correlation coefficient (r) and p-value (using a two-tailed t-test) are provided in
the upper-left corner of each plot.









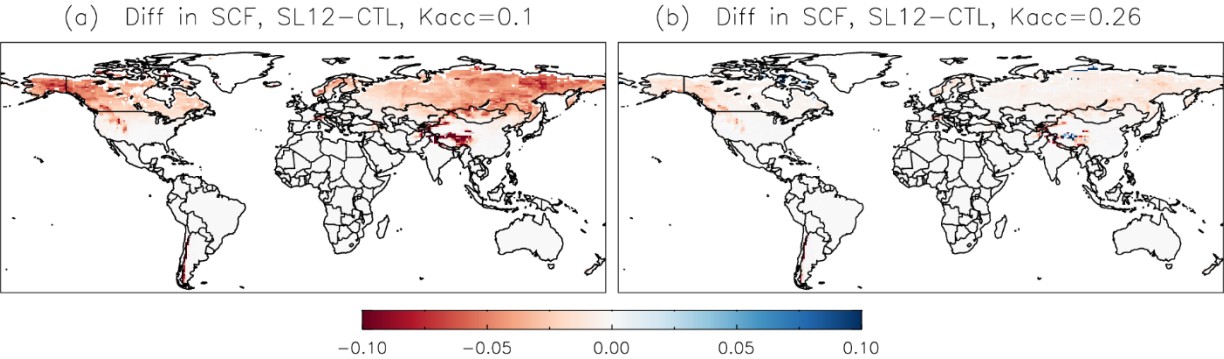


**Figure A6.** The difference in SCF between the SL12 and Control parameterizations during the fall (SON) in model runs using (a) $k_{acc}$=0.1, and (b) $k_{acc}$=0.26 for the SL12 parameterization.




















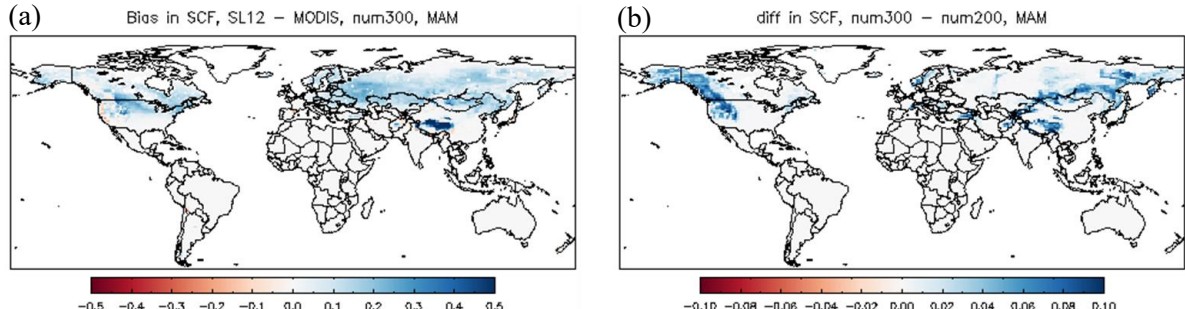

**Figure A7.** (a) Spring (MAM) SCF bias relative to MODIS using an adjusted $N_{melt}$ parameter (numerator=300 in Eq. 3), and (b) the difference in spring SCF in model runs using the adjusted (numerator=300) and default (numerator=200) $N_{melt}$ parameter.