# Peer review of "1. Introduction"

_EGUsphere, 2025_

## Referee Comment (RC1)

**Summary and overarching comments:**

This study evaluates the impacts of an alternate SCF parameterization, that calculates SCF as a function of topographic complexity and SWE, in offline 1-degree CLASSIC model simulations. The alternate SCF parameterization tends to improve the accuracy of SCF simulated in CLASSIC during winter months in topographically complex regions. This study also explores the robustness of results to differing metrological forcing sources, which reveals the large impact of metrological forcing in snow simulation accuracy. This study provides a novel and important advancement for the CLASSIC modeling system that seems to allow land model simulations to better capture SCF, and in turn improve land-atmosphere interactions due to the snow-albedo feedback. Overall, the paper is well written and the study will warrant a publication after addressing the comments below.

I have four overarching critiques for this analysis. (1) A key motivation for improving SCF in model simulations is to enhance simulated albedo. Although the study briefly covers the impacts of SCF on albedo accuracy using the AMBER score, it would be useful to go into more detail on the albedo analysis which is a critical component of this study. (2) MODIS SCF has questionable accuracy, particularly for representing ground SCF which the land model simulates. This point should be more directly addressed with the consideration of other data sources for SCF (e.g., STC-MODSCAG across the western US, see suggestion below). (3) Discrepancies in spatial resolution between reference data used to validate model simulations and the spatial resolution of the model simulations can largely impact results. Please see specific comment below addressing this point. (4) Figure quality should be improved throughout.

**Specific recommendations:**

Paragraph starting in line 81: Note that some land surface models also consider SCF as a function of snow density and land cover classification (e.g., He et al., 2023).

He, C., et al. *The community Noah-MP land surface modeling system technical description version 5.0*. NCAR Technical Note NCAR/TN-575+ STR, doi: 10.5065/ew8g-yr95, 2023.

Section 2.1: please articulate the capacities in which CLASSIC is used, for either research applications or operational modeling.

Section 3.1 and Figure 1: Please add information on the calculation of topographic standard deviation. Specifically, what is the resolution of the elevation product which is used to calculate this metric?

Section 3.2: Another potential issue with MODIS SCF is not just its accuracy, but also whether its retrieval represents pixel scale SCF or just the ground SCF. Many land models simulate ground SCF, rather than total pixel SCF (e.g., including vegetated fractions of the pixel) and thus a comparison with the MODIS data used here may not be appropriate. The STC-MODSCAG data addresses this issue, and the latest version has available data across the mountainous western US (https://nsidc.org/data/stc_modscgdrf_hist/versions/1#anchor-data-access-tools). Please consider using these data as an additional reference to evaluate whether the comparisons against MODIS are reliable.

Lines 315-316: Simulated snow density is also a source of SCF uncertainty, e.g., Abolafia-Rosenzweig et al. (2024), which could be noted here or in the Discussion.

Abolafia-Rosenzweig, Ronnie, et al. "Evaluating and enhancing snow compaction process in the Noah-MP land surface model." *Journal of Advances in Modeling Earth Systems* 16.2 (2024): e2023MS003869.

Section 3.3: These SWE evaluations are likely largely impacted by discrepancies between observed and modelled spatial resolutions. It would be good to emphasize this point further, even in the case of airborne gamma SWE observations. To consider the spatial representativeness of observations, consider comparing time series from in-situ stations contained by the same 1-degree pixel and consider whether there are large discrepancies (e.g., with bias and correlation metrics).

Also, when observations are measured infrequently (e.g., a few times in a month) are the modelled data screened temporally to match the observational frequency prior to comparison?

Line 405: Is there truly no feedback in these offline runs? Land models often calculate 2-m air temperature prognostically which could impact SWE. If this is the case for the CLASSIC model, consider re-wording here.

Lines 424-430: Adding more quantitative information here would be useful.

It looks like the simulations tend to underestimate SWE substantially; however, there is a tendency to overestimate winter SCF, largely in the control simulations and modestly in the SL12 simulations. If the SCF scheme is truly accurate at converting SWE or snow depth to SCF then

we would expect to see underestimates in SCF. Can this point be added, particularly connecting logic between Sections 4.2 and 4.3?

It would be interesting to consider whether there are significant correlations between SCF biases with topographic complexity in each of the simulations, and in particular highlight if the SL12 scheme reduces or removes this relationship.

Section 4.4: It would be valuable to note whether the albedo biases are consistent with SCF biases (e.g., locations with SCF overestimates have albedo overestimates).

Section 5: here are some potentially useful references for land model SWE biases:

He, Cenlin, et al. "What causes the unobserved early-spring snowpack ablation in convection-permitting WRF modeling over Utah Mountains?." *Journal of Geophysical Research: Atmospheres* 126.22 (2021): e2021JD035284.

Abolafia-Rosenzweig, Ronnie, et al. "Implementation and evaluation of a unified turbulence parameterization throughout the canopy and roughness sublayer in Noah-MP snow simulations." *Journal of Advances in Modeling Earth Systems* 13.11 (2021): e2021MS002665.

Chen, Fei, et al. "Modeling seasonal snowpack evolution in the complex terrain and forested Colorado Headwaters region: A model intercomparison study." *Journal of Geophysical Research: Atmospheres* 119.24 (2014): 13-795.

von Kaenel, Manon, and Steven A. Margulis. "Evaluation of Noah-MP snow simulation across site conditions in the western United States." *Journal of Hydrometeorology* 25.9 (2024): 1389-1406.

---

## Author Comment (AC2)

**We thank Referee #2 for their helpful comments. Our replies to their comments are shown in bold below.**

Review of Wang et al., "Impact of topography and meteorological forcing on snow simulation in the Canadian Land Surface Scheme Including Biogeochemical Cycles (CLASSIC)"

This study examines the impact of replacing the default CLASSIC snow cover fraction (SCF) parameterization with an alternative one. The study has two objectives: 1) comparing the two SCF parameterizations and 2) examining the role of meteorological forcing on the simulation of snow. The role of meteorological forcing is highlighted by using three input datasets to force the model. The default parameterization predicts SCF from snow depth using a linear relationship for snow depth below 0.1m; above 0.1m, SCF is 1. The alternative considers the accumulation and ablation seasons separately, and also incorporates topographic information to adjust its behavior spatially based on the topographic variability within a region. The authors find that comparisons to MODIS SCF are more favorable when using the alternate SCF parameterization. In addition, other metrics related to water and energy fluxes show improvement.

General comments:

The authors state that tests changing the lone parameter (0.1) in the CTL parameterization show minimal impacts to their simulations. In contrast, in addition to a structural change, SL12 offers opportunities to improve the SCF simulation by modifying and calibrating equations 1 and 3. While the authors mention a couple of changes to the k_acc and N_melt parameters that led to positive results, they choose not to explore the parameter sensitivity in more detail "because none of the three meteorological forcing datasets used in this study are exempt from biases, there is a limit to how well optimal parameter values can be chosen for use in CLASSIC". The results shown in figure 5 of the manuscript seem to contradict this statement, as the biases seem consistent across forcings. The authors support this by stating "On the global scale, the spatial patterns of SCF bias are similar for all three meteorological forcing choices."

**Thank you for your overall positive review of our manuscript.**

I believe that this study would be improved if the authors were to pursue this path. The authors could perform a few shorter, initialized runs (e.g. from 1980 onwards) to do a sensitivity study of the N_melt parameter. They could then use these results to see if a better function for N_melt as a function of sigma_topo becomes apparent. For example, figure 5 indicates that lower SCF values in winter and spring, irrespective of forcing input, are preferred for flat regions. This implies that the N_melt equation increase too rapidly for small values of sigma_topo. This is perhaps not surprising, given that 1/x blows up as x goes to zero. A bounded function, e.g. a decaying exponential, might improve the results for flat regions, while maintaining the good results for mountainous regions. Similarly, adding a simple dependence on sigma_topo to k_acc might improve the fall bias shown in figure 5 b) without degrading other regions.

**Thank you for your suggestions. We agree that it would be ideal if the SL12 parameterization could be calibrated to improve modelled SCF in both the mountain and flat regions. Initially we tried many sensitivity experiments to try to achieve this goal. The discussion section of the paper provides some generalized results of the experiments and how this process worked (lines 685-702). However, after gaining a better understanding of the uncertainties in the meteorological forcings and observed SCF datasets, we think it is a goal impossible to achieve at present. Below are the reasons:**

(1) Evaluation based on measurements from snow course and airborne gamma data showed that for all three choices of forcing data, modelled SWE is underestimated in the mountain and overestimated in the flat regions throughout the snow season (Fig. 4). Since SCF is directly linked to SWE in the SL12 scheme (see Eq.1 and Eq. 2), these SWE biases can exert a large impact on simulated SCF in the fall and spring seasons in the model (limited impact during the peak SWE period for SCF is usually saturated, details can be found in our reply to RC1). The consistent SCF biases shown in Figure 5 are linked to these consistent SWE biases for all three forcing choices in the model.

(2) SCF derived from satellite optical sensors such as MODIS represents the visible snow cover from space during cloud-free overpasses - that is, from above the vegetation canopy. In contrast, SCF from the CLASSIC model represents ground-level SCF, including snow cover beneath the canopy. As a result, MODIS-derived SCF tends to be biased low in forested regions. This limitation has been noted in previous studies (Hall et al., 2002; Hall and Riggs., 2021). Using high-resolution airborne lidar data from the western United States, Stillinger et al. (2023) found that the MODIS SCF product exhibited a consistent negative bias of approximately -0.10 under intermediate canopy cover, with the bias increasing with greater snow cover, reaching -0.25 under full snow cover conditions (peak snow season). Thus, it may not be ideal to over-tune the model to a specific observational estimate which may still have errors.

(3) Even if the SL12 parameterization and its associated parameters are perfectly specified, and the reference SCF for comparison was a perfect measure of ground truth, biases in the modelled SCF will still arise due to inaccuracies in the meteorological forcing data. We may revisit this if more realistic meteorological forcing datasets are available in the future.

We discussed the impact of meteorological forcing datasets on modelled SWE in Section 5.1. We will revise the text to clearly state the link between biases in SWE and SCF. We will acknowledge the uncertainties of the MODIS SCF product mentioned above and provide a brief discussion on its impact when revising our manuscript.

About you other comments with regard to "…for small values of sigma_topo", note the denominator of Eq. (3) is max(10, $\sigma_{topo}$), so the maximum $N_{melt}$ parameter is 20. Attempting to develop a different SCF parametrization is beyond the scope of the study.

Hall, D.K., Riggs, G.A., Salomonson, V.V., DiGirolamo, N.E., & Bayr, K.J.: MODIS snow-cover products. Remote Sensing of Environment, 83(1–2), 181–194. http://dx.doi.org/10.1016/S0034-4257(02)00095-0, 2002.

Hall, D. K. & Riggs, G. A.: MODIS/Terra Snow Cover Monthly L3 Global 0.05Deg CMG. (MOD10CM, Version 61). Boulder, Colorado USA. NASA National Snow and Ice Data Center Distributed Active Archive Center. https://doi.org/10.5067/MODIS/MOD10CM.061. Date Accessed 06-19-2025, 2021.

Stillinger, T., Rittger, K., Raleigh, M. S., Michell, A., Davis, R. E., and Bair, E. H.: Landsat, MODIS, and VIIRS snow cover mapping algorithm performance as validated by airborne lidar datasets, The Cryosphere, 17, 567–590, https://doi.org/10.5194/tc-17-567-2023, 2023.

Stillinger, T., Rittger, K., Raleigh, M. S., Michell, A., Davis, R. E., and Bair, E. H.: Landsat, MODIS, and VIIRS snow cover mapping algorithm performance as validated by airborne lidar datasets, The Cryosphere, 17, 567–590, https://doi.org/10.5194/tc-17-567-2023, 2023.

Specific comments:

Lines 99,100: add references for CLM5, CESM2

**Thanks for noting this, we will add references when revising our manuscript.**

Line 130: how do the four sub-areas relate to SCF? Do the snow/snow-free areas change dynamically?

**The areal fractions of the four subareas - vegetation over bare soil , bare soil, vegetation over snow, and snow over bare soil, are calculated based on the fractional coverage of the vegetation categories and SCF.**

**Yes, the snow-covered and snow-free areas change dynamically at each time step.**

Line 144: "all vertical layers": I thought there was only 1 layer (line 132)?

**We meant for soil layers beneath the snow layer as well. We will clarify this when revising our manuscript.**

Line 160: is there reason to think k_acc should vary spatially? If so, how might one parameterize it? (discussed in sec 5.2). Also, SL12 mentions that eq 1 assumes snowfall is randomly distributed in the region; is this a valid assumption?

**The SL12 parameterization was developed based on the relationship between snow depth from Snow Data Assimilation System (SNODAS) and SCF from MODIS over the continental US (Swenson and Lawrence, 2012). Topographic dependency between SCF-SND was not observed during the accumulation period. More details can be found in Swenson and Lawrence (2012).**

**About "eq 1 assumes snowfall is randomly distributed", this assumption may not be valid in mountain regions where snowfall affects preferentially high-elevation areas. We will add this when revising our manuscript.**

**Swenson, S. C. and Lawrence, D. M.: A new fractional snow-covered area parameterization for the Community Land Model and its effect on the surface energy balance, J. Geophys. Res.-Atmos., 117, D21107, https://doi.org/10.1029/2012JD018178, 2012.**

Line 173: how are the parameters 200 and 10 chosen? How sensitive are the results to these parameters, and could they instead be calibrated?

**The parameters were determined based on observed relationship between SCF-SND, details can be found in Swenson and Lawrence (2012). We performed sensitivity experiments where 200 in the numerator of Eq. (3) were changed to 300, 100, and 50. An example of the results is discussed in Section 5.2 and another figure is included below in response to your comment on bias in flat regions.**

Line 181: does SL12 implemented in CLM5 use time of year to determine which equation to use?
**No. It is also based on whether SWE is increasing or decreasing with respect to the previous time step.**

Why is equation 4 used?  Isn't W_max based on the evolution of W in the model, i.e. is it the peak SWE of each snow season?

**W_max is the accumulated maximum SWE at each time step, which is different from the peak SWE of each snow season.**

Line 262: does the resolution of the DEM affect the calculation of the standard deviation of the sub-grid terrain?

**To assess the impact of DEM resolution, we compared $\sigma_{topo}$ derived from two DEM datasets: ETOPO1 (1-arc-minute resolution, used in our study) and ETOPO2022 (15-arc-second resolution). The results show that the differences are limited in extent, primarily concentrated along the mountain edges. We also performed a test simulation using $\sigma_{topo}$ derived from ETOPO2022 and compared modelled SCF with that from a run using $\sigma_{topo}$ derived from ETOPO1. The maximum difference was less than 5%.**

**The resolution of the DEM data has limited impact on the calculation of sub-grid topographic variability and the simulated SCF.**

Line 271: 'high mountainous asia' or 'high mountain asia'?

**Thanks for noting this typo.**

Line 297: IMS data could be converted to 1 degree fractional values, then treated similarly to MODIS; is that how IMS data is processed?

**The IMS dataset provides binary snow/no snow information: if more than 50% of the 4 km pixel is covered by snow, it has a value of 1, otherwise 0 (snow free). If we aggregate the 4km IMS snow/no snow data into SCF at 1 degree, the derived SCF would have an uncertainty range from 50% to 100%.**

**In our study, daily IMS data were converted to monthly snow cover duration fraction (SCF = total number of days with snow cover in a month divided by the number of days in the month), which we found more comparable with SCF from MODIS.  This method was used in previous studies (Brown et al., 2010; Wang et al., 2014).**

**Brown et al. 2010, https://doi:10.1029/2010JD013975.**

**Wang et al., 2014, https://doi.org/10.1175/JHM-D-13-086.1.**

Line 315: SND is related to SWE via snow density; how is snow density calculated in CLASSIC?

The density of fresh snow density ($\rho_{s,i}$) is determined as an empirical function of the air temperature (Ta). For Ta <= 0°C, an equation presented by Hedstrom and Pomeroy (1998) is used. For Ta > 0°C, a relation following Pomeroy and Gray (1995) is used, with an upper limit of 200 kg m$^{-3}$:

$$\rho_{s,i} = 67.92 + 51.25 \ \exp[Ta/2.59] \quad Ta < 0°C$$
$$\rho_{s,i} = 119.17 + 20.0Ta \qquad\qquad Ta \geq 0°C$$

Over time, snowpack density ($\rho_s$) increases due to the effects of crystal settlement and metamorphism in the snowpack, sublimation, wind packing, melt, and refreezing. CLASSIC models this using an empirical relationship with time:

$$\rho_s(t) = \rho_s(t-1) \ \exp(-B\Delta t)$$

Where $\Delta t$ is the time step and $B = 0.01/3600$ is a constant. More details can be found in Verseghy (1991) and Bartlett et al. (2006).

Verseghy, D. L., 1991, https://doi.org/10.1002/joc.3370110202.
Bartlett et al., 2006, https://doi:10.3137/ao.440301.

Line 363: are there 21 datasets, or 7 datasets?

A unique feature of AMBER is that it uses multiple reference datasets when available to evaluate the same variable. In the AMBER results shown in our study, seven variables were evaluated using 21 reference datasets (shown in Table 1).

Figure 2: add units to (d) colorbar

Thank you for noting this, we will add units when revising our manuscript.

Figure 3: please label figures with NH or HMA.  Also adding a dashed line to indicate zero for each y-axis would be helpful.

Thank you for your suggestions. We will modify the figure as suggested when revising our manuscript.

Figure 4: why only show NH, but not HMA like other sections?  Does HMASR not provide SWE?

In situ SWE measurements are not available for the HMA region. Though HMASR provides SWE, it is still model outputs with unknown uncertainties, not the same as those from snow course and airborne gamma data shown in Fig.4.

We feel the temperature and precipitation comparison in Fig. 3b and the number of wet days shown in Fig. 10 are sufficient to explain the differences in simulated SCF over the HMA region.

Figure 4: perhaps replace 'mount' and 'flat' with 'Mountainous Regions (sigma > 200m)' and 'Flat Regions (sigma < 200m)'

**Thanks for your suggestions. We will change the title to 'Mountainous Regions' and 'Flat Regions' and include the definition for mountain/flat regions in the caption, to be consistent with other figures.**

Figure 4: how do errors compare to magnitude of SWE, e.g. figure 2 d), which shows maximum values of 115?

**The maximum SWE is up to 2000 mm in the mountain regions. We will replot Figure 2d to show more range for SWE.**

Line 403: CTL and SL12 do not cause any albedo feedbacks to cause changes in SWE (via surface energy balance)?

**Thanks for noting this. Yes, there is snow-albedo feedback, which affects simulated SWE in the model. We will modify the sentence when revising our manuscript.**

Figure 5: improvements mainly in 2nd half of snow season for mountainous regions; is this due to the ablation part of the SL12 parameterization?

**Yes. The parameterization for ablation in SL12 accounts for sub-grid topographic variability.**

Figure 5: SL12 shows similar results in the NA and EA flat regions (the flat region biases begin early in the season (around Dec/Jan)), can that be improved by calibrating with the parameters in eqn 3?

**Yes, that can be improved by tuning parameters in Eq. (3), details and an example of results (Fig. AR1) can be found in our reply below for a related comment. However, we choose not to use the tuned parameters in CLASSIC, please see our reply above to your main comments.**

Figure 6: I would use a colormap that was not white in the middle for panels a) and b). Perhaps simply linear white-to-blue?

**Thanks for noting this, we will use a different colorbar in the revised manuscript.**

Figure 7: SWE evaluation for HMA in section 4 would help understand differences in forcing data. Is the cruja/era5 overestimate due to a SWE high bias, or does it come from the SCF parameterizations?

**Given that the same CTL/SL12 schemes were used across all model runs, the substantial difference in simulated SCF in the run forced by GSWP3W5 (compared to the other two) indicates that the primary cause of the discrepancy is the difference in the forcing data. To improve clarity and logical flow, we will include the above sentence when revising our manuscript.**

**In addition, high SWE bias in ERA5 was well documented in previous studies (Liu et al., 2022; Orsolini et al., 2019), which suggested that excessive snowfall in ERA5 contributed to overestimation of SND, SWE, and SCF across HMA (also in Line 672-675).**

Line 441: SL12 is shown to perform slightly worse than CTL in fall (SON) in mountainous regions but not flat regions in NA. What might cause this? Is it more due to the accumulation equation or the ablation equation?

**Thank you for your comments and suggestions. This is mainly due to the accumulation formula, which can be improved by increasing the $k_{acc}$ parameter in Eq. (1), details can be found in Section 5.2 (L685-694).**

For the flat regions, the spring bias is similar for both SL12 and CTL. What does that say about SL12, i.e. would a more rapid SCF decrease improve the results? Does that imply that equation 3 is not optimal for flat regions, and the 1/sigma_topo behavior might be too large for small sigma_topo?

**We conducted sensitivity experiments by varying the value of 200 in the numerator of Eq. (3) to 300, 100, and 50. Among the tested configurations, the following combination produced the smallest overall SCF bias (Fig. AR1): for grid cells with $\sigma_{topo} \geq 100m$, a numerator of 200 was used; for cells with $\sigma_{topo} < 100m$, a numerator of 50 was applied. The map on the right (using the modified $N_{melt}$) shows reduced positive bias in flat regions compared to the one on the left (using the default $N_{melt}$).**

[Figure]

**Figure AR1. Bias in simulated SCF using the SL12 scheme with the default parameter (left) and the modified parameter (right). SCF from MODIS was used as the reference.**

**However, we choose not to use the tuned parameters in CLASSIC, please see our reply above to your main comments.**

Line 664: does the 'wet day' dependence indicate that one of the accumulation / ablation equations in SL12 has a bigger impact on the SCF evolution?

**The number of wet days influences the frequency of new snowfall, thereby directly affecting SCF during the accumulation period (Eq. 1). It also determines the amount of snow stored on the ground, which in turn impacts SCF through the ablation processes (Eq. 2). The dominant process controlling SCF evolution likely depends on the regional climate. In cold regions, where melting is rare during the accumulation season, SCF evolution is primarily governed by the accumulation**

**process. In contrast, in intermediate and warmer climates where accumulation and melt cycles are more frequent, both accumulation and ablation processes may contribute comparably to SCF evolution.**

---

## Author Comment (AC4)

https://egusphere.copernicus.org/preprints/egusphere-2025-1264#RC1

https://doi.org/10.5194/egusphere-2025-1264-RC1

**We thank Referee #1 for their helpful comments. Our replies to their comments are shown in bold below.**

Summary and overarching comments:

This study evaluates the impacts of an alternate SCF parameterization, that calculates SCF as a function of topographic complexity and SWE, in offline 1-degree CLASSIC model simulations. The alternate SCF parameterization tends to improve the accuracy of SCF simulated in CLASSIC during winter months in topographically complex regions. This study also explores the robustness of results to differing metrological forcing sources, which reveals the large impact of metrological forcing in snow simulation accuracy. This study provides a novel and important advancement for the CLASSIC modeling system that seems to allow land model simulations to better capture SCF, and in turn improve land-atmosphere interactions due to the snow-albedo feedback. Overall, the paper is well written and the study will warrant a publication after addressing the comments below.

**Thank you for your overall positive review of our manuscript.**

I have four overarching critiques for this analysis. (1) A key motivation for improving SCF in model simulations is to enhance simulated albedo. Although the study briefly covers the impacts of SCF on albedo accuracy using the AMBER score, it would be useful to go into more detail on the albedo analysis which is a critical component of this study. (2) MODIS SCF has questionable accuracy, particularly for representing ground SCF which the land model simulates. This point should be more directly addressed with the consideration of other data sources for SCF (e.g., STC-MODSCAG across the western US, see suggestion below). (3) Discrepancies in spatial resolution between reference data used to validate model simulations and the spatial resolution of the model simulations can largely impact results. Please see specific comment below addressing this point. (4) Figure quality should be improved throughout.

**Thank you for your suggestions.**

(1) A key motivation for improving SCF in model simulations is to enhance simulated albedo. Although the study briefly covers the impacts of SCF on albedo accuracy using the AMBER score, it would be useful to go into more detail on the albedo analysis which is a critical component of this study.

**We agree that it would be helpful to provide more details on the impact of the SL12 parameterization on simulated surface albedo in CLASSIC, especially considering the large impact of snow cover on surface albedo. We will include a figure comparing surface albedo simulated by the model runs using the Control and SL12 parameterizations for various regions with observations when revising our manuscript (see figure below in our reply to your Specific recommendations).**

(2) MODIS SCF has questionable accuracy, particularly for representing ground SCF which the land model simulates. This point should be more directly addressed with the consideration of other data sources for SCF (e.g., STC-MODSCAG across the western US, see suggestion below).

We agree that SCF derived from satellite optical sensors like MODIS is viewable snow cover from space during cloud-free overpasses (i.e. from above the canopy), while SCF from CLASSIC represents ground-level SCF (including snow cover beneath the canopy).

Thank you for bringing our attention to the STC-MODSCAG data, which provides snow estimate on the ground and is better suitable for evaluating modelled SCF. However, it is currently only available for the western US, a global dataset is required to evaluate model performance in our study. Previous studies have shown that the accuracy of SCF from MODIS is lower than that from MODSCAG (Painter et al., 2009; Stillinger et al., 2023). Evaluating SCF from standard MODIS and STC-MODSCAG with high resolution airborne lidar data in western US, Stillinger et al. (2023) showed that the median bias (RMSE) was -0.071 (0.127) for MODIS and -0.001 (0.120) for STC-MODSCAG across various snow climates. They also showed that the MODIS SCF product exhibited consistent negative bias of around -0.10 under intermediate canopy cover, with the bias increasing with greater snow cover, reaching -0.25 under full snow cover conditions.

In addition, other evaluation studies suggested the accuracy of MODIS snow products was in the range of 88-93%, and dense forests and steep terrain may obscure the MODIS sensor's view of snow-covered ground, resulting in SCF underestimation (Hall et al., 2002; Hall and Riggs., 2021).

We will acknowledge the uncertainties of the MODIS SCF product mentioned above and provide a brief discussion on its impact on our results when revising our manuscript.

Hall, D.K., Riggs, G.A., Salomonson, V.V., DiGirolamo, N.E., & Bayr, K.J.: MODIS snow-cover products. Remote Sensing of Environment, 83(1–2), 181–194. http://dx.doi.org/10.1016/S0034-4257(02)00095-0, 2002.

Hall, D. K. & Riggs, G. A.: MODIS/Terra Snow Cover Monthly L3 Global 0.05Deg CMG. (MOD10CM, Version 61). Boulder, Colorado USA. NASA National Snow and Ice Data Center Distributed Active Archive Center. https://doi.org/10.5067/MODIS/MOD10CM.061. Date Accessed 06-19-2025, 2021.

Painter, T. H., Rittger, K., McKenzie, C., Slaughter, P., Davis, R. E., and Dozier, J.: Retrieval of subpixel snow covered area, grain size, and albedo from MODIS, Remote Sens. Environ., 113, 868–879, https://doi.org/10.1016/j.rse.2009.01.001, 2009.

Stillinger, T., Rittger, K., Raleigh, M. S., Michell, A., Davis, R. E., and Bair, E. H.: Landsat, MODIS, and VIIRS snow cover mapping algorithm performance as validated by airborne lidar datasets, The Cryosphere, 17, 567–590, https://doi.org/10.5194/tc-17-567-2023, 2023.

(3) Discrepancies in spatial resolution between reference data used to validate model simulations and the spatial resolution of the model simulations can largely impact results. Please see specific comment below addressing this point.

To minimize these issues, we rely on snow courses and airborne gamma measurements because they are more spatially representative than single point measurements (Meromy et al. 2013). Snow courses consist of multiple measurements along a transect several hundreds of metres to kilometres in length that are averaged together to provide a single SWE value. Airborne gamma measurements are averaged across 300 m wide footprints and along 15–20 km long flight lines. In both cases, these measurements better sample the sub-grid-scale variability than a single-point measurement and so are more effective in capturing the larger-scale average. This decision does not fully close the scale difference between observations and gridded product, but it helps substantially.

In addition, analysis in Mortimer et al. (2024) showed that evaluation of gridded products with spatial resolutions ranging from 4km to 1.25° using this type of reference data yielded consistent performance ranking whether evaluated with airborne gamma or snow courses in non-mountain or mountain areas. This means we can make meaningful relative assessments of the gridded product performance.

In this manuscript, our intent is to provide readers with a sense of the relative simulated SWE errors driven by differences in the forcing data. We believe the reference data are appropriate for this purpose. However, at your suggestion, we also looked further into the sampling variability of the bias within a 1x1 degree CLASSIC grid cell. We identified a subset of grid cells containing multiple reference sites with long records. For simplicity, we restricted this demonstration to February. To remove issues related to sampling dates within a month, we only compared reference sites collected on the same date. As Figure AR1 shows, in nearly all cases, the ranking of SWE magnitudes for each of the three products and the reference SWE are similar from year to year. This demonstrates that the relative product errors assessed in the manuscript are likely to be consistent even if temporally sampled less frequently than demonstrated here. In nearly all cases the product SWE also falls outside of the standard deviation of the reference SWE. This demonstrates that the calculated biases presented in the manuscript are likely to be meaningful, even if spatially sampled less frequently than demonstrated here (however in most cases our arguments rely only on the relative product bias anyway).  In rare cases (e.g. red box), the choice of reference site will alter the sign of the bias (but still does not alter the relative sense of bias among the three products).

[Figure]

**Figure AR1. Mean and standard deviation of reference SWE (blue) for sites measured on the same date within the same model grid cell. When there were multiple dates in the same month, the mean and standard**

deviation were calculated for sites measured on the same date and then averaged across the month-year. Dots: modelled SWE for the corresponding grid cell and month. Far-right black line, triangle, and hollow circles show the mean across the time series. For display, only February is shown. Each plot corresponds to one dot on the map (lon/lat listed at top of each plot and colors of plot axes correspond to dots colors on map). Sites in the western US are mountainous, all other are in flat regions as defined in the manuscript.

Meromy, L., Molotch, N. P., Link, T. E., Fassnacht, S. R., and Rice, R.: Subgrid variability of snow water equivalent at operational snow stations in the western USA, Hydrol. Process., 27, 2383–2400, https://doi.org/10.1002/hyp.9355, 2013.

Mortimer, C., Mudryk, L., Cho, E., Derksen, C., Brady, M., and Vuyovich, C.: Use of multiple reference data sources to cross-validate gridded snow water equivalent products over North America, The Cryosphere, 18, 5619–5639, https://doi.org/10.5194/tc-18-5619-2024, 2024.

(4) Figure quality should be improved throughout.

**We apologize for the poor quality of the figures. The quality was fine in the original Microsoft Word version of the manuscript but deteriorated after converting to the pdf file. We will make sure the figures will all have high quality in the revised manuscript.**

Specific recommendations:

Paragraph starting in line 81: Note that some land surface models also consider SCF as a function of snow density and land cover classification (e.g., He et al., 2023). He, C., et al. The community Noah-MP land surface modeling system technical description version 5.0. NCAR Technical Note NCAR/TN-575+ STR, doi: 10.5065/ew8g-yr95, 2023.

**Thank you for your suggestion. We will add this point when revising our manuscript.**

Section 2.1: please articulate the capacities in which CLASSIC is used, for either research applications or operational modeling.

**We agree that it would be nice to include a couple of sentences about the applications of CLASSIC. We will add this when revising our manuscript.**

Section 3.1 and Figure 1: Please add information on the calculation of topographic standard deviation. Specifically, what is the resolution of the elevation product which is used to calculate this metric?

**This was already provided in the manuscript: "Classification of mountain and flat regions is based on standard deviation of the sub-grid terrain from the ETOPO1 elevation data at 1 arc-minute resolution (NOAA, 2009)."**

Section 3.2: Another potential issue with MODIS SCF is not just its accuracy, but also whether its retrieval represents pixel scale SCF or just the ground SCF. Many land models simulate ground SCF, rather than total pixel SCF (e.g., including vegetated fractions of the pixel) and thus a comparison with the MODIS data used here may not be appropriate. The STC-MODSCAG data addresses this issue, and

the latest version has available data across the mountainous western US
([https://nsidc.org/data/stc_modscgdrf_hist/versions/1#anchor-data-access-tools](https://nsidc.org/data/stc_modscgdrf_hist/versions/1#anchor-data-access-tools)).

Please consider using these data as an additional reference to evaluate whether the comparisons against MODIS are reliable.

**Please see our reply above to your main comments. When revising our manuscript, we will include a summary on the evaluation results of the MODIS SCF data from previous studies to provide some uncertainties on our comparison against MODIS.**

Lines 315-316: Simulated snow density is also a source of SCF uncertainty, e.g., Abolafia-Rosenzweig et al. (2024), which could be noted here or in the Discussion. Abolafia-Rosenzweig, Ronnie, et al. "Evaluating and enhancing snow compaction process in the Noah-MP land surface model." Journal of Advances in Modeling Earth Systems 16.2 (2024): e2023MS003869.

**Thank you for your suggestion and providing the reference. We will add this point when revising our manuscript.**

Section 3.3: These SWE evaluations are likely largely impacted by discrepancies between observed and modelled spatial resolutions. It would be good to emphasize this point further, even in the case of airborne gamma SWE observations. To consider the spatial representativeness of observations, consider comparing time series from in-situ stations contained by the same 1 degree pixel and consider whether there are large discrepancies (e.g., with bias and correlation metrics).

**Please see our reply above with Figure AR1. Most of our conclusions about forcing-driven errors are based on the assessed bias.**

Also, when observations are measured infrequently (e.g., a few times in a month) are the modelled data screened temporally to match the observational frequency prior to comparison?

**The model output was only saved at monthly frequency. How well our date-specific samples will represent a true monthly mean will depend on their distribution over the month of interest. We examine two aspects of this in detail below: lack of snow-free reference measurements and the distribution of measurements within a month.**

**Despite the challenges highlighted below, we are confident that for the application used in our study, the data reasonably sample the monthly value outside of the shoulder seasons. Owing to the larger uncertainty during the shoulder seasons, in the revised manuscript we will restrict our evaluations with reference data to January-March. Figure 4 and all associated conclusions and discussion will be revised accordingly. Further, we propose to add the following text to the methods in Section 3.3.**

**"The reference observations do not account for snow-free periods because they are only conducted when there is snow. During the accumulation and ablations seasons, the monthly mean of available reference SWE will therefore often overestimate the true monthly mean value. For this reason, we restrict the comparisons of product SWE with reference SWE to January-March. Additionally, the infrequent sampling of the reference data (Fig. 2 lower left; see also Table 4 in Mortimer and Vionnet, 2025) means that, even when there is continuous snow cover, the monthly value calculated from the available dates with observations may not be representative of the true monthly mean. Investigation of the timing of the in-situ measurements within a month showed that, for the full**

domain, the timing of the observations is fairly well distributed across a month. However, this varies regionally and by network with some networks (e.g. Canada) biased towards the beginning of the month and others (e.g. Russia) biased towards the end of the month. We are unable to account for these biases in our analysis. The statistics calculated from comparisons with in situ data are not intended to be used as absolute performance measures. Rather, we are interested in the relative performance of the SL12 parameterization with the three different forcings and over time."

1. **Lack of snow-free reference observations**

Reference observations are only conducted when there is snow. When we calculated the monthly mean reference SWE we did not account for the snow free period during melt and onset. Thus, if the first (accumulation) or second (melt) half of a month is snow free, the mean reference SWE for that month will be an overestimate of the true mean.

To illustrate, below (Figure AR2) we show the distribution of the bias (reference minus product) for December, February, and May for the western US reference network. In December, there are no reference measurements prior to the 15th so the mean reference SWE calculated from these measurements is not representative of the monthly mean. During the middle of the winter (e.g. February) the sample distribution is concentrated at the beginning and end of the month, capturing the monthly mean (although slightly biased towards the end of the month), and there is no significant trend in the bias versus the day of the month. The problem illustrated in December is not evident in May (melt season) because there are sufficient sites with persistent snow cover. There may, however, be local issues for specific sites that lose snow earlier (since our treatment below lumps data across the western US).

[Figure]

**Figure AR2. Bias versus day of the month for reference sites in the NRCS network in the western US. Product bias for matching reference sites (blue dots) with x-axis location corresponds to the day of month of the reference observation and its trend versus the day of the month (black line). Mean and median bias for each day with reference observations in cyan and red, respectively (mean of the blue dots on each day of the month). Horizontal grey dotted line – mean product SWE calculated from the pool of data in the blue dots. For illustration purposes, only the ERA5 forcing is shown.**

2. **Sample distribution within a month**

If reference observations are not evenly distributed across the month this will introduce a bias in the monthly average reference SWE relative to the true monthly value. However, it is challenging to disentangle the timing of the observation from the landcover type and SWE magnitude because different networks, which often cover different snow classes and land cover types, have different sampling schedules. This error is not accounted for in our analysis.

Outside of the accumulation and melt seasons the data as a whole are fairly evenly distributed across a month. However, there are key regional differences because the sampling schedule varies by network (see Table 4 in Mortimer and Vionnet, 2025). Figure AR3, below, shows the number of reference observations from each network in our reference dataset over the full study period. Observations in Finland are centered around the middle of the month. In Russia, they tend to miss the first 5-10days and are biased towards the latter two thirds of the month. In Canada, observations are concentrated at the beginning and middle of the month with a secondary peak at the end of the month. This means the reference mean will tend to be biased towards the end of the month over Russia and the beginning of the month over Canada.

[Figure]

**Figure AR3. Number of reference observations by network and day of the month during 1980-2014 for the Jan-March period.**

Line 405: Is there truly no feedback in these offline runs? Land models often calculate 2-m air temperature prognostically which could impact SWE. If this is the case for the CLASSIC model, consider re-wording here.

**Thank you for noting this. In CLASSIC, 2-m air temperature is not calculated prognostically, but it affects the surface temperature, which may in turn affect SWE through snowmelt. We will reword the sentence when revising our manuscript.**

Lines 424-430: Adding more quantitative information here would be useful.

**Thanks for your suggestion, we will include more quantitative information when revising our manuscript.**

It looks like the simulations tend to underestimate SWE substantially; however, there is a tendency to overestimate winter SCF, largely in the control simulations and modestly in the SL12 simulations. If the SCF scheme is truly accurate at converting SWE or snow depth to SCF then we would expect to see underestimates in SCF. Can this point be added, particularly connecting logic between Sections 4.2 and 4.3?

**Thanks for raising this point. We think this "inconsistency" between SWE underestimation and SCF overestimation in the mountain regions is likely due to the following:**

During snow accumulation, SCF increases rapidly with snow depth in both the Control and SL12 schemes, as illustrated in the red (Control, only a rough approximation) and cyan (SL12) curves below. SCF reaches 100% in the Control and ~ 80% in the SL12 when snow depth is around 10cm. Snow is usually deep in the mountain regions (e.g. Fig.2d). Though SWE is underestimated in the model, SCF should have reached its maximum value during the peak SWE period (DJF).

[Figure]

Figure AR4. SCF parameterization for accumulation events. The x axis is snow depth in meters, and they axis is SCF. Colors indicate different values of parameter $k_{acc}$ from equation (1) (Fig. 7 from Swenson and Lawrence, 2012).

In addition, dense forests and steep terrain may obscure the MODIS sensor's view of snow-covered ground, resulting in underestimation (Hall et al., 2002; Marchane et al., 2015). The magnitude of winter SCF overestimation by SL12 in the mountain regions is relatively small (Fig. 5). The mean bias is 0.01, 0.02, and -0.02 for runs forced by CRUJRA, ERA5, and GSWP3-W5E5 (Table 2a), which is within the uncertainty range of the MODIS product. We will add the uncertainties of the MODIS product when revising our manuscript.

A. Marchane, L. Jarlan, L. Hanich, A. Boudhar, S. Gascoin, A. Tavernier, N. Filali, M. Le Page, O. Hagolle, B. Berjamy, Assessment of daily MODIS snow cover products to monitor snow cover dynamics over the Moroccan Atlas mountain range, Remote Sensing of Environment, Vol 160, 72-86, https://doi.org/10.1016/j.rse.2015.01.002, 2015.

Swenson, S. C. and Lawrence, D. M.: A new fractional snow-covered area parameterization for the Community Land Model and its effect on the surface energy balance, J. Geophys. Res.-Atmos., 117, D21107, https://doi.org/10.1029/2012JD018178, 2012.

It would be interesting to consider whether there are significant correlations between SCF biases with topographic complexity in each of the simulations, and in particular highlight if the SL12 scheme reduces or removes this relationship.

Thank you for your suggestion. To investigate whether there are significant correlations between SCF biases and topographic complexity, we made scatter plots (Fig. AR5) between SCF and standard deviation of sub-grid topography during the winter and spring seasons for each of the simulations. Below is an example of the plots for model runs forced by CRUJRA.

[Figure]

**Figure AR5, scatter plots between SCF and standard deviation of sub-grid topography during the winter and spring seasons for model runs forced by CRUJRA.**

**As expected, there are significant correlations between SCF biases with topographic complexity in all the simulations. The relationship is reduced by the SL12 scheme, especially in the spring. We will mention this in the main text and include a figure in the supplement when revising our manuscript.**

Section 4.4: It would be valuable to note whether the albedo biases are consistent with SCF biases (e.g., locations with SCF overestimates have albedo overestimates).

**Thank you for your suggestion. We will include the following figure (Fig. AR6) in the revised manuscript. Figure AR6 shows that surface albedo is overestimated by the control scheme, and the overestimation in the mountains is reduced by the SL12 scheme, consistent with the results for SCF.**

**Note the MODIS surface albedo product does not have shading correction, which tends to lead to underestimation in snow albedo in mountians (Bair et al., 2022). In flat regions, underestimation in MODIS SCF in vegetated regions likely contributed to the underestimation in surface albedo (details can be found in our reply to your main comments above), which at least partly explains the relatively large overestimation in the boreal forest regions.**

**Bair et al., 2022, https://doi.org/10.5194/tc-16-1765-2022.**

[Figure]

**Figure AR6. (a) Surface albedo (ALB) bias in a model run using the Control parameterization in the spring, (b) the difference in ALB between the model runs using the SL12 and CTL parameterizations. The surface albedo from MODIS is used as a reference.**

Section 5: here are some potentially useful references for land model SWE biases:

He, Cenlin, et al. "What causes the unobserved early-spring snowpack ablation in convection permitting WRF modeling over Utah Mountains?." Journal of Geophysical Research: Atmospheres 126.22 (2021): e2021JD035284.

Abolafia-Rosenzweig, Ronnie, et al. "Implementation and evaluation of a unified turbulence parameterization throughout the canopy and roughness sublayer in Noah-MP snow simulations." Journal of Advances in Modeling Earth Systems 13.11 (2021): e2021MS002665.

Chen, Fei, et al. "Modeling seasonal snowpack evolution in the complex terrain and forested Colorado Headwaters region: A model intercomparison study." Journal of Geophysical Research: Atmospheres 119.24 (2014): 13-795.

von Kaenel, Manon, and Steven A. Margulis. "Evaluation of Noah-MP snow simulation across site conditions in the western United States." Journal of Hydrometeorology 25.9 (2024): 1389-1406.

**Thanks for providing these helpful references. They will be included in the revised manuscript as appropriate.**

---

## Author Response (AR1)

Dear Dr. Hao,

We appreciate the thorough reviews provided by the three reviewers. We have revised the manuscript in light of their comments and recommendations. A copy of the reviewers' comments, followed by our detailed responses, is attached. Major changes to the manuscript include the following:

1. **Expanded Introduction:** We have expanded the Introduction to provide a more detailed overview of snow cover fraction (SCF) parameterizations (Lines 83–99).

2. **More information on MODIS SCF product:** more details about the MODIS SCF detection algorithm and accuracy have been added to Section 3.2 (Lines 316-324).

3. **Clarified Reference SWE Data Use:** Additional details on the in-situ reference snow water equivalent (SWE) data have been added to Section 3.3 (Lines 389–404). We also limited the evaluation period to January–March. Figure 4 and all associated conclusions and discussions have been revised accordingly.

4. **New Figure on Surface Albedo Biases:** We added a new figure (Figure 9 new) showing the spatial distribution of simulated surface albedo biases in Section 4.4 (Lines 639–648).

5. **SCF Bias–Topographic Complexity Analysis:** We investigated the relationship between SCF biases and topographic complexity (Lines 690–700 in Section 5) and included an illustrative figure of the results in the Supplement (Fig. A5).

6. **New Section on Observational and Methodological Uncertainties (Section 5.3):**

   o Discussed uncertainties associated with the observed SCF datasets used in this study (Lines 818–835);

   o Considered the potential impact of simulated snow density on modeled SCF (Lines 836–842);

   o Assessed the impact of elevation data resolution on sub-grid topography and simulated SCF (Lines 843–853).

We hope that these changes have satisfactorily addressed the comments by the reviewers.

Best Regards,

Libo Wang

Comments from Referee #1

**We thank Referee #1 for their helpful comments. Our replies to their comments are shown in bold below.**

Summary and overarching comments:

This study evaluates the impacts of an alternate SCF parameterization, that calculates SCF as a function of topographic complexity and SWE, in offline 1-degree CLASSIC model simulations. The alternate SCF parameterization tends to improve the accuracy of SCF simulated in CLASSIC during winter months in topographically complex regions. This study also explores the robustness of results to differing metrological forcing sources, which reveals the large impact of metrological forcing in snow simulation accuracy. This study provides a novel and important advancement for the CLASSIC modeling system that seems to allow land model simulations to better capture SCF, and in turn improve land-atmosphere interactions due to the snow-albedo feedback. Overall, the paper is well written and the study will warrant a publication after addressing the comments below.

**Thank you for your overall positive review of our manuscript.**

I have four overarching critiques for this analysis. (1) A key motivation for improving SCF in model simulations is to enhance simulated albedo. Although the study briefly covers the impacts of SCF on albedo accuracy using the AMBER score, it would be useful to go into more detail on the albedo analysis which is a critical component of this study. (2) MODIS SCF has questionable accuracy, particularly for representing ground SCF which the land model simulates. This point should be more directly addressed with the consideration of other data sources for SCF (e.g., STC-MODSCAG across the western US, see suggestion below). (3) Discrepancies in spatial resolution between reference data used to validate model simulations and the spatial resolution of the model simulations can largely impact results. Please see specific comment below addressing this point. (4) Figure quality should be improved throughout.

**Thank you for your suggestions.**

(1) A key motivation for improving SCF in model simulations is to enhance simulated albedo. Although the study briefly covers the impacts of SCF on albedo accuracy using the AMBER score, it would be useful to go into more detail on the albedo analysis which is a critical component of this study.

**We agree that it would be helpful to provide more details on the impact of the SL12 parameterization on simulated surface albedo in CLASSIC, especially considering the large impact of snow cover on surface albedo. We have included a figure (Fig. 9) comparing surface albedo simulated by the model runs using the Control and SL12 parameterizations (see figure and text below in our reply to your Specific recommendations).**

(2) MODIS SCF has questionable accuracy, particularly for representing ground SCF which the land model simulates. This point should be more directly addressed with the consideration of other data sources for SCF (e.g., STC-MODSCAG across the western US, see suggestion below).

**We agree that SCF derived from satellite optical sensors like MODIS is viewable snow cover from space during cloud-free overpasses (i.e. from above the canopy), while SCF from CLASSIC represents ground-level SCF (including snow cover beneath the canopy).**

Thank you for bringing our attention to the STC-MODSCAG data, which provides snow estimate on the ground and is better suitable for evaluating modelled SCF. However, it is currently only available for the western U.S., a global dataset is required to evaluate model performance in our study. Previous studies have shown that the accuracy of SCF from MODIS is lower than that from MODSCAG (Painter et al., 2009; Stillinger et al., 2023). Evaluating SCF from standard MODIS and STC-MODSCAG with high resolution airborne lidar data in parts of western U.S., Stillinger et al. (2023) showed that the median bias (RMSE) was -0.071 (0.127) for MODIS and -0.001 (0.120) for STC-MODSCAG across various snow climates. They also showed that the MODIS SCF product exhibited consistent negative bias of around 10% under intermediate canopy cover.

In addition, other evaluation studies suggested the accuracy of MODIS snow products was in the range of 88-93%, and dense forests and steep terrain may obscure the MODIS sensor's view of snow-covered ground, resulting in SCF underestimation (Hall et al., 2002; Hall and Riggs., 2021). However, as noted by Riggs et al. (2019), snow commission errors, often related to residual cloud contamination, are among the most common sources of error in MODIS snow products. As a result, the SCF derived from MODIS in this study may be subject to both underestimation and overestimation. After careful consideration, we have included more information about the algorithm and accuracy of the MODIS SCF product in Section 3.2 (Lines 316-324), and a discussion of the uncertainties associated with the observed SCF data used in this study in the newly added Section 5.3 of the revised manuscript (Lines 818-835).

Section 3.2 (Lines 316-324):

"The MODIS snow detection algorithm, which is based on the Normalized Difference Snow Index (NDSI), applies processing steps to alleviate snow detection commission errors and to flag uncertain snow detections (Hall et al., 2002). Due to spectral similarities between cloud and snow, cloud/snow confusion situations remain in MODIS version 6.1 snow products despite continued efforts in improving cloud masking and snow mapping algorithms (Riggs et al., 2019). Regardless of these inherent challenges, the NDSI-based snow detection technique has proven to be a robust indicator of snow presence under diverse situations, as demonstrated by numerous studies reporting accuracy statistics in the range of 88–93% (Riggs et al., 2019)."

Section 5.3 (Lines 818-835):

"SCF derived from satellite optical sensors such as MODIS represents the viewable snow cover from space during cloud-free overpasses (i.e., from above the canopy). Dense forests and steep terrain may obscure the MODIS sensor's view of snow-covered ground, leading to underestimation of SCF (Hall et al., 2002; Marchane et al., 2015). For example, Stillinger et al. (2023) found a consistent negative bias of approximately 10% under intermediate canopy cover when comparing MODIS SCF with high-resolution airborne lidar data in parts of the western U.S. The SCF overestimation in flat regions (Fig. 5d–f) may be partially attributable to this underestimation by MODIS. However, as noted by Riggs et al. (2019), snow commission errors, often related to residual cloud contamination, are among the most common sources of error in MODIS snow products. As a result, the SCF derived from MODIS in this study may be subject to both underestimation and overestimation.

While the IMS snow system primarily relies on visible satellite imagery, it also incorporates surface station observations and passive microwave data. Therefore, SCF derived from IMS is generally less affected by cloud cover and forest canopy than that from MODIS. Previous studies have shown that IMS tends to report higher SCF than MODIS (e.g., Brown et al., 2010), which is consistent with

**our results (Fig. 5). Nevertheless, SCF estimates from MODIS and IMS are largely consistent across all regions except the HMA, suggesting that our evaluation results are reasonably robust despite known uncertainties."**

**Brown, R., Derksen, C., and Wang, L: A multi-data set analysis of variability and change in Arctic spring snow cover extent, 1967–2008. J. Geophys. Res., 115, D16111, doi:10.1029/2010JD013975, 2010.**

**Hall, D.K., Riggs, G.A., Salomonson, V.V., DiGirolamo, N.E., & Bayr, K.J.: MODIS snow-cover products. Remote Sensing of Environment, 83(1–2), 181–194. http://dx.doi.org/10.1016/S0034-4257(02)00095-0, 2002.**

**Hall, D. K. & Riggs, G. A.: MODIS/Terra Snow Cover Monthly L3 Global 0.05Deg CMG. (MOD10CM, Version 61). Boulder, Colorado USA. NASA National Snow and Ice Data Center Distributed Active Archive Center. https://doi.org/10.5067/MODIS/MOD10CM.061. Date Accessed 06-19-2025, 2021.**

**Painter, T. H., Rittger, K., McKenzie, C., Slaughter, P., Davis, R. E., and Dozier, J.: Retrieval of subpixel snow covered area, grain size, and albedo from MODIS, Remote Sens. Environ., 113, 868–879, https://doi.org/10.1016/j.rse.2009.01.001, 2009.**

**Riggs, G.A., Hall, D.K. and Roman, M.O. 2019. MODIS Snow Products Collection 6.1 User Guide. NASA Goddard Space Flight Center, Greenbelt, MD, https://modis-snow-ice.gsfc.nasa.gov/uploads/snow_user_guide_C6.1_final_revised_april.pdf.**

**Stillinger, T., Rittger, K., Raleigh, M. S., Michell, A., Davis, R. E., and Bair, E. H.: Landsat, MODIS, and VIIRS snow cover mapping algorithm performance as validated by airborne lidar datasets, The Cryosphere, 17, 567–590, https://doi.org/10.5194/tc-17-567-2023, 2023.**

(3) Discrepancies in spatial resolution between reference data used to validate model simulations and the spatial resolution of the model simulations can largely impact results. Please see specific comment below addressing this point.

**To minimize these issues, we rely on snow courses and airborne gamma measurements because they are more spatially representative than single point measurements (Meromy et al. 2013). Snow courses consist of multiple measurements along a transect several hundreds of metres to kilometres in length that are averaged together to provide a single SWE value. Airborne gamma measurements are averaged across 300 m wide footprints and along 15–20 km long flight lines. In both cases, these measurements better sample the sub-grid-scale variability than a single-point measurement and so are more effective in capturing the larger-scale average. This decision does not fully close the scale difference between observations and gridded product, but it helps substantially.**

**In addition, analysis in Mortimer et al. (2024) showed that evaluation of gridded products with spatial resolutions ranging from 4km to 1.25° using this type of reference data yielded consistent performance ranking whether evaluated with airborne gamma or snow courses in non-mountain or mountain areas. This means we can make meaningful relative assessments of the gridded product performance.**

**In this manuscript, our intent is to provide readers with a sense of the relative simulated SWE errors driven by differences in the forcing data. We believe the reference data are appropriate for this purpose. However, at your suggestion, we also looked further into the sampling variability of**

the bias within a 1x1 degree CLASSIC grid cell. We identified a subset of grid cells containing multiple reference sites with long records. For simplicity, we restricted this demonstration to February. To remove issues related to sampling dates within a month, we only compared reference sites collected on the same date. As Figure AR1 shows, in nearly all cases, the ranking of SWE magnitudes for each of the three products and the reference SWE are similar from year to year. This demonstrates that the relative product errors assessed in the manuscript are likely to be consistent even if temporally sampled less frequently than demonstrated here. In nearly all cases the product SWE also falls outside of the standard deviation of the reference SWE. This demonstrates that the calculated biases presented in the manuscript are likely to be meaningful, even if spatially sampled less frequently than demonstrated here (however in most cases our arguments rely only on the relative product bias anyway). In rare cases (e.g. red box), the choice of reference site will alter the sign of the bias (but still does not alter the relative sense of bias among the three products).

[Figure]

**Figure AR1. Mean and standard deviation of reference SWE (blue) for sites measured on the same date within the same model grid cell. When there were multiple dates in the same month, the mean and standard deviation were calculated for sites measured on the same date and then averaged across the month-year. Dots: modelled SWE for the corresponding grid cell and month. Far-right black line, triangle, and hollow circles show the mean across the time series. For display, only February is shown. Each plot corresponds to one dot on the map (lon/lat listed at top of each plot and colors of plot axes correspond to dots colors on map). Sites in the western U.S. are mountainous, all other are in flat regions as defined in the manuscript.**

Meromy, L., Molotch, N. P., Link, T. E., Fassnacht, S. R., and Rice, R.: Subgrid variability of snow water equivalent at operational snow stations in the western USA, Hydrol. Process., 27, 2383–2400, https://doi.org/10.1002/hyp.9355, 2013.

Mortimer, C., Mudryk, L., Cho, E., Derksen, C., Brady, M., and Vuyovich, C.: Use of multiple reference data sources to cross-validate gridded snow water equivalent products over North America, The Cryosphere, 18, 5619–5639, https://doi.org/10.5194/tc-18-5619-2024, 2024.

(4) Figure quality should be improved throughout.

**We apologize for the poor quality of the figures. The quality was fine in the original Microsoft Word version of the manuscript but deteriorated after converting to the pdf file. We will double-check to ensure that all figures are of high quality in the revised manuscript.**

Specific recommendations:

Paragraph starting in line 81: Note that some land surface models also consider SCF as a function of snow density and land cover classification (e.g., He et al., 2023). He, C., et al. The community Noah-MP land surface modeling system technical description version 5.0. NCAR Technical Note NCAR/TN-575+ STR, doi: 10.5065/ew8g-yr95, 2023.

**Thank you for your suggestion. We have expanded the paragraph starting in line 81 and included a more detailed summary of SCF parameterizations as in the following (L83-99):**

**"Some early SCF parameterizations assumed a linear increase in snow cover with snow depth or snow water equivalent (SWE), reaching 100% SCF once a specified threshold was met (e.g., Verseghy, 1991; Bonan, 1996). Other approaches incorporated surface roughness length into the SCF–SND (or SWE) relationships (e.g., Dickinson et al., 1986; Marshall and Oglesby, 1994), and distinguished SCF estimates between bare ground and vegetated areas (Douville et al., 1995; Yang et al., 1997). Large uncertainties in modeled SCF from these early schemes motivated efforts to refine parameterizations by accounting for terrain heterogeneity or incorporating sub-grid snow distribution (Roesch et al., 2001; Liston, 2004). More recent SCF parameterizations have included snow density (e.g., Niu and Yang, 2007; Lalande et al., 2023) and land cover type (e.g., He et al., 2023), with some schemes adopting separate formulations for snow accumulation and melt periods (Swenson and Lawrence, 2012). Some of these parameterizations account for sub-grid topographic variability (e.g., Douville et al., 1995; Roesch et al., 2001; Swenson and Lawrence, 2012; Lalande et al., 2023), which has been shown to be crucial for accurate SCF simulation in mountainous regions (Miao et al., 2022)."**

Section 2.1: please articulate the capacities in which CLASSIC is used, for either research applications or operational modeling.

**We agree that it would be nice to include a couple of sentences about the applications of CLASSIC. We have included the following in Section 2.1 (L140-147):**

**"The CLASSIC model simulations can be performed at point, regional, and global scales both in coupled and offline modes. CLASSIC has been applied in an offline context, i.e. forced with observed meteorology (e.g. Bailey et al., 2000; Bartlett et al., 2006; Melton et al., 2019), as the**

**physical land surface component of regional climate models, e.g. CRCM (Wang et al., 2014; Ganji et al., 2015) and CanRCM (Scinocca et al., 2016), and integrated into each version of the Canadian Atmospheric Model (CanAM; von Salzen et al., 2013), and Earth System Model (CanESM; Arora et al., 2011; Swart et al., 2019) since the early 1990s."**

Section 3.1 and Figure 1: Please add information on the calculation of topographic standard deviation. Specifically, what is the resolution of the elevation product which is used to calculate this metric?

**This was already provided in the manuscript: "Classification of mountain and flat regions is based on standard deviation of the sub-grid terrain from the ETOPO1 elevation data at 1 arc-minute resolution (NOAA, 2009)."**

Section 3.2: Another potential issue with MODIS SCF is not just its accuracy, but also whether its retrieval represents pixel scale SCF or just the ground SCF. Many land models simulate ground SCF, rather than total pixel SCF (e.g., including vegetated fractions of the pixel) and thus a comparison with the MODIS data used here may not be appropriate. The STC-MODSCAG data addresses this issue, and the latest version has available data across the mountainous western US (https://nsidc.org/data/stc_modscgdrf_hist/versions/1#anchor-data-access-tools).

Please consider using these data as an additional reference to evaluate whether the comparisons against MODIS are reliable.

**Thank you for your suggestion. Stillinger et al. (2023) evaluated SCF from standard MODIS and STC-MODSCAG with high resolution airborne lidar data in parts of western U.S. They showed that the median bias (RMSE) was -0.071 (0.127) for MODIS and -0.001 (0.120) for STC-MODSCAG across various snow climates. Their analysis also revealed that the MODIS SCF product exhibited consistent negative bias of approximately 10% under intermediate canopy cover. These findings are consistent with results from earlier studies indicating that MODIS tends to underestimate SCF in forested regions (e.g. Hall et al., 2002).**

**Based on these, we have added a discussion of the uncertainties associated with the observed SCF data used in this study in Section 5.3 of the revised manuscript (Lines 818-835). Additional details are provided in our response above to your main comments.**

Lines 315-316: Simulated snow density is also a source of SCF uncertainty, e.g., Abolafia-Rosenzweig et al. (2024), which could be noted here or in the Discussion. Abolafia-Rosenzweig, Ronnie, et al. "Evaluating and enhancing snow compaction process in the Noah-MP land surface model." Journal of Advances in Modeling Earth Systems 16.2 (2024): e2023MS003869.

**Thank you for your suggestion and for providing the reference. We have incorporated this point in the newly added Section 5.3 in the revised manuscript (Lines 836-842):**

**"In LSMs, snow depth is typically diagnosed from SWE and snow density. As a result, uncertainties in modeled snow density can propagate to uncertainties in SCF, particularly when the SCF parameterization depends on snow density and/or snow depth, as demonstrated by Abolafia-Rosenzweig et al. (2024). In CLASSIC, these uncertainties influence SCF simulated by the control parameterization but do not directly affect SCF in the SL12 parameterization (Section 2.2). Since our focus is on the SL12 parameterization in this study, we do not explore this issue further."**

Section 3.3: These SWE evaluations are likely largely impacted by discrepancies between observed and modelled spatial resolutions. It would be good to emphasize this point further, even in the case of airborne gamma SWE observations. To consider the spatial representativeness of observations, consider comparing time series from in-situ stations contained by the same 1 degree pixel and consider whether there are large discrepancies (e.g., with bias and correlation metrics).

**Please see our reply above with Figure AR1. Most of our conclusions about forcing-driven errors are based on the assessed bias.**

Also, when observations are measured infrequently (e.g., a few times in a month) are the modelled data screened temporally to match the observational frequency prior to comparison?

**The model output was only saved at monthly frequency. How well our date-specific samples will represent a true monthly mean will depend on their distribution over the month of interest. We examine two aspects of this in detail below: lack of snow-free reference measurements and the distribution of measurements within a month.**

**Despite the challenges highlighted below, we are confident that for the application used in our study, the data reasonably sample the monthly value outside of the shoulder seasons. Owing to the larger uncertainty during the shoulder seasons, in the revised manuscript we have restricted our evaluations with reference data to January-March. Figure 4 and all associated conclusions and discussion have been revised accordingly. Further, we have added the following text to the methods in Section 3.3 (Lines 389-404).**

**"The reference SWE observations do not account for snow-free periods because they are only conducted when there is snow. During the accumulation and ablation seasons, the monthly mean of available reference SWE will therefore often overestimate the true monthly mean value. For this reason, we restrict the comparisons of product SWE with reference SWE to January-March. Additionally, the infrequent sampling of the reference data (Fig. 2c; see also Table 4 in Mortimer and Vionnet, 2025) means that, even when there is continuous snow cover, the monthly value calculated from the available dates with observations may not be representative of the true monthly mean. Investigation of the timing of the in-situ measurements within a month showed that, for the full domain, the timing of the observations is fairly well distributed across a month. However, this varies regionally and by network with some networks (e.g. Canada) biased towards the first half of the month and others (e.g. Russia) slightly biased towards the latter two thirds of the month (Fig. A1). We are unable to account for these biases in our analysis. The statistics calculated from comparisons with in-situ data are not intended to be used as absolute performance measures. Rather, we are interested in assessing how the relative performance of CLASSIC SWE varies under the three choices of forcings; as Mortimer et al. (2024) demonstrates, the reference data is well able to discern relative performance of SWE products."**

**1. Lack of snow-free reference observations**

**Reference observations are only conducted when there is snow. When we calculated the monthly mean reference SWE we did not account for the snow free period during melt and onset. Thus, if the first (accumulation) or second (melt) half of a month is snow free, the mean reference SWE for that month will be an overestimate of the true mean.**

To illustrate, below (Figure AR2) we show the distribution of the bias (reference minus product) for December, February, and May for the western US reference network. In December, there are no reference measurements prior to the 15th so the mean reference SWE calculated from these measurements is not representative of the monthly mean. During the middle of the winter (e.g. February) the sample distribution is concentrated at the beginning and end of the month, capturing the monthly mean (although slightly biased towards the end of the month), and there is no significant trend in the bias versus the day of the month. The problem illustrated in December is not evident in May (melt season) because there are sufficient sites with persistent snow cover. There may, however, be local issues for specific sites that lose snow earlier (since our treatment below lumps data across the western US).

[Figure]

Figure AR2. Bias versus day of the month for reference sites in the NRCS network in the western US. Product bias for matching reference sites (blue dots) with x-axis location corresponds to the day of month of the reference observation and its trend versus the day of the month (black line). Mean and median bias for each day with reference observations in cyan and red, respectively (mean of the blue dots on each day of the month). Horizontal grey dotted line – mean product SWE calculated from the pool of data in the blue dots. For illustration purposes, only the ERA5 forcing is shown.

2. **Sample distribution within a month**

If reference observations are not evenly distributed across the month this will introduce a bias in the monthly average reference SWE relative to the true monthly value. However, it is challenging to disentangle the timing of the observation from the landcover type and SWE magnitude because different networks, which often cover different snow classes and land cover types, have different sampling schedules. This error is not accounted for in our analysis.

Outside of the accumulation and melt seasons the data as a whole are fairly evenly distributed across a month. However, there are key regional differences because the sampling schedule varies by network (see Table 4 in Mortimer and Vionnet, 2025). Figure AR3, below, shows the number of reference observations from each network in our reference dataset over the full study period. Observations in Finland are centered around the middle of the month. In Russia, they tend to miss the first 5-10days and are biased towards the latter two thirds of the month. In Canada, observations are concentrated at the beginning and middle of the month with a secondary peak at the end of the month. This means the reference mean will tend to be slightly biased towards the latter two thirds of the month over Russia and the first half of the month over Canada.

[Figure]

**Figure AR3. Number of reference observations by network and day of the month during 1980-2014 for the Jan-March period.**

Line 405: Is there truly no feedback in these offline runs? Land models often calculate 2-m air temperature prognostically which could impact SWE. If this is the case for the CLASSIC model, consider re-wording here.

**Thank you for noting this. In CLASSIC, 2m air temperature is not calculated prognostically, but it affects the surface temperature, which may in turn affect SWE through snowmelt. We have reworded the sentence to the following (now Line 470):**

**"… and there is limited feedback in offline runs."**

Lines 424-430: Adding more quantitative information here would be useful.

**Thanks for your suggestion, we have modified the text to include more quantitative information as in the following (Lines 475-493):**

**"In the mountainous regions, the biases are similar for GSWP3W5-SL12 (-129.4) and CRUJRA-SL12 (-136.6) and the lowest for ERA5-SL12 (-90.8). In flat regions, GSWP3W5-SL12 (50.3) has more than twice the SWE bias seen in either CRUJRA-SL12 (15.0) or ERA5-SL12 (17.5), which is mainly due to SWE overestimation in eastern NA and northern Europe (Fig. A3)."**

It looks like the simulations tend to underestimate SWE substantially; however, there is a tendency to overestimate winter SCF, largely in the control simulations and modestly in the SL12 simulations. If the

SCF scheme is truly accurate at converting SWE or snow depth to SCF then we would expect to see underestimates in SCF. Can this point be added, particularly connecting logic between Sections 4.2 and 4.3?

**Thanks for raising this point. We think this "inconsistency" between SWE underestimation and SCF overestimation in the mountain regions is likely due to the following:**

**During snow accumulation, SCF increases rapidly with snow depth in both the Control and SL12 schemes, as illustrated in the red (Control, only a rough approximation) and cyan (SL12) curves below. SCF reaches 100% in the Control and ~ 80% in the SL12 when snow depth is around 10cm. Snow is usually deep in the mountain regions (e.g. Fig.2d). Though SWE is underestimated in the model, SCF should have reached its maximum value during the peak SWE period (DJF).**

[Figure]

**Figure AR4. SCF parameterization for accumulation events. The x axis is snow depth in meters, and they axis is SCF. Colors indicate different values of parameter $k_{acc}$ from equation (1) (Fig. 7 from Swenson and Lawrence, 2012).**

**In addition, dense forests and steep terrain may obscure the MODIS sensor's view of snow-covered ground, resulting in underestimation (Hall et al., 2002; Marchane et al., 2015). The magnitude of winter SCF overestimation by SL12 in the mountain regions is relatively small (Fig. 5). The mean bias is 0.01, 0.02, and -0.02 for runs forced by CRUJRA, ERA5, and GSWP3-W5E5 (Table 2a), which is likely within the uncertainty range of the MODIS product. We have included a discussion on the uncertainties of the MODIS product in the revised manuscript (Lines 824 – 841).**

**A. Marchane, L. Jarlan, L. Hanich, A. Boudhar, S. Gascoin, A. Tavernier, N. Filali, M. Le Page, O. Hagolle, B. Berjamy, Assessment of daily MODIS snow cover products to monitor snow cover dynamics over the Moroccan Atlas mountain range, Remote Sensing of Environment, Vol 160, 72-86, https://doi.org/10.1016/j.rse.2015.01.002, 2015.**

**Swenson, S. C. and Lawrence, D. M.: A new fractional snow-covered area parameterization for the Community Land Model and its effect on the surface energy balance, J. Geophys. Res.-Atmos., 117, D21107, https://doi.org/10.1029/2012JD018178, 2012.**

It would be interesting to consider whether there are significant correlations between SCF biases with topographic complexity in each of the simulations, and in particular highlight if the SL12 scheme reduces or removes this relationship.

**Thank you for the suggestion. To investigate whether significant correlations exist between SCF biases and topographic complexity, we have generated scatter plots and examined the correlations between SCF bias and the standard deviation of sub-grid topography during the winter and spring seasons for each simulation. An example of these plots, based on model runs forced by CRUJRA, has been included in the Supplement (Fig. A5) and shown below. The following explanatory text has been added to Section 5 (Lines 690–700).**

**"Biases in modeled SCF vary between flat and mountainous regions for both SCF parameterizations (Table 2, Fig. 5, and Fig. 6). Previous studies have highlighted the importance of accounting for sub-grid topography on SCF simulations in mountainous regions (Swenson and Lawrence, 2012; Miao et al., 2022). Are the modelled SCF biases related to topographic complexity in this study? To explore this, we generated scatter plots and examined the correlations between SCF biases and the standard deviation of sub-grid topography during the winter and spring seasons for each simulation. As expected, significant correlations were found in all simulations, indicating that SCF biases tend to increase with increasing topographic complexity. However, this relationship is notably reduced under the SL12 scheme, particularly in spring. An example of these scatter plots, based on model runs forced by CRUJRA, is presented in Figure A5."**

[Figure]

**Figure AR5. Scatter plots showing the relationship between SCF bias and the standard deviation of sub-grid topography during the winter (left) and spring (right) seasons for model runs using the CTL (top) and SL12 (Bottom) schemes forced by CRUJRA. The correlation coefficient (r) and p-value (using a two-tailed t-test) are provided in the upper-left corner of each plot.**

Section 4.4: It would be valuable to note whether the albedo biases are consistent with SCF biases (e.g., locations with SCF overestimates have albedo overestimates).

**Thank you for your suggestion. We have included a figure (Fig. AR6) in the revised manuscript (Figure 9 new) that shows the surface albedo biases and the differences between model runs using the SL12 and CTL parameterizations. The following explanatory text has been added to Section 4.4 (Lines 639–648). Note that the MODIS surface albedo product does not have shading correction, which tends to lead to underestimation in snow albedo in mountainous areas (Bair et al., 2022). The underestimation of MODIS SCF in dense forest regions likely leads to underestimation in its surface albedo (details can be found in our reply to your main comments above). These limitations may have contributed, at least in part, to the albedo overestimation shown in Fig. AR6.**

**Bair et al., 2022, https://doi.org/10.5194/tc-16-1765-2022.**

**"For surface albedo, the relatively large differences are observed in the spatial distribution score (Sdist), suggesting better characterization of the spatial patterns when using the SL12 parameterization (Fig. 9). Figure 9 shows that surface albedo is generally overestimated by the control scheme (Fig. 9a), with this overestimation notably reduced in the mountainous regions when the SL12 scheme is applied (Fig. 9b), consistent with the improvements seen in SCF. Previous studies have indicated that the MODIS surface albedo product may exhibit biases due to the absence of shading corrections in the mountainous areas and underestimation of snow cover in dense forest regions (Hall et al., 2002; Bair et al., 2022). These limitations may have contributed, at least in part, to the albedo overestimation shown in Figure 9a."**

[Figure]

**Figure AR6. (a) Surface albedo (ALB) bias (relative to observed from MODIS) in a model run forced by ERA5 using the Control parameterization in the spring, (b) the difference in ALB between the model runs using the SL12 and CTL parameterizations, with red colours indicating lower albedo simulated by the SL12 parameterization.**

Section 5: here are some potentially useful references for land model SWE biases:

He, Cenlin, et al. "What causes the unobserved early-spring snowpack ablation in convection permitting WRF modeling over Utah Mountains?." Journal of Geophysical Research: Atmospheres 126.22 (2021): e2021JD035284.

Abolafia-Rosenzweig, Ronnie, et al. "Implementation and evaluation of a unified turbulence parameterization throughout the canopy and roughness sublayer in Noah-MP snow simulations." Journal of Advances in Modeling Earth Systems 13.11 (2021): e2021MS002665.

Chen, Fei, et al. "Modeling seasonal snowpack evolution in the complex terrain and forested Colorado Headwaters region: A model intercomparison study." Journal of Geophysical Research: Atmospheres 119.24 (2014): 13-795.

von Kaenel, Manon, and Steven A. Margulis. "Evaluation of Noah-MP snow simulation across site conditions in the western United States." Journal of Hydrometeorology 25.9 (2024): 1389-1406.

**Thank you for providing these references, which will be helpful for improving SWE simulation in CLASSIC in future studies.**

Comments from Referee #2

**We thank Referee #2 for their helpful comments. Our replies to their comments are shown in bold below.**

Review of Wang et al., "Impact of topography and meteorological forcing on snow simulation in the Canadian Land Surface Scheme Including Biogeochemical Cycles (CLASSIC)"

This study examines the impact of replacing the default CLASSIC snow cover fraction (SCF) parameterization with an alternative one. The study has two objectives: 1) comparing the two SCF parameterizations and 2) examining the role of meteorological forcing on the simulation of snow. The role of meteorological forcing is highlighted by using three input datasets to force the model. The default parameterization predicts SCF from snow depth using a linear relationship for snow depth below 0.1m; above 0.1m, SCF is 1. The alternative considers the accumulation and ablation seasons separately, and also incorporates topographic information to adjust its behavior spatially based on the topographic variability within a region. The authors find that comparisons to MODIS SCF are more favorable when using the alternate SCF parameterization. In addition, other metrics related to water and energy fluxes show improvement.

General comments:

The authors state that tests changing the lone parameter (0.1) in the CTL parameterization show minimal impacts to their simulations. In contrast, in addition to a structural change, SL12 offers opportunities to improve the SCF simulation by modifying and calibrating equations 1 and 3. While the authors mention a couple of changes to the k_acc and N_melt parameters that led to positive results, they choose not to explore the parameter sensitivity in more detail "because none of the three meteorological forcing datasets used in this study are exempt from biases, there is a limit to how well optimal parameter values can be chosen for use in CLASSIC". The results shown in figure 5 of the manuscript seem to contradict this statement, as the biases seem consistent across forcings. The authors support this by stating "On the global scale, the spatial patterns of SCF bias are similar for all three meteorological forcing choices."

**Thank you for your overall positive review of our manuscript.**

I believe that this study would be improved if the authors were to pursue this path. The authors could perform a few shorter, initialized runs (e.g. from 1980 onwards) to do a sensitivity study of the N_melt parameter. They could then use these results to see if a better function for N_melt as a function of sigma_topo becomes apparent. For example, figure 5 indicates that lower SCF values in winter and spring, irrespective of forcing input, are preferred for flat regions. This implies that the N_melt equation increase too rapidly for small values of sigma_topo. This is perhaps not surprising, given that 1/x blows up as x goes to zero. A bounded function, e.g. a decaying exponential, might improve the results for flat

regions, while maintaining the good results for mountainous regions. Similarly, adding a simple dependence on sigma_topo to k_acc might improve the fall bias shown in figure 5 b) without degrading other regions.

**Thank you for your suggestions. We agree that it would be ideal if the SL12 parameterization could be calibrated to improve modelled SCF in both the mountain and flat regions. Initially we tried many sensitivity experiments to try to achieve this goal. The discussion section of the paper provides some generalized results of the experiments and how this process worked (Lines 792-809). However, after gaining a better understanding of the uncertainties in the meteorological forcings and observed SCF datasets, we think it is a goal impossible to achieve at present. Below are the reasons:**

(1) **Evaluation based on measurements from snow course and airborne gamma data showed that for all three choices of forcing data, modelled SWE is underestimated in the mountain and overestimated in the flat regions during the snow season (Fig. 4). Since SCF is directly linked to SWE in the SL12 scheme (see Eq.1 and Eq. 2), these SWE biases can exert a large impact on simulated SCF in the fall and spring seasons in the model (limited impact during the peak SWE period for SCF is usually saturated, details can be found in our reply to RC1). The consistent SCF biases shown in Figure 5 are linked to these consistent SWE biases for all three forcing choices in the model.**

(2) **SCF derived from satellite optical sensors such as MODIS represents the visible snow cover from space during cloud-free overpasses - that is, from above the vegetation canopy. In contrast, SCF from the CLASSIC model represents ground-level SCF, including snow cover beneath the canopy. As a result, MODIS-derived SCF tends to be biased low in forested regions. This limitation has been noted in previous studies (Hall et al., 2002; Hall and Riggs., 2021). Using high-resolution airborne lidar data from the western U.S., Stillinger et al. (2023) found that the MODIS SCF product exhibited a consistent negative bias of approximately 10% under intermediate canopy cover. Thus, it may not be ideal to over-tune the model to a specific observational estimate which may still have uncertainties.**

(3) **Even if the SL12 parameterization and its associated parameters are perfectly specified, and the reference SCF for comparison was a perfect measure of ground truth, biases in the modelled SCF will still arise due to inaccuracies in the meteorological forcing data. We may revisit this if more realistic meteorological forcing datasets are available in the future.**

**We discussed the impact of meteorological forcing datasets on modelled SWE in Section 5.1. We have included the following text in Section 5.1 to clearly state the link between biases in SWE and SCF (Lines 711-715).**

**"Since SCF is directly linked to SWE in the SL12 scheme (see Eq.1 and Eq. 2), these SWE biases can exert a large impact on simulated SCF in the fall and spring seasons in the model (limited impact during the peak SWE period because SCF is usually saturated). The consistent SCF biases shown in Figure 5 are linked to these consistent SWE biases for all three forcing choices in the model."**

**We have included a discussion on the uncertainties of the observed SCF products used in this study in Section 5.3 (Lines 818-835) in the revised manuscript (details can be found in our reply to RC1).**

About your other comments: "This is perhaps not surprising, given that 1/x blows up as x goes to zero… ". Note the denominator of Eq. (3) is max(10, $\sigma_{topo}$), so the maximum $N_{melt}$ parameter is 20. Attempting to develop a different SCF parametrization is beyond the scope of the study.

Hall, D.K., Riggs, G.A., Salomonson, V.V., DiGirolamo, N.E., & Bayr, K.J.: MODIS snow-cover products. Remote Sensing of Environment, 83(1–2), 181–194. http://dx.doi.org/10.1016/S0034-4257(02)00095-0, 2002.

Hall, D. K. & Riggs, G. A.: MODIS/Terra Snow Cover Monthly L3 Global 0.05Deg CMG. (MOD10CM, Version 61). Boulder, Colorado USA. NASA National Snow and Ice Data Center Distributed Active Archive Center. https://doi.org/10.5067/MODIS/MOD10CM.061. Date Accessed 06-19-2025, 2021.

Stillinger, T., Rittger, K., Raleigh, M. S., Michell, A., Davis, R. E., and Bair, E. H.: Landsat, MODIS, and VIIRS snow cover mapping algorithm performance as validated by airborne lidar datasets, The Cryosphere, 17, 567–590, https://doi.org/10.5194/tc-17-567-2023, 2023.

Specific comments:

Lines 99,100: add references for CLM5, CESM2

Thanks for noting this, we have added references for CLM5 and CESM2 in the revised manuscript (Lines 111-114).

"The SL12 parameterization was implemented in the Community Land Model version 5 (CLM5, Lawrence et al., 2019), the land surface component in the Community Earth System Model version 2 (CESM2, Danabasoglu et al., 2020)."

Line 130: how do the four sub-areas relate to SCF? Do the snow/snow-free areas change dynamically?

The areal fractions of the four subareas - vegetated , bare soil, vegetated with snow cover, and snow cover over bare soil, are calculated based on the fractional coverage of the vegetation categories and SCF.

Yes, the snow-covered and snow-free areas change dynamically at each time step.

Line 144: "all vertical layers": I thought there was only 1 layer (line 132)?

We meant for soil layers beneath the snow layer as well. We have modified the sentence to the following (now Line 165):

"In CLASSIC, the thicknesses of all layers (snow and soil) are recommended to be greater than 0.1 m to avoid numerical instability problems."

Line 160: is there reason to think k_acc should vary spatially?  If so, how might one parameterize it?  (discussed in sec 5.2).  Also, SL12 mentions that eq 1 assumes snowfall is randomly distributed in the region; is this a valid assumption?

**The SL12 parameterization was developed based on the relationship between snow depth from Snow Data Assimilation System (SNODAS) and SCF from MODIS over the continental U.S. (Swenson and Lawrence, 2012). Topographic dependency between SCF-SND was not observed during the accumulation period. More details can be found in Swenson and Lawrence (2012).**

**With regard to your comment "SL12 mentions that eq 1 assumes snowfall is randomly distributed in the region; is this a valid assumption?",  this assumption may not be valid in mountain regions where snowfall affects preferentially high-elevation areas. We have included the following text in the revised manuscript (Lines 185-188):**

**"Eq. (1) assumes that precipitation is randomly distributed across the region, which may be questionable in mountainous areas where snowfall tends to preferentially accumulate at higher elevations. Nevertheless, SCF simulated using the SL12 parameterization from coarse-resolution climate models shows reasonable agreement with observations (e.g. Lalande et al., 2023)."**

**Swenson, S. C. and Lawrence, D. M.: A new fractional snow-covered area parameterization for the Community Land Model and its effect on the surface energy balance, J. Geophys. Res.-Atmos., 117, D21107, https://doi.org/10.1029/2012JD018178, 2012.**

Line 173: how are the parameters 200 and 10 chosen?  How sensitive are the results to these parameters, and could they instead be calibrated?

**The parameters were determined based on observed relationship between SCF-SND, details can be found in Swenson and Lawrence (2012). We performed sensitivity experiments where 200 in the numerator of Eq. (3) were changed to 50, 100, and 300. Some examples of the results are discussed in Section 5.2 and another figure is included below in response to your comment on bias in flat regions.**

Line 181: does SL12 implemented in CLM5 use time of year to determine which equation to use?
**No. It is also based on whether SWE is increasing or decreasing with respect to the previous time step.**

Why is equation 4 used?  Isn't W_max based on the evolution of W in the model, i.e. is it the peak SWE of each snow season?

**W_max is the accumulated maximum SWE at each time step, which is different from the peak SWE of each snow season.**

Line 262: does the resolution of the DEM affect the calculation of the standard deviation of the sub-grid terrain?

To assess the impact of DEM resolution, we compared $\sigma_{topo}$ derived from two DEM datasets: ETOPO1 (1-arc-minute resolution, used in our study) and ETOPO2022 (15-arc-second resolution). The results show that the differences are limited in extent, primarily concentrated along the edges of mountain ranges. We also performed a test simulation using $\sigma_{topo}$ derived from ETOPO2022 and compared modelled SCF with that from a run using $\sigma_{topo}$ derived from ETOPO1. The maximum difference was less than 5%. These suggest that the resolution of the DEM data has limited impact on the calculation of sub-grid topographic variability and the simulated SCF.

We have included the following text in Section 5.3 (Lines 843-853):

"Additional uncertainties may arise from the elevation data used to compute the standard deviation of sub-grid topography ($\sigma_{topo}$), particularly related to its spatial resolution. We compared $\sigma_{topo}$ derived from two elevation datasets: ETOPO1 (1-arc-minute resolution, used in this study) and ETOPO2022 (15-arc-second resolution). The results indicate that the differences are limited in spatial extent and are primarily concentrated along edges of mountain ranges. To assess the impact on model results, we conducted a test simulation using $\sigma_{topo}$ derived from ETOPO2022 and compared the simulated SCF with that from a run based on $\sigma_{topo}$ derived from ETOPO1. The maximum difference in SCF between the two runs was less than 5% (not shown). These findings suggest that the resolution of the elevation data has a limited effect on the calculation of sub-grid topographic variability and simulated SCF, consistent with sensitivity tests reported by Lalande et al. (2023)."

Line 271: 'high mountainous asia' or 'high mountain asia'?

Thanks for noting this typo, fixed.

Line 297: IMS data could be converted to 1 degree fractional values, then treated similarly to MODIS; is that how IMS data is processed?

The IMS dataset provides binary snow/no snow information: if more than 50% of the 4 km pixel is covered by snow, it has a value of 1, otherwise 0 (snow free). If we aggregate the 4km IMS snow/no snow data into SCF at 1 degree, the derived SCF would have an uncertainty range from 50% to 100%.

In our study, daily IMS data were converted to monthly snow cover duration fraction (SCF = total number of days with snow cover in a month divided by the number of days in the month), which we found more comparable with SCF from MODIS. This method was used in previous studies (Brown et al., 2010; Wang et al., 2014).

Brown et al. 2010, https://doi:10.1029/2010JD013975.

Wang et al., 2014, https://doi.org/10.1175/JHM-D-13-086.1.

We have expanded the paragraph to provide more details about the IMS data (Lines 325-338):

"To mitigate the uncertainties in the MODIS product due to frequent cloud cover and/or complex terrains, SCF from the Interactive Multisensor Snow and Ice Mapping System (IMS) produced by the U.S. National Ice Center (2008) was also used as a reference in our analysis. The IMS snow

cover analysis system consists of an interactive workstation for snow cover mapping by a snow analyst (Ramsay,1998; Helfrich et al., 2007). It relies mainly on visible satellite imagery (including MODIS data) but is augmented by station observations and passive microwave data. The IMS dataset consists of binary snow/no snow information on a 4 km resolution polar stereographic projection grid (Helfrich et al. 2007). Though the binary format of this dataset is not ideal for SCF estimation, especially in areas around the snow line, SCF estimates from IMS are included because the resolution of our model is coarse (1º) and IMS data has been used to evaluate modelled SCF in previous studies (e.g. Wang et al., 2014; Orsolini et al., 2019). Daily IMS data were converted to monthly snow cover duration fraction (SCF = total number of days with snow cover in a month divided by the number of days in the month) following the method in Brown et al. (2010)."

Line 315: SND is related to SWE via snow density; how is snow density calculated in CLASSIC?

**The density of fresh snow density ($\rho_{s,i}$) is determined as an empirical function of the air temperature (Ta). For Ta <= 0ºC, an equation presented by Hedstrom and Pomeroy (1998) is used. For Ta > 0ºC, a relation following Pomeroy and Gray (1995) is used, with an upper limit of 200 kg m$^{-3}$:**

$$\rho_{s,i} = 67.92 + 51.25 \exp[Ta/2.59] \quad Ta < 0ºC$$
$$\rho_{s,i} = 119.17 + 20.0Ta \quad\quad\quad\quad Ta \geq 0ºC$$

**Over time, snowpack density ($\rho_s$) increases due to the effects of crystal settlement and metamorphism in the snowpack, sublimation, wind packing, melt, and refreezing. CLASSIC models this using an empirical relationship with time:**

$$\rho_s (t) = \rho_s (t-1) \exp(-B\Delta t)$$

**Where $\Delta t$ is the time step and B = 0.01/3600 is a constant. More details can be found in Verseghy (1991) and Bartlett et al. (2006), which are both referenced in our manuscript.**

**Verseghy, D. L., 1991, https://doi.org/10.1002/joc.3370110202.**
**Bartlett et al., 2006, https://doi:10.3137/ao.440301.**

Line 363: are there 21 datasets, or 7 datasets?

**A unique feature of AMBER is that it uses multiple reference datasets when available to evaluate the same variable. In the AMBER results shown in our study, seven variables were evaluated using 21 reference datasets (shown in Table 1).**

Figure 2: add units to (d) colorbar

**Thank you for noting this, we have added units to the colorbars in Figure 2.**

Figure 3: please label figures with NH or HMA. Also adding a dashed line to indicate zero for each y-axis would be helpful.

**Thank you for your suggestions. We have modified the figure as suggested.**

Figure 4: why only show NH, but not HMA like other sections? Does HMASR not provide SWE?

**In situ SWE measurements are not available for the HMA region. Though HMASR provides SWE, it is still model outputs with unknown uncertainties, not the same as those from snow course and airborne gamma data shown in Figure 4.**

**We feel the temperature and precipitation comparison in Figure 3b and the number of wet days shown in Figure 10 (now Figure 11) are sufficient to explain the differences in simulated SCF over the HMA region.**

Figure 4: perhaps replace 'mount' and 'flat' with 'Mountainous Regions (sigma > 200m)' and 'Flat Regions (sigma < 200m)'

**Thanks for your suggestions. We have changed the title to 'Mountainous Regions' and 'Flat Regions', to be consistent with titles used for other figures.**

Figure 4: how do errors compare to magnitude of SWE, e.g. figure 2 d), which shows maximum values of 115?

**The maximum SWE (the mean Jan-Mar SWE) is over 1000 mm in the mountain regions. We have modified Figure 2d (AR7) to show more range for SWE.**

[Figure]

**Figure AR7**. **Distribution of in situ reference data. (a) Number of monthly 1°x1° grid cells with reference data during 1980-2014 (each monthly 1° grid with reference data is a data point), (b) Number of months during Nov-May 1980-2014 with reference observations by 1° grid. (c) Temporal distribution of raw in situ SWE observations. (d) Mean February-March reference SWE for grid cells with at least 5 months of data. Vertical lines in (a) and (c) indicate Nov-May period used in the analysis.**

Line 403: CTL and SL12 do not cause any albedo feedbacks to cause changes in SWE (via surface energy balance)?

**Thanks for noting this. Yes, there is snow-albedo feedback, which affects simulated SWE in the model. We have modified the sentence (details can be found to our reply to RC1).**

Figure 5: improvements mainly in 2nd half of snow season for mountainous regions; is this due to the ablation part of the SL12 parameterization?

**Yes. The parameterization for ablation in SL12 accounts for sub-grid topographic variability.**

Figure 5: SL12 shows similar results in the NA and EA flat regions (the flat region biases begin early in the season (around Dec/Jan)), can that be improved by calibrating with the parameters in eqn 3?

**Yes, that can be improved by tuning parameters in Eq. (3), details and an example of results (Fig. AR8) can be found in our reply below for a related comment. However, we choose not to use the tuned parameters in CLASSIC, which we explained in our reply above to your main comments.**

Figure 6: I would use a colormap that was not white in the middle for panels a) and b). Perhaps simply linear white-to-blue?

**Thank you for your suggestion. In the revised manuscript, we have applied a linear colorbar to the top panels of Figures 6 and 7.**

Figure 7: SWE evaluation for HMA in section 4 would help understand differences in forcing data. Is the cruja/era5 overestimate due to a SWE high bias, or does it come from the SCF parameterizations?

**Given that the same CTL/SL12 schemes were used across all model runs, the substantial difference in simulated SCF in the run forced by GSWP3W5 (compared to the other two) indicates that the primary cause of the discrepancy is the difference in the forcing data. To improve clarity and logical flow, we have included the following text in the revised manuscript (Lines 603-606).**

**"Given that all three simulations use the same CTL parameterization (Fig. 7d–f), the substantial differences in simulated SCF, particularly in the GSWP3W5-forced run, suggest that the primary source of the discrepancy lies in the forcing data. This will be discussed further in Section 5."**

**In addition, high SWE bias in ERA5 was well documented in previous studies (Liu et al., 2022; Orsolini et al., 2019), which suggested that excessive snowfall in ERA5 contributed to overestimation of SND, SWE, and SCF across HMA (noted in Lines 779-782 of our manuscript).**

**Liu, Y., Fang, Y., Li, D., and Margulis, S. A.: How well do global snow products characterize snow storage in High Mountainous Asia? Geophysical Research Letters, 49, e2022GL100082. https://doi.org/10.1029/2022GL100082, 2022.**

**Orsolini, Y., Wegmann, M., Dutra, E., Liu, B., Balsamo, G., Yang, K., de Rosnay, P., Zhu, C., Wang, W., Senan, R., and Arduini, G.: Evaluation of snow depth and snow cover over the Tibetan Plateau in global reanalyses using in situ and satellite remote sensing observations, The Cryosphere, 13, 2221–2239, https://doi.org/10.5194/tc-13-2221-2019, 2019.**

Line 441: SL12 is shown to perform slightly worse than CTL in fall (SON) in mountainous regions but not flat regions in NA. What might cause this? Is it more due to the accumulation equation or the ablation equation?

**Thank you for your comments and suggestions. This is mainly due to the accumulation formula, which can be improved by increasing the $k_{acc}$ parameter in Eq. (1), details can be found in Section 5.2 (Lines 792-801).**

For the flat regions, the spring bias is similar for both SL12 and CTL. What does that say about SL12, i.e. would a more rapid SCF decrease improve the results? Does that imply that equation 3 is not optimal for flat regions, and the 1/sigma_topo behavior might be too large for small sigma_topo?

**We conducted sensitivity experiments by varying the value of 200 in the numerator of Eq. (3) to 50, 100, and 300. Among the tested configurations, the following combination produced the smallest overall SCF bias (Fig. AR8): for grid cells with $\sigma_{topo} \geq 100m$, a numerator of 200 was used; for cells with $\sigma_{topo} < 100m$, a numerator of 50 was applied. The map on the right (using the modified $N_{melt}$) shows reduced positive bias in flat regions compared to the one on the left (using the default $N_{melt}$).**

[Figure]

**Figure AR8. Bias in simulated SCF using the SL12 scheme with the default parameter (left) and the modified parameter (right). SCF from MODIS was used as the reference.**

**However, we chose not to use the tuned parameters in CLASSIC, as explained in our response to your main comments above. For clarification, we have added the following sentence to Section 5.2 (Lines 813–815).**

**"In addition, it may not be ideal to over-tune the model to a specific observational estimate which may still have uncertainties (Section 5.3)."**

Line 664: does the 'wet day' dependence indicate that one of the accumulation / ablation equations in SL12 has a bigger impact on the SCF evolution?

**The number of wet days influences the frequency of new snowfall, thereby directly affecting SCF during the accumulation period (Eq. 1). It also determines the amount of snow stored on the ground, which in turn impacts SCF through the ablation processes (Eq. 2). The dominant process controlling SCF evolution likely depends on the regional climate. In cold regions, where melting is rare during the accumulation season, SCF evolution is primarily governed by the accumulation process. In contrast, in intermediate and warmer climates where accumulation and melt cycles are**

**more frequent, both accumulation and ablation processes may contribute comparably to SCF evolution.**

Comments from Referee #3

**We thank Referee #3 for their helpful comments. Our replies to their comments are shown in bold below.**

Review of manuscript "Impact of topography and meteorological forcing on snow simulation in the Canadian Land Surface Scheme Including Biogeochemical Cycles (CLASSIC)" By Libo Wang et al. This paper discusses the impacts of using an alternative snow cover fraction scheme in the Canadian Land Surface Scheme. The parameterization adopted here was developed by Swenson and Lawrence (2012) and is currently used in other land models (e.g., CESM) and appears more physically realistic than the one currently used in the model here. In particular, the new formulation accounts for topographic effects while the previous one did not. The authors evaluate the new model setup using multiple datasets and satellite observations finding that the performance of the model in the new configuration overall improves. The authors also explore the sensitivity of the (offline) land model forced by three meteorological forcing datasets.

While not presenting a new snow scheme or parameterization but implementing an existing one in a different land model, I think the paper is interesting and well written, and would be of interest to the readership of GMD. Furthermore, the authors perform a comprehensive evaluation of multiple land surface variables which is useful in characterizing the performance of their model forced by multiple reanalysis datasets. Below are my comments on the paper, which the authors should address or respond to before the paper is considered for publication. The style and writing is overall good, although there are a few typos or sentences that could be improved. Please see a list of recommendations below.

**Thank you for your overall positive review of our manuscript.**

**Main comments**

Why was this particular parametrization chosen? I understand it is a clear improvement and probably one of the best options in the literature, but I still feel the paper lacks a brief review of what existing options for SCF parameterizations are, and what is used in other snow models. Given the scope of the paper, I think this would be useful to the reader. E.g., I believe adding some detail to the text currently at lines 81 - 87 would improve the discussion and help justify the choice made here.

**Thanks for your suggestions. We have expanded on the information provided at lines 81-87 to the following in the revised manuscript (Lines 83-99):**

**"Some early SCF parameterizations assumed a linear increase in snow cover with snow depth or snow water equivalent (SWE), reaching 100% SCF once a specified threshold was met (e.g., Verseghy, 1991; Bonan, 1996). Other approaches incorporated surface roughness length into the SCF–SND (or SWE) relationships (e.g., Dickinson et al., 1986; Marshall and Oglesby, 1994), and distinguished SCF estimates between bare ground and vegetated areas (Douville et al., 1995; Yang et al., 1997). Large uncertainties in modeled SCF from these early schemes motivated efforts to refine parameterizations by accounting for terrain heterogeneity or incorporating sub-grid snow distribution (Roesch et al., 2001; Liston, 2004). More recent SCF parameterizations have included snow density (e.g., Niu and Yang, 2007; Lalande et al., 2023) and land cover type (e.g., He et al.,**

**2023), with some schemes adopting separate formulations for snow accumulation and melt periods (Swenson and Lawrence, 2012). Some of these parameterizations account for topographic effects of sub-grid terrain on SCF (e.g., Douville et al., 1995; Roesch et al., 2001; Swenson and Lawrence, 2012; Lalande et al., 2023), which have been shown to be crucial for accurate SCF simulation in mountainous regions (Miao et al., 2022)."**

Figure 3 does not seem to have good quality in the pdf. Please improve the quality (e.g., increase resolution, use a larger font, ..). Also, I struggle to fully understand the implications of this comparison: It seems for precipitation ERA5 has a very large (compared to the other datasets) positive bias on mountain regions. I wonder if it would be better to present a measure of relative error rather than the absolute difference in (mm/month) for the entire dataset?

**We apologize for the poor quality of the figures. The quality was fine in the original Microsoft Word version of the manuscript but deteriorated after converting to the pdf file. We will ensure the figures will all have high quality in the revised manuscript.**

**First, we note that the larger bias for ERA5 with respect to the CRU reference data as shown on Figure 3 reflects that both CRUJRA and GSWP3-W5E5 forcing datasets already incorporate CRU reference data for bias-correction (see Section 5.1 for a discussion on why there are still residual biases in these two datasets). Beyond that point, our interpretation of the results is based on the assumption that the relative precipitation amounts are accurate, but that there is uncertainty in how well the CRU reference data reflects precipitation amounts in mountain regions. Figure 3 demonstrates that in a relative sense, ERA5 has more precipitation in mountain regions compared to CRUJRA and GSWP3-W5E5 (this precipitation would typically be snowfall given the seasonal timing and mountain location). Despite this additional snowfall, CLASSIC SWE output forced by ERA5 still has a negative bias in mountain regions (Fig. 4), although less than the larger negative SWE biases identified for both CRUJRA and GSWP3-W5E5 forced output. As stated at lines 716-722 in Section 5.1, we interpret these two apparent contradictions to mean that the CRU precipitation measurements have a sampling bias (supported by Nijssen et al., 2001; Adler et al, 2003; Shi et al., 2017) such that the hemispheric ERA5 precipitation bias in mountain regions is actually smaller in reality than those from CRUJRA and GSWP3-W5E5 (i.e. all forcings have too little precipitation in mountain regions but ERA5 has the closest to accurate amount).**

**To provide context for the bias plots in Figure 3, we plotted air temperature and precipitation from the three forcing datasets over the NH (a and b) and the HMA (c and d) regions. Figure AR9 (b) and (d) show that ERA5 has more precipitation in mountain regions compared to CRUJRA and GSWP3-W5E5. However, it is hard to tell the differences in temperature in the NH regions shown in Figure AR9 (a), which were illustrated better in the bias plots shown in Figure 3(a). We have included Figure AR9 in the supplement (Fig. A2) in the revised manuscript.**

[Figure]

**Figure AR9. Monthly mean air temperature (a and c) and precipitation (b and d) in the NH mountainous (solid line) and flat (dashed line) regions (a and b) and the HMA mountainous regions (c and d) over the 1980-2014 period.**

L428: "Overall, ERA5-SL12 outperforms the other two model runs with lower bias and better correlation in mountainous regions and it shows similar performance as CRUJRA-SL12 in flat regions" -> Is this in part due to CLASSIC underestimating SWE and ERA5 overestimating winter precipitation compared to other datasets, as shown in Figure 3?

**This statement is consistent with the interpretation provided that precipitation in all the forcings is biased low, but that ERA5 precipitation is the least biased and most accurate. This additional precipitation / snowfall, and therefore SWE improves the simulation of other downstream land surface variables in mountain regions (and that ERA5 and CRUJRA forcings lead to comparable performance in flat regions).**

Figure 5: As for Figure 3, please also improve the quality / resolution of this figure.

**We will ensure that all figures are of high quality in the revised manuscript.**

It seems the standard deviation of evolution is an important parameter for the parameterization used here. At what scales / resolution was it computed, and is this consistent with previous applications and with Swenson and Lawrence (2012)? Does this matter?

**ETOPO1 elevation data at 1 arc-minute resolution (~1.85 km at the equator; NOAA, 2009) was used to compute the standard deviation of topography in our study. Swenson and Lawrence (2012) did not provide information about the elevation data used in their study.**

**To assess the impact of DEM resolution, we compared $\sigma_{topo}$ derived from two DEM datasets: ETOPO1 (1 arc-minute resolution) and ETOPO2022 (15 arc-second resolution, ~500 m). The results show that the differences are limited in extent, primarily concentrated along the edges of mountain ranges. We also performed a test simulation using $\sigma_{topo}$ derived from ETOPO2022 and compared modelled SCF with that from a run using $\sigma_{topo}$ derived from ETOPO1. The maximum difference was less than 5%. Thus, the resolution of the DEM data has limited impact on the calculation of sub-grid topographic variability and the simulated SCF.**

**Baes on the above, we have included the following text in Section 5.3 in the revised manuscript (Lines 843-853):**

**"Additional uncertainties may arise from the elevation data used to compute the standard deviation of sub-grid topography ($\sigma_{topo}$), particularly related to its spatial resolution. We compared $\sigma_{topo}$ derived from two elevation datasets: ETOPO1 (1-arc-minute resolution, used in this study) and ETOPO2022 (15-arc-second resolution). The results indicate that the differences are limited in spatial extent and are primarily concentrated along edges of mountain ranges. To assess the impact on model results, we conducted a test simulation using $\sigma_{topo}$ derived from ETOPO2022 and compared the simulated SCF with that from a run based on $\sigma_{topo}$ derived from ETOPO1. The maximum difference in SCF between the two runs was less than 5% (not shown). These findings suggest that the resolution of the elevation data has a limited effect on the calculation of sub-grid topographic variability and simulated SCF, consistent with sensitivity tests reported by Lalande et al. (2023)."**

Figure 2: Could you specify the units of the two colorbars?

**We have added units to the colorbars of Figure 2 in the revised manuscript.**

**Minor comments or typos**

L64: comma after "Therefore"

**Done.**

L66: "Snow depth (SND) varies at scales from about 10 to 100 m" reads unclear - could you maybe state more precisely the range of scales at which it can vary? Do you mean "up to" 10-100m? Surely it is variables at larger scales too.

**Thanks for noting this. We have modified the sentence to the following in the revised manuscript (Lines 66-68):**

**"In principle, snow depth (SND) should vary considerably at sub-grid scales of global climate models as a result of multiple heterogeneities in land cover, terrain, and meteorological conditions (Liston 2004)."**

L94: "so that" neither was implemented?

**Thanks for the suggestion. We have added "so that" in the revised manuscript.**

L131 "each with and without snow cover " -> do you mean to say that each of these fractions can be partially covered by snow? Please clarify

**The four sub-areas are: vegetated, bare soil, vegetated with snow cover, and snow cover over bare soil. They are calculated based on the fractional coverage of the vegetation categories and SCF. The snow-covered and snow-free areas change dynamically at each time step. We have modified the sentence in the revised manuscript (Lines 149-151):**

**"As a first-order treatment of subgrid-scale heterogeneity, each grid cell is divided into four sub-areas: vegetated, bare soil, vegetated with snow cover, and snow cover over bare soil."**

L132: "which includes" -> please clarify: does the snow layer include these processes, or the snow scheme?

**Thanks for noting this. We meant "snow scheme" here, we have modified the sentence to the following in the revised manuscript (Lines 152-154):**

**"Snow is represented as a single layer, and canopy snow processes such as interception, unloading, sublimation and melt are included (Bartlett et al., 2006; Verseghy et al., 2017)."**

L150: "increasing or decreasing this threshold value" -> which threshold value, the minimum snow depth 0.1m? It seems strange this parameter has little effect since it would significantly affect SFC and surface albedo. Could you comment on this?

**Yes, the minimum snow depth 0.1m.**

**The small effect of varying this threshold is mainly because areas with snow depth from 5-10 cm and from 10-15 cm are very limited in the model.**

L155: "but also on month of the year" -> would it be more correct to say on the age of the snowpack?

**It is not about the age of the snowpack, but whether snow mass is increasing (accumulation) or decreasing (ablation). We have modified the sentence to the following in the revised manuscript (Line 177):**

**"but also whether snow mass is increasing (accumulation) or decreasing (ablation)."**

L612: Add comma: "Yet, among …"

**Done.**